# Subjective socioeconomic status and income inequality are associated with self-reported morality across 67 countries

Christian T. Elbæk [1] ✉, Panagiotis Mitkidis [1,2], Lene Aarøe [3] & Tobias Otterbring [4]

Individuals can experience a lack of economic resources compared to others, which we refer to as subjective experiences of economic scarcity. While such experiences have been shown to shift cognitive focus, attention, and decision-making, their association with human morality remains debated. We conduct a comprehensive investigation of the relationship between subjective experiences of economic scarcity, as indexed by low subjective socioeconomic status at the individual level, and income inequality at the national level, and various self-reported measures linked to morality. In a pre-registered study, we analyze data from a large, cross-national survey ($N = 50{,}396$ across 67 countries) allowing us to address limitations related to cross-cultural generalizability and measurement validity in prior research. Our findings demonstrate that low subjective socioeconomic status at the individual level, and income inequality at the national level, are associated with higher levels of moral identity, higher morality-as-cooperation, a larger moral circle, and increased prosocial intentions. These results appear robust to several advanced control analyses. Finally, exploratory analyses indicate that observed income inequality at the national level is not a statistically significant moderator of the associations between subjective socioeconomic status and the included measures of morality. These findings have theoretical and practical implications for understanding human morality under experiences of resource scarcity.

Subjective experiences of economic scarcity, hereinafter defined as the perceived lack of economic resources as a result of social comparison, are a structural characteristic of modern societies and a persistent cause of concern[1,2]. Such experiences are not necessarily the same as the actual circumstances of having low wealth or low income, and may not always correlate strongly with objective indicators of scarcity[3,4]. Nevertheless, subjective experiences of scarcity are increasingly indexed by low subjective socioeconomic status (SES) at the individual level and income inequality (GINI) at the national level[5–9].

An expanding body of literature has found that subjective experiences of economic scarcity, as well as experiences of other types of resource deprivation like hunger or thirst, can shift human cognition and judgment, changing intentions and subsequent behavior[7,10–15]. Specifically, subjective experiences of economic scarcity have been shown to alter executive functioning and fluid intelligence, increase future discounting, and lead to more impulsive and risk-seeking behavioral manifestations[7,10–14,16–19], while simultaneously decreasing psychological well-being[4]. Yet, the implications of such experiences for

[1]Department of Management, Aarhus University, 8210 Aarhus V, Denmark. [2]Social Science Research Institute, Duke University, 27701 Durham, NC, USA. [3]Department of Political Science, Aarhus University, 8000 Aarhus C, Denmark. [4]Department of Management, University of Agder, 4630 Kristiansand, Norway. ✉ e-mail: chel@mgmt.au.dk

*moral* judgment and decision-making remain highly debated and extant findings are contradictory[20].

At a general level, moral decision-making is rooted in the concept of human morality, which could be defined as "a collection of biological and cultural solutions to the problems of cooperation recurrent in human social life" (ref. 21, p. 47). This implies that morality is essential for promoting collaboration[22–25], but also that when individuals engage in moral decision-making, they not only consider the direct outcome, but also how the moral valence of such decisions might affect both their own and other people's view of themselves. As such, moral decision-making is a multidimensional term.

Existing research on the relationship between subjective experiences of economic scarcity and human morality appears to be split between two theoretical paradigms, with one predicting mainly *negative* outcomes on moral judgment and decision-making, and with the other largely arguing for the reverse. Concerning research suggesting negative effects, a selection of studies has found that resource-deprived individuals act greedier[18,26], are more inclined to engage in dishonest behaviors to obtain resources[27–30], exhibit less prosocial intentions[31,32], and tend to donate less of their personal income to charitable giving[33,34]. These findings may reinforce destructive but prevalent stereotypes and folk beliefs depicting individuals with low SES as irresponsible, dishonest, and "milking the system" (see ref. 35. for a review).

In contrast to this line of literature, other studies have suggested that individuals who subjectively experience economic scarcity are more inclined to emphasize the importance of moral values such as reciprocation, to act in less unethical ways, and to exhibit more prosocial responses[9,20,36–43]. One of the most prominent studies from this body of research has shown that individuals who perceive themselves as being of lower social class act in a more generous, charitable, helpful, and trusting way compared to those who perceive themselves as being of higher social class[37]. The main theoretical argument behind such findings is that subjective experiences of economic scarcity, in the form of low social class perceptions, are assumed to increase individuals' contextual orientation in their display of moral behavior[36]. That is, individuals who perceive themselves as having lower social class demonstrate an externally-focused cognitive and relational orientation, which enables them to exhibit greater empathy, more compassion, and more prosocial behavior toward their peers[36], because they know that their social relationships can aid them in achieving better prospective life outcomes[44]. Several findings have supported these results by showing that individuals with lower incomes elicit greater prosociality, especially toward peers in the same situation[40], as manifested through more altruistic actions[37] and a greater proportion of income donated to charity compared to higher-income individuals[45]. However, other studies have not been able to replicate the positive relationship between subjective experiences of economic scarcity—in terms of low social class perceptions—and prosociality[31,46]. Some studies also suggest that there is a large degree of country-level variability in this relationship[31], indicating that the effects of subjective experiences of economic scarcity on prosociality might be highly context dependent, consistent with the notion that macro-level economic inequality might be an important moderator for this relationship[6,47,48].

Lastly, and of particular importance for the current investigation, prior studies have limitations related to cross-cultural generalizability, statistical conclusion validity, and measurement validity. First, most extant studies have limited generalizability as they have predominantly relied on data from a single country—the United States—despite indications in past research of potential contextual sensitivity by variations in time, culture, or location[49–52]. Contextual sensitivity might thus explain part of the inconsistencies in the literature[26,31]. Still, no prior study has implemented a cross-national research design to conduct a systematic, large-scale test of the relationship between subjective experiences of economic scarcity and morality.

Second, statistical conclusion validity is also limited in the literature as many extant studies rely on underpowered laboratory experiments[53–58]. This is important as studies have found both replicability problems in the form of null-findings and results that are in the opposite direction of those reported in the original research[46].

Third, extant studies on the relationship between subjective experiences of economic scarcity and morality typically focus on a single measure of moral decision-making (e.g., prosocial behavior[31,40] or unethical behavior[27,38]). Yet, moral decision-making is a multidimensional construct that may include both perceptions of moral identity and character, moral values, prosocial intentions to benefit others, and moral circle defined as the boundary we draw around individuals who we think deserve moral consideration[59]. By studying only one or a few indicators of moral decision-making in isolation, studies decrease measurement validity and have a higher risk of not detecting if subjective experiences of economic scarcity affect some types of morality but crowds out other types.

To address the mixed findings and to increase cross-cultural generalizability, statistical conclusion validity, and measurement validity in the literature, we conduct a comprehensive pre-registered test of the relationship between subjective experiences of economic scarcity and measures linked to moral judgment and decision-making. We rely on an extensive and partly representative cross-national survey ($N = 50,396$ across 67 countries, including 28 nationally representative samples in terms of age and gender; see Fig. 1 for a country and region overview of the sample). This research design provides an opportunity to achieve four important main objectives. Specifically, this research design (1) maximizes cross-cultural generalizability in comparison to previous studies, which have typically been restricted to data from a single country; (2) increases statistical conclusion validity by ensuring a statistically well-powered test of the relationship between morality and perceptions of economic scarcity as indexed by low subjective SES at the individual level and income inequality at the national level; and (3) allows us to examine whether the level of economic inequality at the national level moderates the relationship between subjective SES and morality, while also increasing measurement validity by including four measures associated with morality.

Regarding the measurement of morality, we expand previous research by using four measures associated with different aspects of moral decision-making that are considered essential to human morality. These are (1) *Moral Identity*, which measures how important and central moral issues are to a person's self-concept[60]; (2) *Morality-as-Cooperation*, which concerns the moral valence of seven cooperative behaviors considered to be "morally good" across cultures (e.g., helping kin, reciprocating, dividing distributed resources) and thus measures the individual's judgment of the importance of these behaviors[21]; (3) the size of an individual's *Moral Circle*, which indicates the self-reported number of individuals and entities in the world considered to be worthy of moral consideration[61]; and (4) *Prosocial Intentions to Benefit Others*, which captures the amount of monetary resources an individual reports being willing to donate to a national and international charity if given a daily median income in one's respective country[62,63]. In sum, instead of examining only one aspect of moral decision-making, we expand previous research by investigating how subjective experiences of economic scarcity might influence multiple fundamental measures associated with morality, thereby increasing measurement validity and providing more opportunities to compare patterns of findings across indicators within the same study.

For our main independent variables on subjective experiences of economic scarcity, we use (1) the MacArthur socioeconomic ladder scale to measure subjective SES at the individual level; and (2) GINI coefficients from the World Bank[64] at the national level as an indirect measure of macro-level differences in perceptions of subjective economic scarcity. For the MacArthur scale, individuals are asked to place themselves on a ladder with 11 steps, where selecting the lowest step

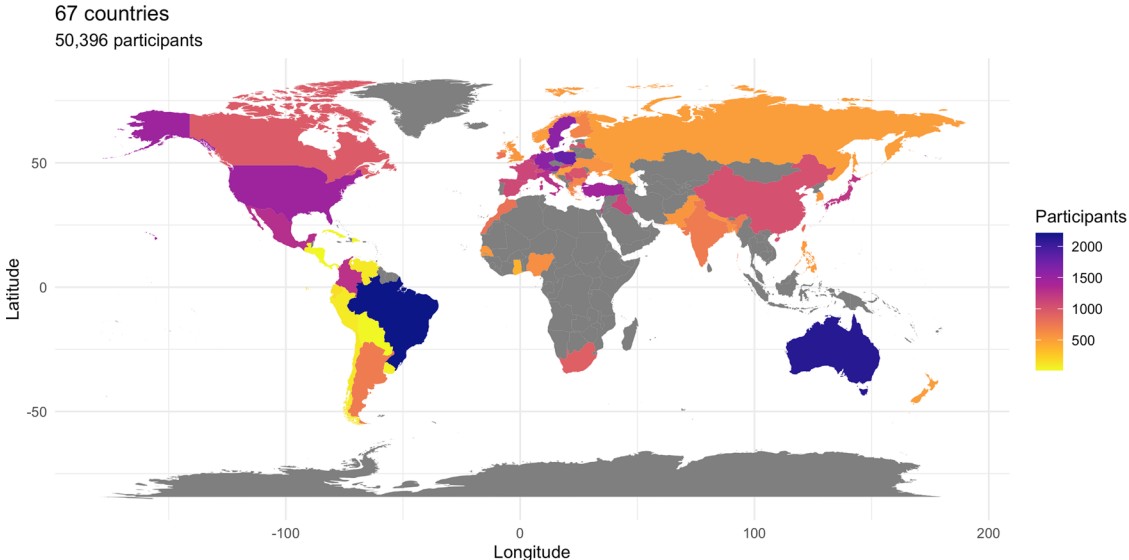

**Fig. 1 | World map of ICSMP survey.** The map highlights the countries and regions where data were collected for the ICSMP Survey[62,63]. Sample sizes are scaled to color. Gray areas identify areas where it was not possible to obtain samples.

(*1*) indicates that you would place yourself among the people with the least financial resources, least education, and least attractive jobs in your respective country, while placing yourself on the top of the ladder (*11*) indicates that you place yourself among the people with the most financial resources, best education, and most attractive jobs. Hence, this measure is subjective and oriented toward differences in the perception of scarcity compared to more objective individual-level measures (such as personal income or household income) that are rather oriented toward the potential material sources of scarcity. This distinction is important as prior research has found that subjective experiences of scarcity can shift cognitive attention and alter decision-making strategies more than extreme, absolute scarcity (i.e., extreme poverty)[3,10–12]. The MacArthur scale has exhibited strong construct validity[65] and strong predictive validity regarding outcomes that are often associated with experiencing economic scarcity, such as lower health status[66,67] and lower subjective well-being[68]. Moreover, recent work has provided evidence suggesting that a reason for the scale's high predictive validity in such domains is its ability to measure two central constructs: economic circumstances and social class[69]. Therefore, we rely on the MacArthur measure of subjective SES to map individual-level perceptions of economic scarcity to gain a more nuanced understanding of the link between subjective experiences of scarcity and morality.

With respect to our use of the GINI Index as our national-level indicator of subjective experiences of economic scarcity, this measure is an indicator of the dispersion of financial resources among individuals in a specific economy. It ranges from 0 to 100, where 0 indicates that every single individual in the respective economy has the same income, while 100 indicates that one and only one person earns the entire income. Although the GINI is an objective national-level measure of dispersion of financial resources, in the current investigation we build on prior work arguing that economic inequality can elicit perceptions of economic scarcity[6,70–73] and thus conceptualize this measure as a macro-level indicator that indirectly probes differences in subjective experiences of economic scarcity[74].

Prior research has found that higher income inequality increases social comparison in a given context[75], which can exacerbate social class divisions by saliently outlining one's standing on the "social ladder"[70]. In turn, this breeds competition for resources[72], increases risk-taking[76], heightens anxiety associated with status striving[77], and probes perceptions of relative deprivation[71], thereby explaining why income inequality has been discussed as an important factor linked to psychological differences in perceptions of economic scarcity[6,76,78,79]. Accordingly, in the current investigation, the GINI index is used as a macro-level indicator of the magnitude of exposure to salient differences in income, thereby indirectly probing differences in subjective experiences of economic scarcity[74].

Because the GINI here indexes subjective experiences of scarcity—as triggered by objective wealth discrepancies—it is a more indirect indicator than our individual level SES measure, which is directly focused on subjective experiences of economic scarcity. Some individuals across our studied 67 countries might not be considered to live in actual economic scarcity as indexed by objective measures (e.g., household income), and prior meta-analytic estimates only indicate a moderate association ($r = 0.32$) between subjective and objective measures of SES[68]. Nevertheless, the use of subjective experiences of economic scarcity allows us to examine how individuals who perceive themselves as "having too little"[4] respond on essential indicators of human morality.

In this work, we show, across 67 countries, that individual experiences of economic scarcity in the form of low subjective SES are associated with higher self-reported levels of Morality-as-Cooperation as well as Prosocial Intentions in the form of hypothetical donations toward national and international charities. Moreover, utilizing the GINI coefficient as a crude measure of macro-level experiences of economic scarcity, we show that this relationship, at least in part, holds even at the national level, such that individuals living in countries with high economic inequality, and thus a greater degree of experiences of economic scarcity, report a stronger Moral Identity and a greater Moral Circle. Our research highlights that individual and macro-level experiences of economic scarcity are associated with multiple dimensions of self-reported human morality.

## Results

We begin by analyzing the relationship between subjective experiences of economic scarcity and moral judgment using multi-level modeling, both at the individual level (SES) and at the country level (GINI), while also testing whether there might be any country-level differences in the individual-level relationship. Table 1 reports full models for our four dependent measures, for all 67 countries (*All*) and for countries where samples were nationally representative with respect to age and gender (*Nat. Rep.*). We report standardized

**Table 1 | Multilevel models**

| | Moral Identity | | Morality-as-Cooperation | | Moral Circle | | Prosocial Intention | |
|---|---|---|---|---|---|---|---|---|
| | **All**<br>Std. β (P)<br>[95% CI]<br>t(df) | **Nat. Rep.**<br>Std. β (P)<br>[95% CI]<br>t(df) | **All**<br>Std. β (P)<br>[95% CI]<br>t(df) | **Nat. Rep.**<br>Std. β (P)<br>[95% CI]<br>t(df) | **All**<br>Std. β (P)<br>[95% CI]<br>t(df) | **Nat. Rep.**<br>Std. β (P)<br>[95% CI]<br>t(df) | **All**<br>Std. β (P)<br>[95% CI]<br>t(df) | **Nat. Rep.**<br>Std. β (P)<br>[95% CI]<br>t(df) |
| *Predictors* | | | | | | | | |
| Subj. SES | −0.14 (<0.001)<br>[−0.15, −0.13]<br>−5.54 (45832) | −0.17 (<0.001)<br>[−0.18, −0.16]<br>4.52 (27948) | −0.07 (<0.001)<br>[−0.08, −0.06]<br>−0.36 (45946) | −0.08 (<0.001)<br>[−0.09, −0.07]<br>4.31 (28052) | 0.01 (0.003)<br>[0.00, 0.02]<br>1.09 (46646) | −0.02 (<0.001)<br>[−0.01, −0.03]<br>−0.31(28289) | −0.08 (<0.001)<br>[−0.09, −0.07]<br>−2.57(45649) | −0.09<br>( < 0.001)<br>[−0.11, −0.08]<br>−2.77 (28317) |
| GINI Index | 0.13 (0.003)<br>[0.04, 0.21]<br>2.98 (45832) | 0.22 (<0.001)<br>[0.12, 0.31]<br>6.83 (27948) | 0.03 (0.479)<br>[−0.05, 0.10]<br>1.53 (45946) | 0.13 (0.007)<br>[0.03, 0.22]<br>4.70 (28052) | 0.08 (0.003)<br>[0.03, 0.14]<br>2.94 (46646) | 0.03 (0.387)<br>[−0.04, 0.10]<br>0.40 (28289) | 0.06 (0.118)<br>[−0.02, 0.11]<br>1.78 (45649) | 0.01 (0.873)<br>[0.12, 0.16]<br>0.02 (28317) |
| Gender[Female] | 0.12 (<0.001)<br>[0.10, 0.14]<br>13.33 (45832) | 0.13 (<0.001)<br>[0.11, 0.16]<br>11.86 (27948) | 0.09 (<0.001)<br>[0.07, 0.11]<br>10.03 (45946) | 0.09 (<0.001)<br>[0.07, 0.11]<br>7.57 (28052) | 0.17 (<0.001)<br>[0.15, 0.19]<br>18.45 (46646) | 0.17 (<0.001)<br>[0.15, 0.20]<br>14.82 (28289) | 0.15 (<0.001)<br>[0.13, 0.17]<br>16.87(45649) | 0.14 (<0.001)<br>[0.12, 0.16]<br>12.43 (28317) |
| Age | 0.04 (<0.001)<br>[0.03, 0.05]<br>8.44 (45832) | 0.02 (0.003)<br>[0.01, 0.03]<br>3.01 (27948) | 0.02 (0.002)<br>[0.00, 0.02]<br>3.14 (45946) | 0.02 (0.002)<br>[0.01, 0.03]<br>3.10 (28052) | 0.07 (<0.001)<br>[0.06, 0.08]<br>13.99 (46646) | 0.05 (<0.001)<br>[0.04, 0.06]<br>8.81 (28289) | 0.05 (<0.001)<br>[0.04, 0.06]<br>10.02 (45649) | 0.06 (<0.001)<br>[0.05, 0.07]<br>10.31 (28317) |
| SES × GINI | −0.00 (0.792)<br>[−0.01, 0.01]<br>−0.26 (45832) | −0.05 (<0.001)<br>[−0.06, 0.04]<br>−8.82 (27948) | −0.01 (0.012)<br>[−0.02, −0.00]<br>−2.51 (45946) | −0.04 (<0.001)<br>[−0.05, 0.03]<br>−6.25 (28052) | −0.00 (0.0603)<br>[−0.01, 0.01]<br>−0.52 (46646) | −0.01 (0.400)<br>[−0.01, 0.02]<br>0.84 (28289) | −0.00 (0.381)<br>[−0.01, 0.01]<br>−0.88 (45649) | −0.00 (0.617)<br>[−0.01, 0.01]<br>0.50 (28317) |
| *Random effects* | | | | | | | | |
| $\sigma^2$ | 177.01 | 171.47 | 122.29 | 117.37 | 26.08 | 25.62 | 1104.56 | 1112.33 |
| $\tau_{00}$ | 24.94 country | 15.04 country | 13.03 country | 8.22 country | 1.23 country | 0.98 country | 131.69 country | 131.37 country |
| ICC | 0.12 | 0.08 | 0.10 | 0.07 | 0.04 | 0.04 | 0.11 | 0.11 |
| N | 67 country | 28 country | 67 country | 28 country | 67 country | 28 country | 67 country | 28 country |
| Observations | 45840 | 27956 | 45954 | 28060 | 46654 | 28297 | 45657 | 28325 |
| Marg. $R^2$/ Cond. $R^2$ | 0.040 / 0.158 | 0.076 / 0.150 | 0.008 / 0.103 | 0.023 / 0.087 | 0.017 / 0.061 | 0.011 / 0.047 | 0.018 / 0.123 | 0.016 / 0.120 |
| AIC | 367674.974 | 223308.308 | 351580.974 | 213497.600 | 284807.550 | 172227.290 | 449780.494 | 279213.302 |

Label "All" denotes models using all 67 countries. Label "Nat. Rep." denotes models using only nationally representative samples. All models are linear-mixed effects models (two-sided).

$\beta$-coefficients allowing for a direct comparison of the effect sizes and model fit statistics. Sample sizes for each model are reported, given that these varied slightly due to some participants being able to refrain from replying to certain measures in the survey (see section; *Methods*). As a robustness check, we also run our models with imputed data where the missing values are estimated using non-parametric random-forest estimations. The results remain robust to this imputation of data and are depicted in Supplementary Table S4.

**Individual-level economic scarcity and self-reported morality**
As illustrated in Table 1, after controlling for age and gender, lower individual-level subjective SES predicts higher Moral Identity ($t(45832) = -5.54$, $p < 0.001$, $\beta = -0.14$, 95% CI [−0.15, −0.13]), higher Morality-as-Cooperation ($t(45946) = -0.36$, $p < 0.001$, $\beta = -0.07$, 95% CI [−0.08, −0.06]), higher Prosocial Intentions to donate to national and international charities ($t(45649) = -2.57$, $p = 0.010$, $\beta = -0.08$, 95% CI [−0.09, −0.07]), and a larger Moral Circle, although this latter association is negligible ($t(46646) = 1.09$, $p = 0.003$, $\beta = 0.01$, 95% CI [0.00, 0.02]). These associations are robust in cross-validations with supervised machine learning algorithms (10-folds, 200 repetitions; Supplementary Table S15). Replicating our analysis with only the 28 nationally representative samples yields comparable results, although the associations become slightly stronger (see Table 1 columns denoted Nat.Rep.). Visualizations of these results are shown in Fig. 2 (see Supplementary Fig. S1 for a visualization using only nationally representative samples) and a country-level summary of the direction of the regression slopes for all 67 countries and for the 28 nationally representative samples, respectively, appears in Table 2 (i.e., the number of countries with positive or negative associations on our focal outcomes).

**Macro-level economic scarcity and self-reported morality**
Consistent with the individual-level results, as seen in Table 1, higher degrees of country-level economic inequality (i.e., GINI) predicts higher individual-level Moral Identity ($t(45832) = 2.98$, $p = 0.003$, $\beta = 0.13$, 95% CI [0.04, 0.21]), and a greater Moral Circle ($t(46646) = 2.94$, $p = 0.003$, $\beta = 0.08$, 95% CI [0.03, 0.13]). This type of country-level economic inequality, however, is not statistically significantly associated with individual-level differences in Morality-as-Cooperation ($t(45946) = 1.53$, $p = 0.479$, $\beta = 0.03$, 95% CI [−0.05, 0.10]) or Prosocial Intentions ($t(45649) = 1.78$, $p = 0.118$; $\beta = 0.06$, 95% CI [−0.02, 0.11]), as is the case with subjective SES. Hence, these results indicate that individuals living in contexts of greater economic inequality attribute greater importance in both symbolizing and internalizing a moral identity (i.e., standing out as a moral individual to peers and thinking of oneself as a moral individual), as well as reporting a larger moral circle. Visualizations of these associations are illustrated in Fig. 3 (see Supplementary Figure S2 for a visualization using only nationally representative samples).

Supporting the notion that the GINI index could be used as an indirect macro-level measure of perceptions of economic scarcity, we find that our measure of subjective SES is significantly associated with higher inequality as indexed by GINI ($t(47315) = -14.79$, $p < .001$, $r = -0.07$, 95% CI [−0.08, −0.06]) and that the lower quantile of our individual-level measure of subjective SES significantly predicts higher inequality ($t(46306) = 11.02$, $p < 0.001$, $\beta = 0.13$, 95% CI [0.11, 0.15]). Although the correlations are not strong, the findings are consistent with the argument in prior research that contexts with higher wealth inequality make people engage more in social comparisons with individuals with greater resources and therefore decreases subjective SES[72,75].

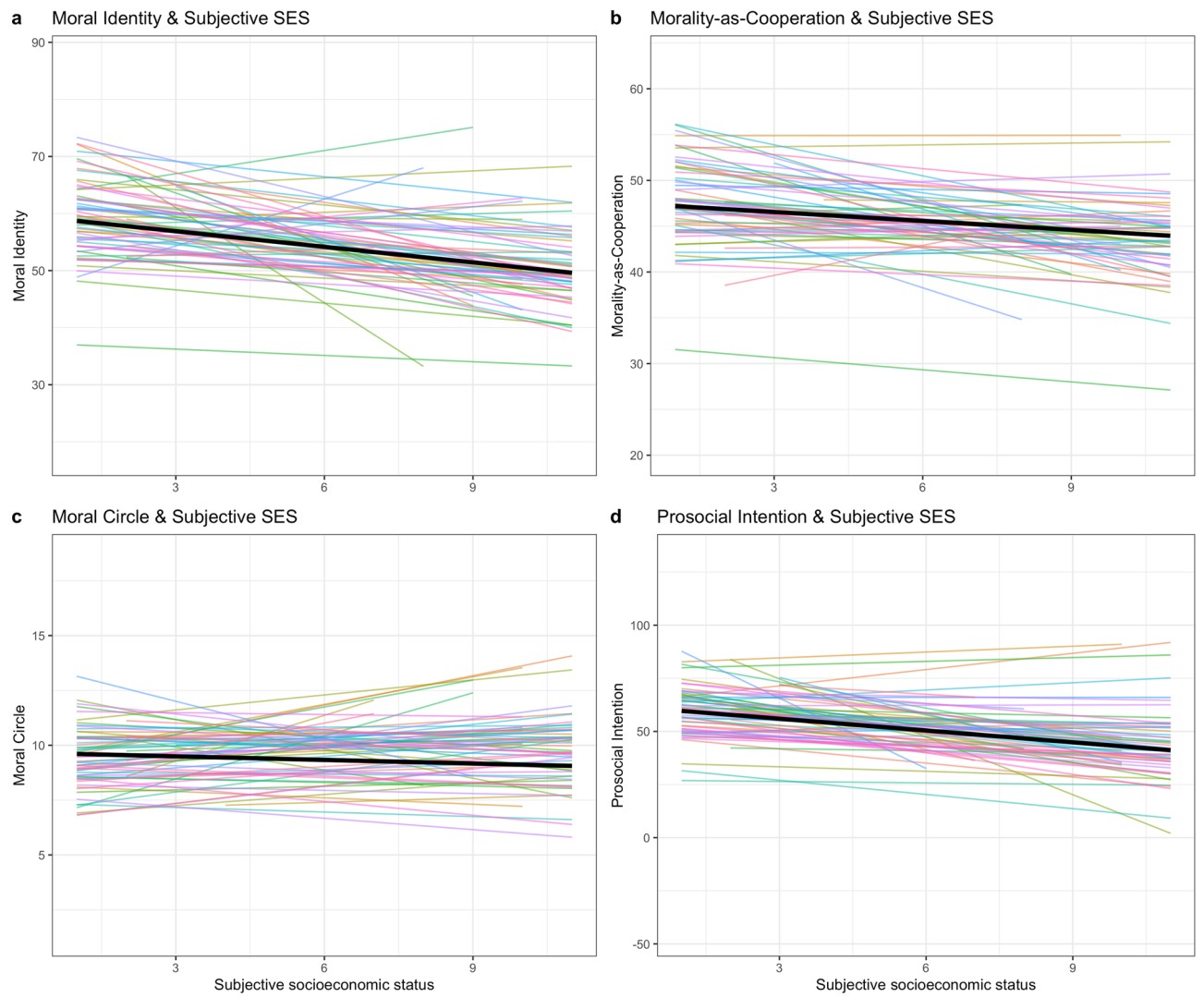

**Fig. 2 | Within-country and between-country associations between Moral Identity, Morality-as-Cooperation, size of Moral Circle, Prosocial Intentions, and Subjective Socioeconomic Status (SES).** For all four panels (**a**, **b**, **c**, **d**), colored lines indicate within-country associations highlighting a main pattern where most associations are negative, while simultaneously outlining the degree of heterogeneity between-countries, as a selection of within-country associations are positive. The bolded black line for each panel indicates the overall relationship across the 67 countries. **a** Association between Subjective SES and individual-level Moral. **b** Association between Subjective SES and Morality-as-Cooperation. **c** Association between Subjective SES and the size of one's Moral Circle, where size indicates the circle of people or other entities for which one is concerned whether right or wrong is done toward them. **d** Association between Subjective SES and Prosocial Intentions, measured as the amount of money (out of a median income) one would be willing to donate to a national and international charity.

Importantly, however, economic inequality does not meaningfully moderate the effect of low subjective SES on any of our indicators of morality (see Table 1). Thus, we do not find evidence that individual differences in subjective SES predict any of our measures of morality differently in more economically unequal countries compared to more economically equal countries. Instead, our results indicate that low subjective SES might have parallel effects with national level income GINI measures of inequality on morality.

### Within- and between country associations

Variance partition coefficients[80] for the models indicated that most of the variance could be attributed to the individual level (Moral Identity = 87.7%, Morality-as-Cooperation = 90.4%, Moral Circle = 95.5%, Prosocial Intentions = 89.3%) with variance at the country-level ranging from 4.4% at the lowest (Moral Circle) to 12.3% at the highest (Moral Identity). We assume that some of this lack of country-level variance is due to common-method variance inflating the estimates at the individual level[81,82]. Additionally, this lack of country-level variance indicates that the relationship between relative economic scarcity, both on the individual level and country level, and moral character and intentions might be fairly robust across different cultural contexts. The cross-cultural robustness of these relationships also aligns with recent findings using the same data to investigate differences in donation responses, in-group favoritism, and age across the 67 countries[83].

**Table 2 | Summary of country-level of regression slopes**

| | Moral Identity | | Morality-as-Cooperation | | Moral Circle | | Prosocial Intentions | |
|---|---|---|---|---|---|---|---|---|
| | All | Nat. Rep. | All | Nat. Rep. | All | Nat. Rep. | All | Nat. Rep. |
| Negative slopes | 58 | 27 | 52 | 22 | 28 | 8 | 58 | 27 |
| Positive slopes | 9 | 1 | 15 | 6 | 39 | 20 | 9 | 1 |

Label "All" denote models using all 67 countries. Label "Nat. Rep." denote models using only the 28 nationally representative samples.

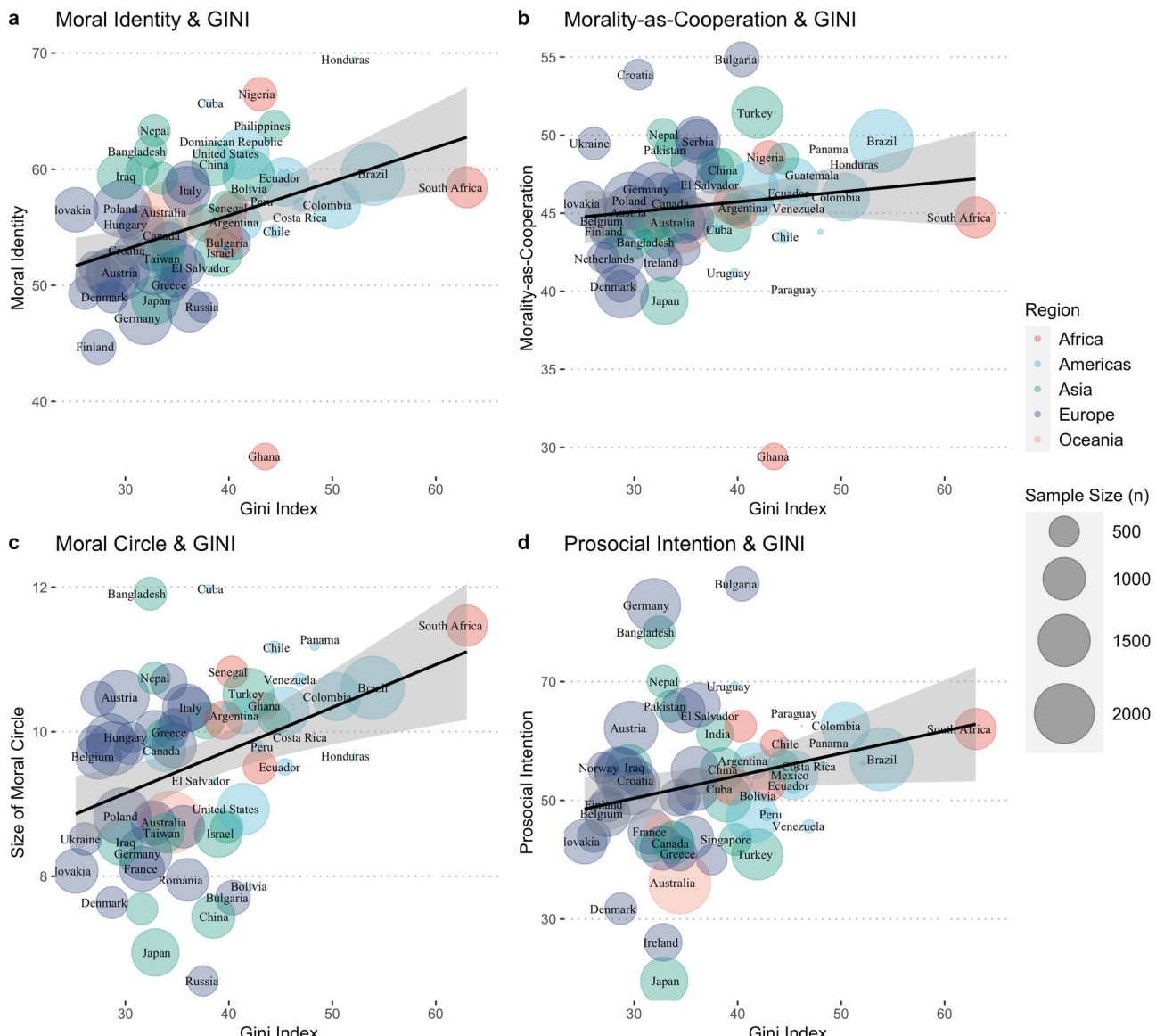

**Fig. 3 | Country and region-level relationships between Moral Identity, Morality-as-Cooperation, size of Moral Circle, Prosocial Intentions, and level of Income Inequality (GINI). a** Association between GINI and Moral Identity. **b** Association between GINI and Morality-as-Cooperation. **c** Association between GINI and the size of one's Moral Circle. **d** Association between GINI and Prosocial Intentions (willingness to donate to a national and international charity). The gray shade area, in all panels, represents the 95% confidence interval.

Still, these results do not allow us to identify whether the overall associations delineated in Table 1 exist because (1) within countries, individuals with higher levels of subjective SES tend to score lower on our four dependent measures; or (2) because countries with higher mean subjective SES contain individuals with lower scores on our four dependent measures; or (3) a combination of these potential explanations. Next, we therefore decomposed the associations between our individual-level measure of relative economic scarcity, subjective SES, and our four main dependent measures into their within-country and between-country components following the method proposed by ref. 84. Here, we found that the within-country component of subjective SES was negatively associated with Moral Identity ($t(45833) = -29.98$, $p < .001$, $\beta = -0.13$, 95% CI [$-0.14$, $-0.12$]), Morality-as-Cooperation ($t(45947) = -14.63$, $p < 0.001$, $\beta = -0.07$, 95% CI [$-0.07$, $-0.06$]), and Prosocial Intentions ($t(45650) = -18.03$, $p < 0.001$, $\beta = -0.08$, 95% CI [$-0.09$, $-0.07$]), while the between-country component had no significant predictive power in any of these relationships

(see Supplementary Tables S6-9). Overall, these findings support the interpretation that within the 67 countries, individuals with lower subjective SES report higher levels of Moral Identity, Morality-as-Cooperation, and Prosocial Intentions, respectively. Yet, for the measure of Moral Circle, we found that the within-country component of subjective SES was associated with a larger Moral Circle ($t(46647) = 3.06$, $p = 0.002$, $\beta = 0.01$, 95% CI [0.00, 0.02]), but that the between-country component was significantly associated with a smaller Moral Circle ($t(46712) = -2.89$, $p = 0.004$, $\beta = -0.07$, 95% CI [$-0.13$, $-0.02$]). Thus, our results suggest a complicated relationship wherein individuals with lower subjective SES report to have a slightly smaller Moral Circle within countries, but primarily that countries with higher mean subjective SES contain individuals who report to have a smaller Moral Circle (see Supplementary Tables S6–S9 *for the full models*). This finding implies that the relationship between subjective economic scarcity and Moral Circle is more sensitive to contextual factors.

## Contextual differences

In further support of the above interpretation, when exploring the relationship between our measures of morality and the individual-level measure of subjective economic scarcity (SES) on the country and region level, the directions of the effects generally remain stable. Still, when running country-level Nested OLS models (another form of multi-level modeling) for all dependent variables, notable differences in effect sizes emerge (see Supplementary Tables S11–S14 and Supplementary Figs. S3–S6). For instance, when comparing country-level associations between subjective SES and Prosocial Intentions, the associations were stronger in countries such as India ($t(615) = -6.56$, $p < .001$, $\beta = -0.26$, 95% CI [−0.33, −0.18]) and South Africa ($t(340) = -2.77$, $p = 0.006$, $\beta = -0.15$, 95% CI [−0.25, −0.04]), while weaker in, for instance, Sweden ($t(1564) = -3.00$, $p = 0.003$, $\beta = -0.08$, 95% CI [−0.13, −0.03]). Also, region-level clustered correlations (Supplementary Table S1) indicated that associations between the same variables were larger in world regions such as the Americas ($t(7402) = -10.80$, $p < .001$, $r = -0.12$, 95% CI = [−0.15, −0.10]), while smaller in Europe ($t(24009) = -12.01$, $p < 0.001$, $r = -0.08$, 95% CI = [−0.09, −0.06]). These findings illustrate notable contextual differences between how and when subjective experiences of economic scarcity at the individual level might be associated with moral decision-making.

## Robustness checks and results summary

Lastly, to add further robustness to our main results, we formulated a set of supplementary models using adjusted disposable net-income as an objectively oriented national-level indicator of experiences of economic scarcity. Here, we show that the associations go in the same direction as the results reported in our main analyses (see Supplementary Table S10). That is, adjusted disposable net-income is negatively associated with Moral Identity, Morality-as-Cooperation, and Prosocial Intentions, but not significantly related to the size of an individual's Moral Circle.

Overall, our results indicate robust associations between individual-level (subjective SES) and country-level (GINI) subjective experiences of economic scarcity and moral judgment. Contrary to existing theoretical paradigms concerning resource scarcity and morality[7,16,18,26,27] as well as our own pre-registered prediction (https://aspredicted.org/727eq.pdf), we find evidence for the notion that subjective experiences of economic scarcity are not associated with a "depletion" of moral character or less prosocial intentions. Instead, it seems that such subjective experiences are associated with a stronger preference for identifying and acting as a moral individual, engaging in cooperative behaviors with a clear moral foundation (e.g., helping kin or reciprocating), and having the intention to engage in prosocial charitable giving.

## Discussion

Research investigating how subjective experiences of economic scarcity affects human moral judgment and decision-making has yielded mixed and at times contradictory results. Based on studies showing that scarcity can shift cognitive functioning[11,13,19,85–87], together with a selection of recent findings highlighting a possible causal link between individual perceptions of relative resource scarcity and unethical or antisocial behavior[20,26,27,88,89], our pre-registered analysis tested the claim that individuals with lower subjective SES and those living in more economically unequal societies would attribute lower importance to acting as moral individuals in terms of identity, cooperation, and prosocial intentions. We found the exact opposite: Conducting a large cross-cultural investigation of these relationships and relying on a dataset including data from 67 countries, our results showed that within countries, individual differences in subjective SES were negatively associated with Moral Identity (Fig. 1a), Morality-as-Cooperation (Fig. 1b) as well as Prosocial Intentions in the form of hypothetical

donation intentions toward national and international charities (Fig. 1d). Furthermore, between countries, individual differences in subjective SES were negatively associated with the size of one's Moral Circle (Fig. 1c). As such, individuals who subjectively experience economic scarcity not only seem more inclined to perceive themselves as moral individuals (i.e., Moral Identity), but also seek to project such morality-related aspects toward their peers and in-group members (i.e., Morality-as-Cooperation, Moral Circle, and Prosocial Intentions). Importantly, we show that this relationship, at least to some extent, holds even at the national level, such that individuals living in countries with high economic inequality (GINI), and thus a greater degree of subjective economic scarcity, report a stronger Moral Identity (Fig. 2a) but also a greater Moral Circle (Fig. 2c). These associations are robust in cross-validations (10-folds, 200 repetitions; Supplementary Table S15).

Yet, how should these findings be interpreted? Previous research (refs. [36,37,90,91]) has argued that individuals who perceive themselves to be of low social class have an increased contextual social (vs. individualistic) orientation. Hence, the link between subjective experiences of economic scarcity and prosocial intentions could reflect the possibility that individuals with lower subjective SES exhibit more prosocial intentions toward others to aid in generating better future life outcomes (e.g., through reciprocation). This interpretation aligns with previous research, which has suggested that individuals with lower SES donate a larger proportion of their income to charity[92]. Similarly, recent data from the World Giving Index suggest that countries characterized by high levels of economic inequality and an objectively large number of individuals living below the poverty line tend to score higher on this index with respect to prosocial behaviors, such as helping a stranger in need, volunteering, and donating to charity organizations[93].

The relationship we find between subjective experiences of economic scarcity and Moral Identity suggests that individuals who perceive such scarcity might aim to act more moral, because they are more attentive to their social environment as their life tends to be influenced by forces which they cannot necessarily control (e.g., relying on government policies, help from charity organizations, and decisions of job managers[36,94]. see also ref. [95]). Acting as a moral individual might not be as important if you perceive the world from a more individualistic perspective, which people with subjective higher SES tend to do[36].

The observed relationship between subjective experiences of economic scarcity and Morality-as-Cooperation indicates that the moral valence of cooperation principles receives higher importance in populations where resources are scarce. At a general level, managing external constraints and depending on others require some, albeit differing, degrees of cooperation in order to gain fruitful outcomes[96]. Consequently, morality is considered a central foundation of cooperative behavior[97,98] and one of the main functions of morality is to promote fruitful cooperation[23,24,96–99]. The concept of Morality-as-Cooperation rests upon the assumption that certain forms of cooperative behavior, such as helping a family or group member, reciprocating, and sharing resources, are considered morally good across cultures[21,97]. Our results indicate that individuals living with subjective experiences of economic scarcity are more inclined to value whether someone helped a member of their family or worked to unite a community when they decide on whether something is right or wrong[21,97]. Thus, our findings on Morality-as-Cooperation suggest that these individuals are particularly prone to consider their external environment when contemplating on moral decisions, likely because they know that they depend on such cooperative connections to obtain more favorable life outcomes.

Regarding the links between subjective experiences of economic scarcity and the size of one's Moral Circle, the magnitude of these associations implies that they should be interpreted with caution.

Between countries, we find suggestive evidence that individuals tend to have a slightly smaller moral circle in countries that are characterized by higher average SES, with the same being true for individuals living in more economically unequal societies. However, for subjective SES, it should be stressed that this association is very small and complicated by substantial heterogeneity across societies. Having noted that, these results are in line with previous findings showing that individuals from lower social classes exhibit greater empathic accuracy[100], which might be reflected in the Moral Circle measure used herein (i.e., exhibiting empathy and care toward a greater number of individuals). Nevertheless, considering the vast heterogeneity in this measure, further studies on this specific association are needed to determine the relationship between subjective economic scarcity (vs. abundance) and the size of one's moral circle.

While the results of the present study originate from a large, cross-cultural research design including 67 countries and appear highly robust, the magnitude of the reported associations is relatively small, and the general explanatory power of our models is modest by conventional standards. However, psychological and cognitive phenomena related to human morality are expected to be influenced by a plethora of different factors[22,101–103], which means that small effect sizes are to be expected as long as these phenomena are not examined in controlled lab conditions, but rather in real-world settings[104–108]. Therefore, while the effect sizes from our analysis are small, this does not imply that they lack practical relevance[109–111]. Effect sizes that are considered small by arbitrary standards can have a large impact when evaluated over time[101,112] or at scale[113–115] (but see ref. 116). This is particularly true for human psychology, where effects can accumulate over time, thus underscoring the fact that while an effect might be small when measured at a single point in time, it can have large ultimate consequences[101]. Also, psychological processes, especially regarding morality, are characterized by "difficult-to-influence" dependent variables, which emphasize that robust small effects can be theoretically important[117]. For instance, our findings demonstrate that an increase of one standard deviation in subjective SES is associated with a decrease of 8% in donation value toward national and international charities, which might seem trivial when considered at the individual level, but can have large consequences for societal outcomes at the population level[60,101,103,118].

In the same vein, it is worthwhile to note that although the current study expands the current state-of-the-art on how subjective experiences of economic scarcity are associated with moral decision-making by studying four well-validated measures linked to morality, other measures might have been relevant to include as well. Therefore, it should be stressed that the current investigation does not aim to conclude whether low (vs. high) subjective SES makes people more or less moral in general. Instead, the present study outlines that certain types of morality seem to be more pronounced under subjective experiences of economic scarcity, speculatively because such morality could aid in producing more fruitful prospective outcomes for individuals subjectively experiencing to be living with less resources. Thus, we do not support a consequentialist view but rather recognize that other types of morality measures might be more (or less) pronounced in individuals with more abundant resources, depending on the precise context and the specific type of morality measures used.

Relatedly, our individual-level and macro-level measures of subjective experiences of scarcity map how subjective, and not objective, experiences of economic scarcity are associated with our four indicators of moral judgment. Therefore, the current work only outlines how perceptions of "having less" are associated with moral judgment, which naturally comes with both limitations and strengths: Limitations in the form of lacking individual-level objective measures of economic scarcity (e.g., household income), but strengths in the form of focusing directly on the subjective experience of economic scarcity, which can hold irrespective of the economic development of a given society. We

urge scholars to build on these findings and further investigate how subjective experiences, as well as objective indicators of resource scarcity, might be intertwined and are potentially differentially associated with moral judgment and decision-making.

Regarding the macro-level measure of income inequality used as a proxy for perceptions of economic scarcity (i.e., the GINI index), it should be noted that this measure constitutes a crude indirect measure of country-level perceptions in economic scarcity. That is, while prior work has shown that higher economic inequality increases competition[72], social comparison[75], and a sense of relative deprivation[71]—factors that are all strongly associated with subjective perceptions of economic scarcity[7,119]—it is crucial to note that using the GINI index to capture such perceptions has its limitations. For example, the GINI index cannot directly measure subjective perceptions of economic inequality and is only a single-parameter measure of the distribution of financial resources in a given economy, meaning that it cannot necessarily highlight at what part of the income distribution said inequality is concentrated[120]. Therefore, future work should also assess how individuals in different contexts and across societies directly experience economic scarcity as a result of economic inequality[74]. At the macro-level, using the two-parameter *Ortega*-model[120] to distinguish between inequality concentrated at the bottom- and top-income percentiles, respectively, could provide a more detailed perspective on this issue and may spur more fine-grained investigations on how psychological differences in the experience of economic inequality might affect judgment and decision-making within but also beyond the morality domain.

Moreover, it is important to acknowledge that the results reported herein rely on self-reported responses. This is a central limitation of the current investigation, as it is debatable whether these measures are capable of capturing responses on metrics such as real, observable behavior. However, previous research has found that self-reported donation intentions are highly correlated with real donations[54] and that self-reported unethical behavior correlates with real-life lying[121], suggesting that self-report responses are at least somewhat predictive of unethical behavior. For example, our self-reported measure of subjective SES could possibly be influenced by personality differences. However, previous work has indicated strong support for the construct validity of the MacArthur scale[65] and the scale has been used extensively to study how subjective indicators of SES are predictive of outcomes related to subjective well-being[68] as well as physical and mental health[122]. Therefore, considering the robust associations documented herein and the 67 societies involved, our findings contribute to the literature on human morality but should be complemented with future field-based investigations to counter concerns linked to external and ecological validity[105,123–125]. Still, we welcome future research to examine our obtained associations in more realistic environments, preferably using behavioral measures, experimental approaches, and a larger range of control variables (e.g., personality dimensions such as conscientiousness) to allow for causal inferences.

A final note of caution pertains to the fact that the data used in the present investigation were collected during the COVID-19 pandemic. While the general idea regarding the pandemic seems to be that "COVID-19 does not discriminate," recent studies have shown that vulnerable individuals, such as those living with less economic resources, have higher mortality rates than their less vulnerable counterparts[126,127]. The results of the current study not only show a general link between subjective experiences of economic scarcity and human moral judgment, but also suggest that this association is present when people who perceive themselves to have the least resources experience an extraordinary increase in the level of risk and exposure to threat. As research has argued that hostile environments motivate people with less available resources to engage in prosocial behavior[37,128], our findings may therefore be stronger than similar investigations conducted during pre- or post-pandemic times or data

collected in the absence of other public crises (e.g., financial recessions, droughts, terrorism attacks, and wars)[129].

In conclusion, the present research demonstrates that subjective SES and income inequality are associated with multiple dimensions of human morality. These findings underline the complex relationships between social class perceptions and inequalities, and the way individuals morally think, respond, and act. We urge future research to disentangle how moral character and behavior might be associated with not only subjective but also objective experiences of economic scarcity.

## Methods

The study was pre-registered on AsPredicted before the data was accessed (https://aspredicted.org/727eq.pdf, August 3rd, 2020). While we generally adhered to the pre-registered analysis plan, some deviations still exist. Specifically, for our main analysis, we employed multilevel correlation analysis, nested OLS regressions, multi-level modeling (Linear Mixed Effects models), and cross-validations instead of standard Pearson's correlations and non-nested OLS regressions, thus addressing the same questions as pre-registered but with more sophisticated and robust methods. We have not reported the originally planned analysis in the *Supplementary Information* as this analysis plan is flawed because it does not account for the inherent clustering in the data in terms of countries. All statistical tests reported are two-tailed and the alpha-level is set at 0.05. Furthermore, our pre-registration noted that we would include household income on the individual level in all of our models. However, this was not possible given that the dataset from the ICSMP project did not include such data[62,63]. That is, because we only had access to the Danish sample in the pre-registration stage, we assumed that the data from all 67 countries would include this household income, which was not the case. Nevertheless, to add further robustness to our results, we augmented our data with a measure of Adjusted Disposable Net-Income from the World Bank (see Supplementary Table S10).

The data were obtained from the International Collaboration on Social & Moral Psychology of COVID-19 (ICSMP)[62,63]. This project was a large-scale international collaboration between more than 200 researchers from 67 different countries with a goal to create an online survey to measure psychological factors underlying the attitudes and behavioral intentions related to COVID-19. The project received ethical approval from the institutional review board at the University of Kent (ID 202015872211976468) and informed consent was obtained from all participants prior to their voluntary participation in the study. No additional ethics approval was needed for the research reported in this paper.

The dataset contains self-reported demographics and social and moral psychology data from 51,089 individuals from 67 countries and 5 different regions of the world. Each national team responsible of collecting data in their country translated the English survey into their nations' language using the standard forward-backward translation method. Members of every participating country were asked to collect data from at least 500 participants, nationally representative with respect to gender and age. No statistical method was used to pre-determine the sample size, considering that the estimated final sample of several thousand participants would have sufficient statistical power to detect very small effect sizes by conventional standards. The data were collected from online platforms or panel agencies during April-May 2020 and were administered using an online survey. Every participating individual answered questions regarding demographics and self-reported public health behaviors as well as a series of psychological measures. Scale order was randomized for every participant. The dataset was cleaned by the lead methodologists from the ICSMP project for the initial publications using the dataset[62,63]. A total of 53,269 participants answered the survey. Of these, 2049 participants were excluded for not having completed the full survey, and 131 participants

were excluded for being younger than 18 y/o or older than 100 y/o. Furthermore, we removed 526 participants who failed attention checks and 167 individuals reported "Other" as their gender identification. For gender identification, we excluded these participants from our formal analysis to reduce the risk that this category of the covariate would inflate the results obtained from our models, given that the "Other" category was too infrequently represented in the data for meaningful country comparisons. That is, to maintain a balanced representation of the gender covariate in the dataset, the small number of participants identifying as "Other" were excluded, ensuring robust estimation and interpretation of regression coefficients related to gender. This resulted in a final sample of 50,396 participants. Of the 67 countries, 28 countries used fully representative samples with respect to gender and age. Nationally representative samples were collected using stratified sampling, while non-representative samples were collected using convenience sampling. A total of 44 countries included more than 500 participants. Mean age was 43 years and 52% of participants reported their gender as female.

In addition to data from the ICSMP, we obtained the most recent GINI Indexes from the World Bank[64] for every country included in the study, with some rare exceptions. Taiwan GINI data were obtained from Statista[130], Cuban GINI data were obtained from Reuters[131] and New Zealand and Singapore GINI data were obtained from Knoema[132,133]. Region names were obtained from the World Bank Development Indicators[134].

### Variables

Moral Identity was measured using a scale of 10-items[60] such as "It would make me feel good to be a person who has these characteristics", which would be answered based on a description of a person who has the characteristics: "caring, compassionate, fair, friendly, generous, helpful, hardworking, honest, kind". Each item was measured using a 10-point slider with three labels: 0 = "Strongly disagree", 5 = "Neither agree nor disagree", 10 = "Strongly agree". Items 3 and 4 were reverse scored. Results were aggregated into a single-scale (Cronbach's $\alpha = 0.729$), instead of two subscales (internalization and symbolization), as in the original publication developing the scale[60]. Hence, our aggregated measure of moral identity indicates how important moral identity is to one's self-definition (*internalization*) and to what degree an individual expresses this moral identity (*symbolization*) but does not distinguish between these two aspects. To investigate and validate the equivalence of the factor structure of this scale across societies, we carried out Multiple-Group Factor Analysis Alignment as proposed by ref. 135. In this analysis, results showed that for factor loadings the scale exhibited 8.5% of non-invariance for item parameters and 7.8% for intercepts, which indicates that the majority of non-invariance is absorbed by our country-varying factor means and variances. Hence, following ref. 135 suggestions of a cut-off value of 25% in order to consider a scale non-invariant, we deemed the scale suitable for use in the current investigation, consistent with recent projects who have used the data and the specific scales[136]. The full results of the Multiple-Group Factor Analysis Alignment for Moral Identity can be found in Supplementary Tables S16-S21.

Morality-as-Cooperation was measured using a 7-item scale adapted from ref. 25. Each item represented one question out of three from each of the seven "relevance items" from the Morality-as-Cooperation questionnaire[25]. The questions chosen from the original scale were the ones with the highest predictive validity[62]. Individuals were initially asked the following: "When you decide whether something is right or wrong, to what extent are following considerations relevant to your thinking?". Here, the "family" item was labeled "Whether or not someone helped a member of their family". The "group" item was labeled "Whether or not someone worked to unite a community". The "reciprocity" item was labeled "Whether or not someone showed courage in the face of adversity". The "deference"

item was labeled "Whether or not someone deferred to those in authority". The "fairness" item was labeled "Whether or not someone kept the best part for themselves". The "property" item was labeled "Whether or not someone kept something that didn't belong to them." Each item was measured using a 10-point slider with three labels: 0 = "Strongly disagree", 5 = "Neither agree nor disagree", 10 = "Strongly agree". All 7 items were aggregated into our single measure of Morality-as-Cooperation (Cronbach's α = 0.732). In the original publication developing the scale, test-retest correlations for the full scale was shown to range from 0.79 to 0.89[25]. Again, to investigate and validate the equivalence of the factor structure of the scale across societies, we carried out Multiple-Group Factor Analysis Alignment. Here, the results showed that for factor loadings the scale exhibited 10.4% of non-invariance for item parameters and 14.1% for intercepts, which, as for Moral Identity, indicated that the majority of non-invariance was absorbed by our country-varying factor means and variances. Based on the same argumentation as for the Moral Identity scale, we therefore deemed the factor structure of this scale sufficient for use in the analyses reported in this paper, in line with previous investigations that have used this scale in a multi-national context across 60 countries[21]. The full results of the Multiple-Group Factor Analysis Alignment for Morality-as-Cooperation can be found in Supplementary Tables S22–S27.

Moral Circle was measured using a single-item scale with 16 levels[137], asking participants to indicate the extent of their moral circle, where moral circle means "the circle of people or other entities for which you are concerned about right and wrong done toward them." The scale ranges from 1 = "all of your immediate family" to 16 = "all things in existence". Test-retest reliability of the scale has previously been shown to be .61[138] and the scale has been validated and used in numerous previous investigations across different disciplines (see refs. 61,137,139–141).

Prosocial intentions were measured using a hypothetical choice task with three items. In this task, individuals were asked how much (in percent), if given a daily median income, they would be willing to 1) keep to themselves, 2) donate to a *national* charity and 3) donate to an international charity. We formed our measure of prosocial intentions by aggregating the second and third item across individuals.

Subjective socioeconomic status (SES) was measured using the single-item MacArthur ladder scale[138]. This scale uses a picture of an 11-step ladder and asks participants where, in their country, they would stand if the top indicated the people who are the best off—those who have the most money, the most education, and the most respected jobs, while the bottom are the people who are the worst off—those who have the least money, least education, and the least respected jobs or no jobs. Participants indicated their standing in their respective society from 0 (absolute bottom) to 10 (absolute top). The scale has been used extensively in previous research to capture subjective social class across disciplines (see ref. 122 for a review and meta-analysis).

GINI Index, as measured by the World Bank, is based on primary household survey data obtained from statistical agencies and World Bank country departments[64]. It measures the amount of income inequality, where 0 = total equality and 100 = total inequality. For more information on specific measurement and methodology, see PovcalNet from the World Bank (iresearch.worldbank.org/PovcalNet/index.htm).

## Correlations

Due to the nested structure of our data, the correlations between the variables; Subjective Socioeconomic Status (SES)[138], Moral Identity[60], Morality-as-Cooperation[25], Moral Circle[137], and Prosocial Intentions were calculated using multilevel Pearson's correlations with country as the random intercept. We also calculated grouped correlation coefficients for every country and region in the dataset to identify country-level and region-level differences of interest (Supplementary Dataset 1 *and* Supplementary Table S1). Correlations between our dependent variables and the independent variable "GINI Index" were calculated as single-level Pearson correlations without the multilevel nesting, as the GINI Index would not be different per individual measure, as it constitutes a country-level measure. Lastly, for exploratory purposes, we also calculated a simple correlation between the dependent measures of Moral Identity, Morality-as-Cooperation, Moral Circle, and Prosocial Intentions (split into its national and international components, see section *Measures*), which can be found in Supplementary Tables S2–S3.

## Multilevel models

To probe the internal validity and contextual sensitivity of our results, we rely on advanced statistical methods in the form of multi-level modeling and cross-validations using supervised machine learning algorithms. These methodological approaches allow us to identify robust individual and country-level associations between subjective experiences of economic scarcity and morality. In doing so, we contribute with a rigorous cross-cultural and generalizable extension of previous research[20,37,38,46,49,90,142] on how our studied facets of economic scarcity might affect moral judgment and decision-making. To test the robustness of our results, we also include adjusted disposable net-income in each included country as a more objectively oriented national-level indicator of experiences of economic scarcity in a set of supplementary models (*see* Supplementary Table S10).

We performed three specific forms of multi-level modeling. Firstly, we performed linear mixed effects modeling[143], where we regressed our dependent variables with our two main independent variables and covariates, while using country as the random intercept. Two-tailed significance testing (α = 0.05) was applied for all analyses. For ease of reporting and interpretation, we standardized parameters to report β-coefficients and 95% confidence-intervals of our analysis. 95% Confidence Intervals (CIs) and p-values were computed using the Wald approximation. These models use the Restricted Maximum Likelihood (REML) algorithm. In this setup, our models use pairwise deletion of missing values before the maximum likelihood estimation. To add robustness to our results, we ran a set of identical models, where we imputed missing data based on a Random Forest estimation. Our results appear robust to this change in the data structure (see Supplementary Table S4).

Secondly, we performed linear mixed effects modeling[143], this time focusing on the independent variable of subjective socioeconomic status (SES), where we decomposed our associations into within-country and between-country effects[84]. To do this, we formed three new variables from our original measure of subjective socioeconomic status; 1) a grand-mean centered measure of SES, that subtracts the grand mean from each individual observation, 2) a within-country centered measure of SES which captures variations relative to each country's average by subtracting the raw observation from the country-specific mean, 3) a between-country centered measure of SES which reflects between-country differences in SES, obtained by subtracting within-country centered values from the grand mean centered values. Using these new variables, we formulated a selection of models where we regressed our dependent variables on morality with our new within-country and between-country measures of socioeconomic status, GINI Index, covariates, and country as the random intercept. Again, to add robustness to our results, we ran a set of identical models, where we imputed missing data based on a Random Forest estimation.

Thirdly, we performed Nested Ordinary Least Squares (OLS) regressions on all dependent variables, with SES as the independent variable, as only this variable would differ at the individual level, given that GINI is a country-level measure. Country was used as nesting, such that we simultaneously ran 67 OLS regressions for each of our dependent variables. This approach allowed us to identity the individual country-level coefficients for each of our models. Full results of these models are reported in Supplementary Tables S11, S12, S13, S14, and visualizations are reported in Supplementary Figs. S3, S4, S5, S6.

## Cross validations

As a robustness check to assess the predictive power of our models, we applied 10-fold cross validation, with 200 repetitions on all multi-level models. Cross-validation is a form of supervised machine learning which splits the dataset into *K* number of independent datasets (in our case 10) and then uses every dataset in turn as the validation set, where the other *K-1* datasets then act as the calibration sets. Each fold of the data leads to a different estimate of the *Root Mean Square Error* (RMSE) of the model and therefore the process is repeated multiple times (in our case, 200) to get reliable estimates. While cross-validation can be used as a procedure in model selection, in this article we used the procedure to validate the robustness of our models[144]. That is, our cross validations provide confidence in the reported findings, by illustrating that the included model results are in fact the ones with the lowest *RMSE*. The full results of all cross validations can be found in Supplementary Table S15.

## Reporting summary

Further information on research design is available in the Nature Portfolio Reporting Summary linked to this article.

## Data availability

The raw and preprocessed ICSMP data are publicly available at OSF (raw: https://doi.org/10.17605/osf.io/tfsza, preprocessed: https://osf.io/y7ckt/)[62,63]. The processed GINI and Adjusted Net-Income data are publicly available at OSF (https://doi.org/10.17605/OSF.IO/dxvmk)[145] and was obtained from the World Bank[64].

## Code availability

The analysis code was written in the statistical environment *R* (version 4.0.3) and the script is openly available at OSF (https://doi.org/10.17605/OSF.IO/dxvmk)[145]. This analysis code can directly reproduce all figures and tables reported in the paper, the Supplementary Data and the Supplementary Information.

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

## Acknowledgements

The authors thank the International Collaboration on the Social & Moral Psychology of COVID-19 (ICSMP) for allowing us to use the data before the initial study using the dataset was published. Also, we thank Prof. Paulo Sérgio Boggio for help with the script for Fig. 1. No funding for any of the authors was provided for the research reported in this article.

## Author contributions

C.T.E., P.M., L.A., and T.O. designed the study outline. C.T.E. analyzed the data. C.T.E., P.M., L.A., and T.O. interpreted the results. C.T.E., P.M., L.A., and T.O. wrote and revised the paper.

## Competing interests

The authors declare no competing interests.
