## [Peer Review File · Nature Communications]

Reviewers' Comments:

Reviewer #1:

Remarks to the Author:

The paper reports a really interesting cross-national dataset on the relationships between SES and morality/cooperation. They find mainly negative relationships (higher SES, lower morality), addressing a literature that has previously been very mixed in terms of direction of associations. The dataset is well worth publishing and the findings are useful, but some more work is needed before the paper is publishable or interpretable. Below I list major then minor issues.

1. Analysis. For SES, the authors need to decompose the associations into their within-country and between-country components., using the method of Van der Pol and Wright (*Animal Behaviour* 77(3):753-758). Given that countries differ in mean SES, the overall findings could be explained by (a) within countries, individuals with higher SES have lower morality; or (b) countries with higher mean SES contain individuals with lower morality; or some combination of the two. The within- and between-country associations can even be in opposite directions (looking at figure 2c and table 1 that looks like it could be true here for moral circle). Anyway, neither the variance partition analysis reported at 144 nor including a random effect of country achieves this, but it is essential to interpret what is going on.

2. Visualization. Figure 2 is not the right visualization for the SES analyses, exactly because it plots a between-country association for results that appear to be driven by the within-country associations. Figure 2c even shows a trend in the opposite direction to the main finding. Instead, I suggest plots with 68 regression lines: 67 faint ones, one per country, and then the overall model to show the central tendency. There would be too many lines to colour code by country, but the reader would at least be able to see whether most of them had similar slopes or not. In fact, it would be great because the reader would be able to see BOTH the main pattern (most lines slope down) and the heterogeneity (a few go the opposite way). It is really easy to make these plots in R (`ggplot2::geom_smooth(method="lm", aes(group=factor(country)))`), or whatever. BTW, another useful thing I thought of was a table saying, for each DV, how many countries showed a significant positive, significant negative, or non-significant association for SES. This information is in the supplement but could be summarised in the main paper (around line 160 I thought of this).

3. Framing. The authors have an unfortunate tendency to claim that poverty 'distorts' or 'impedes' cognition (e.g. 44, 188). This is contentious and potentially stigmatizing. It is more balanced to say that poverty 'shifts' cognition. People in poverty may be better at some kinds of cognition. Moreover, discounting the future and other 'impairments' could be perfectly rational in an uncertain environment. So I would urge a more neutral framing here (see <https://doi.org/10.1177/0963721419881154> or <https://doi.org/10.1017/S0140525X1600234X>). It would actually fit better with the authors' argument, which is that people in poverty shift to greater moral interdependence as a strategy to cope with their environments.

4. Interpretation. I have never been a big fan of the 'are poorer people more or less prosocial' debate. This is because the answer seems to me obvious: both are somewhat prosocial, but in different ways. Therefore, the direction of result you get is going to be totally dependent on the measure of prosociality (or morality) that you choose (who is the prosocial interaction with, what is the resource, etc.). I think the authors should bear this in mind more in their discussion. Both sides of the debate are probably right, given that poor people do more of some kinds of morality and less of others. This study, big as it is, does not settle the matter because it has only limited measures of morality/prosociality. Other measures might have produced opposite associations.

5. External validity. At a couple of places, the authors confuse external validity with generalizability (e.g. 88, 257). These are not the same thing at all. I would say the present study has high generalizability, because of the large sample across many countries, but very low external validity. External validity is to do with the mapping of your measures onto real-world behaviour. These rather abstract short self-report measures probably have very low external validity in this sense (if you interpret them as proxies for moral or prosocial behaviour). A behavioural economist would dismiss these findings as interesting because all the measures are hypothetical 'cheap talk'. They probably say nothing about how those people would allocate actual time or resources. This

needs to be stated as a limitation. Relatedly, the authors should not claim to have measured 'prosocial behaviour' (e.g 32, 196). No behaviour was measured in this study. 'Hypothetical prosocial intentions' is quite another thing.

Minor points

Line 67. 'their immediate environment' is unclear; presumably this means 'their social relationships'

Sentences beginning at line 72 are poorly written and ungrammatical

109. What the four DVs are needs to be explained, either in the Introduction or when they are introduced in the results. At present, results come without any prior explanation of what the measures are. This can often be a problem in formats where the methods section comes at the end. A brief sentence or two introducing the measures (both the DVs and the predictors) is therefore needed in the introduction or in the results text.

116. If the alternative analyses use imputation, how do the main models treat missing data?

Table 1: For Gender, give the ref category to make the direction of effect interpretable

Reviewer #2:

Remarks to the Author:

This paper reports the results of a survey conducted across many countries that measures subjective SES and then correlates this with measures of moral identity, morality as cooperation, moral circle, and prosocial behavior. The authors use these data to test pre-registered predictions that higher subjective SES would be associated with higher moral character and behavior. However, they found the opposite to those predictions. They found people reporting lower subjective SES reported higher moral character and behaviors. I have some concerns about the sampling strategy and measures used in the study that may limit the study's ability to test hypotheses about the relation between "chronic resource scarcity" and "Moral Character and Behavior" across countries.

I am concerned about the sampling strategy used in this study, and how it could affect the author's ability to test the hypotheses across the countries. It sounds like the people in the different countries collecting data could collect data in any way that they wanted, but were requested to try and get a representative sample for age and gender (which many countries were unable to do). I would think it would be important to get representativeness for household income to provide a good test of this hypothesis across societies. I am assuming that many of the people collecting data were other social scientists and that if they were to recruit participants from their personal network (or University), then this would bias the samples to have a higher SES. This would be a serious limitation of the study, especially for being able to test for any differences between the countries. The differences between the countries could be due to the different sampling strategies used in each country. Also, this sample strategy could result in a restriction of range of SES, and not sample many people who have "chronic resource scarcity".

I also have some concerns about the measures. Specifically, I am concerned that there is not equivalence of the factor structure of the scales across all the societies. This would affect how we can interpret the correlations across societies. I would suggest the authors report a structural equivalence test across the societies for moral identity and morality as cooperation. Furthermore, the authors only include a single item measuring "moral circle". They also do not report evidence for the validity of this item. In fact, it sounds like this is the first time this item has been used in prior research. This severely limits the value of this item to be used to test theory. There is some data in the present study that could be used as a validity test: Does the measure of moral circle correlate with a relatively higher amount donated to an international, versus national, charity? The authors operationalize "chronic resource scarcity" with a measure of subjective SES. I would have preferred a more objective measure of chronic resource scarcity, such as household income. The paper could benefit from at least reporting the relation between subjective SES and other

measures of resource scarcity in prior work. Currently the paper just says this is a common way of measuring SES, while there are also several other methods used in the literature.

Another concern I have is that with such a large dataset, everything is statistically significant. For example, the Beta value for SES and moral circle is tiny. Is it even worth discussing this result as supporting a theoretical framework? How many participants would be required to detect the effect size in a subsequent study? I think it would be useful for a paper like this to identify an effect size that they would suggest is reasonable to interpret. I agree with the authors that small effect sizes can still be valuable, but there must be a point when we think its just not important, or it is so trivial that it is not really supporting a prediction from a theoretical framework.

In figure 2b, it looks like there is no relation between SES and morality as cooperation, but then there is a negative relation between SES and moral circle. However, in Table 1 the beta value is larger for morality as cooperation, compared to a moral circle. Why is there a discrepancy between Table 1 and Figure 2?

When I was reading the paper, I did not know the definition of many concepts when they were first raised in the paper, such as moral identity, moral circle, and morality-as-cooperation. I also don't understand the definition of morality that is used in this work. Of course, prosocial behavior (i.e., behaviors that benefit others) can also be immoral, such as helping a prisoner escape from prison. And people can cooperate together to do some very horrible, immoral behaviors. In general, I thought the paper could be improved substantially in terms of communicating about the concepts discussed in the paper. For example, a major conclusion of the paper is that: "chronic resource scarcity...results in more moral outcomes", and I was left confused about what "moral outcomes" even mean in this context.

Reviewer #3:

Remarks to the Author:

In this paper the authors report on the relationship between measures inequality (Subjective personal SES, National GINI coefficient) and various measures of "morality" (moral identity, morality-as-cooperation, size of moral circle, and [hypothetical] prosocial behaviour).

This paper aims to address a very interesting question that has been hotly debated: Does higher economic status affect one's moral values and behavior? As the authors explain in a clearly written introduction, the literature on this is mixed (although there are factors that may explain the apparent contradictions in the literature). The present study aims to address this question using a massive data set (over 46,000 participants from 67 nations). They report that economic deprivation is, in various ways, associated with more "moral" outcomes.

This study has some notable strengths, such as the aforementioned large international sample as well as some sophisticated statistical features. However, it seems to me that it is flawed in some very basic ways. It's not clear to me whether these problems are correctable. It seems that they should be, but at the same time, the fact that the authors didn't address these problems already makes me worry that the fixing the problems leads to a very different set of conclusions.

The main problem is that authors use measures of SES/inequality without taking into account absolute wealth—both at the individual and national level. The SES measure asks for a subjective judgment of where the respondent falls on an SES ladder. As one might expect, people's responses are heavily influenced by the local context. As an illustration, the mean SES rating on this measure is *lower* in wealthy Denmark than in poor Nigeria. It doesn't follow from this that this measure is meaningless. It tells us that the average Dane sees themselves as lower on the *local* SES ladder than the average Nigerian does. But this is not a meaningful measure of actual SES, deprivation, or anything of that sort.

The same holds for The GINI coefficient, the other major predictor variable. Germany and Bangladesh have similar GINI coefficients, but obviously the experience of inequality and deprivation are very different for the average German as compared to the average Bangladeshi.

The way to deal with this is obvious: One must (at the very least) account for wealth at the individual and national levels in when using measures of perceived relative SES and (objective)

national-level inequality. This is so obvious, I was surprised that the authors didn't do this. And I was especially surprised given that the analysis plan was pre-registered. So, I looked at the pre-registration and, sure enough, the authors specify household income as a key predictor variable. But this variable is not used at all in the paper.

I also have concerns about the key outcome measures related to morality. In the introduction it sounds as if the authors are weighing in on the debate about whether higher SES makes people more or less "moral", meaning more or less inclined to do things that are widely regarded as morally good (e.g. being generous) or morally bad (e.g. cheating). But only the generosity measure falls into this category. And here the authors rely on self-report of generosity, which could vary enormously by culture and/or be generally unreliable. At the very least, one would want to corroborate these subjective self-reports of generosity with country-level data on things like actual charitable giving. And for the other measures, it's not clear whether they exhibit the same kind of local/cultural relativity that the SES ladder measures do. For example, what counts as having a large vs. small moral circle in Nigeria vs. the US?

Finally, it's not clear whether the preregistration is really a preregistration. The authors explain that "some" of the data have already been collected and then refer to this explanation: "The data has already been collected and is a part of an international project on moral psychology during the corona (COVID-19) pandemic. The data will initially be used in the paper for which it was collected."

This doesn't tell us which data were collected prior to the pre-registration and which were not. And, most importantly, it doesn't say whether the authors were aware of the outcomes of these analyses prior to the pre-registration. Given that the analyses reported on in the paper differ from the analyses outlined in the pre-registration, it seems that they didn't know the outcomes in advance, which is good—except that we only know this because the authors deviated so significantly from their pre-registered plan.

This is a rich data set, and it seems possible to me that the authors could analyze it in a way that makes more sense. However, the fact that the authors pre-registered a more sensible approach and then abandoned it suggests that this would lead to rather different conclusions. Likewise, it's possible that the data could be augmented with nation-level objective measures that would allow them to more directly address the questions they seem to want to answer in the introduction. But, in its present form, this paper has some very significant limitations.

Revision Notes

Reviewer 1:

1. *The paper reports a really interesting cross-national dataset on the relationships between SES and morality/cooperation. They find mainly negative relationships (higher SES, lower morality), addressing a literature that has previously been very mixed in terms of direction of associations. The dataset is well worth publishing and the findings are useful, but some more work is needed before the paper is publishable or interpretable. Below I list major then minor issues.*

Thank you very much for your constructive review of our paper. We are pleased to hear that you find our research and the data to be interesting, useful, and worth publishing. Below you will find our answers to your comments, with direct references to the revised manuscript.

2. *1. Analysis. For SES, the authors need to decompose the associations into their within-country and between-country components., using the method of Van der Pol and Wright (Animal Behaviour 77(3):753-758). Given that countries differ in mean SES, the overall findings could be explained by (a) within countries, individuals with higher SES have lower morality; or (b) countries with higher mean SES contain individuals with lower morality; or some combination of the two. The within- and between-country associations can even be in opposite directions (looking at figure 2c and table 1 that looks like it could be true here for moral circle). Anyway, nether the variance partition analysis reported at 144 nor including a random effect of country achieves this, but it is essential to interpret what is going on.*

Thank you for this detailed and valuable suggestion. We have now implemented the suggested analyses to complement our original analytic approach. We find that our effects are primarily explained by within-country differences, such that within countries, individuals with higher SES seem to have lower Moral Identity, lower Morality-as-Cooperation, and less Prosocial Intentions. Yet, for the measure of Moral Circle, we find that the within-country component of subjective SES was very weakly associated with a larger Moral Circle and that the relationship between subjective SES and the size of one's Moral Circle is primarily explained by between-country differences (pp. 15-16, line 21-5; and Supplementary Tables S7, S8, S9 and S10). We have revised the analyses and the interpretation of them to reflect these new findings.

3. *2. Visualization. Figure 2 is not the right visualization for the SES analyses, exactly because it plots a between-country association for results that appear to be driven by the within-country associations. Figure 2c even shows a trend in the opposite direction to the main finding. Instead, I suggest plots with 68 regression lines: 67 faint ones, one per country, and then the overall model to show the central tendency. There would be too many lines to colour code by country, but the reader would at least be able to see whether most of them had similar slopes or not. In fact, it would be great because the reader would be able to see BOTH the main pattern (most lines slope down) and the heterogeneity (a few go the opposite way). It is really easy to make these plots in R (ggplot2::geom_smooth(method="lm", aes(group=factor(country))), or whatever). BTW, another useful thing I thought of was a table saying, for each DV, how many countries showed a significant positive, significant negative, or non-significant*

association for SES. This information is in the supplement but could be summarised in the main paper (around line 160 I thought of this).

We thank you for these helpful suggestions. As requested, we have implemented these plots, which give a much clearer illustration of the central tendency in the relationship between subjective SES and each of our measures of morality as well as the variation around the central tendencies (see Figure 2, p. 12). Furthermore, we have also included a table that summarizes the number of countries showing a significant positive, significant negative, or non-significant association for SES (Table 2, p. 13). An even more fine-grained overview is available in the Supplementary Materials.

4. *3. Framing. The authors have an unfortunate tendency to claim that poverty 'distorts' or 'impedes' cognition (e.g. 44, 188). This is contentious and potentially stigmatizing. It is more balanced to say that poverty 'shifts' cognition. People in poverty may be better at some kinds of cognition. Moreover, discounting the future and other 'impairments' could be perfectly rational in an uncertain environment. So I would urge a more neutral framing here (see <https://doi.org/10.1177/0963721419881154> or <https://doi.org/10.1017/S0140525X1600234X>). It would actually fit better with the authors' argument, which is that people in poverty shift to greater moral interdependence as a strategy to cope with their environments.*

Thank you for this important comment. The use of the more negatively-laden framing was adopted from seminal research in the field (see e.g., Mani et. al, 2013 in *Science: Poverty impedes cognitive functioning*). Yet, based on your comment, we fully agree that this framing can be potentially stigmatizing to individuals with lower levels of resources, which goes directly against one of the main points of our research. Consequently, we have changed the framing throughout the manuscript to be more neutral in language. For example, we now describe that chronic resource scarcity can shift cognitive focus and attention (e.g., p. 2, line 5, p 3., lines 3-6). We believe that this new framing is more balanced and we hope that you agree.

5. *4. Interpretation. I have never been a big fan of the 'are poorer people more or less prosocial' debate. This is because the answer seems to me obvious: both are somewhat prosocial, but in different ways. Therefore, the direction of result you get is going to be totally dependent on the measure of prosociality (or morality) that you choose (who is the prosocial interaction with, what is the resource, etc.). I think the authors should bear this in mind more in their discussion. Both sides of the debate are probably right, given that poor people do more of some kinds of morality and less of others. This study, big as it is, does not settle the matter because it has only limited measures of morality/prosociality. Other measures might have produced opposite associations.*

Thank you for raising this point. We agree that our study only measures one specific form of prosociality in the form of donation intentions towards charity organizations. Yet, we argue that by including our additional measures of moral character and cooperation, our study offers a nuanced view on whether subjective economic scarcity is related to morality. In the manuscript, we have now added a clear conceptualization of moral decision-making as a multidimensional term (p. 3, line 11-17) as well as we have strengthened our argument for the value of including four different indicators of morality (p. 5, line 12-20, pp. 6-7, line 11-2). Moreover, we have expanded our discussion by emphasizing that although the current study expands the current state-of-

the-art on how experiences of relative economic scarcity influence moral decision-making through the four well-validated morality measures that we selected, other measures might have been relevant to include as well (p. 21, line 12-22). We acknowledge that our investigation does not aim to conclude whether low (vs. high) SES makes people more or less moral *in general*. Thus, other types of morality measures might well be more (or less) pronounced in individuals with more abundant resources, depending on the precise context and the specific type of morality measures used. Finally, we also discuss what other types of prosocial behavior would be interesting to explore and how such behavior might differ depending on the demographic profile of participants.

6. *5. External validity. At a couple of places, the authors confuse external validity with generalizability (e.g. 88, 257). These are not the same thing at all. I would say the present study has high generalizability, because of the large sample across many countries, but very low external validity. External validity is to do with the mapping of your measures onto real-world behaviour. These rather abstract short self-report measures probably have very low external validity in this sense (if you interpret them as proxies for moral or prosocial behaviour). A behavioural economist would dismiss these findings as interesting because all the measures are hypothetical 'cheap talk'. They probably say nothing about how those people would allocate actual time or resources. This needs to be stated as a limitation. Relatedly, the authors should not claim to have measured 'prosocial behaviour' (e.g. 32, 196). No behaviour was measured in this study. 'Hypothetical prosocial intentions' is quite another thing.*

Thank you for this valuable comment. We have rewritten the parts about external validity and added supporting references to clarify that we are referring to cross-cultural generalizability (pp. 5-6, line 21-10). Additionally, we have also clarified that we are studying prosocial intentions and self-reported moral constructs (e.g., pp. 6, line 11-2). Lastly, we also discuss our use of self-report measures as a limitation in the discussion and provide recommendations for future research (pp. 21-22, line 23-9). We believe these revisions mitigated the confusion between external validity and generalizability and we hope that our discussed limitations related to relying on self-reported dependent measures of morality are now more saliently stated.

7. *Minor points.*
Line 67. 'their immediate environment' is unclear; presumably this means 'their social relationships'

We have changed this sentence in accordance with your suggestion.

8. *Sentences beginning at line 72 are poorly written and ungrammatical*

Thank you for pointing this out. We have rewritten these sentences to improve language.

9. *109. What the four DVs are needs to be explained, either in the Introduction or when they are introduced in the results. At present, results come without any prior explanation of what the measures are. This can often be a problem in formats where the methods section comes at the end. A brief sentence or two introducing the measures*

(both the DVs and the predictors) is therefore needed in the introduction or in the results text.

We thank you for highlighting that the DVs needed further explanation. We have added a clear description of our four DVs and our primary predictors in the main text (pp. 6-8, line 11-4). Additionally, we have expanded the description of them in the Methods section (pp. 25-28, line 5-3).

10. 116. *If the alternative analyses use imputation, how do the main models treat missing data?*

The linear mixed effects models reported in manuscript use the Restricted Maximum Likelihood algorithm, which relies on pair-wise deletion before the maximum likelihood algorithm is applied. As we have a very large dataset and a relatively small amount of missing data, we opted for this algorithm based on the notion that we would not lose a lot of statistical power in these calculations. However, to demonstrate robustness of our findings, we also imputed missing data and we now clearly report the results of these models in the Supplementary Materials. To further enhance transparency, we have added additional information to the Methods section that clarifies these analytic decisions (pp. 28-29, lines 25-4).

11. *Table 1: For Gender, give the ref category to make the direction of effect interpretable*

Thank you noting this. We have now added the reference category to Table 1.

Reviewer 2:

1. *This paper reports the results of a survey conducted across many countries that measures subjective SES and then correlates this with measures of moral identity, morality as cooperation, moral circle, and prosocial behavior. The authors use these data to test pre-registered predictions that higher subjective SES would be associated with higher moral character and behavior. However, they found the opposite to those predictions. They found people reporting lower subjective SES reported higher moral character and behaviors. I have some concerns about the sampling strategy and measures used in the study that may limit the study's ability to test hypotheses about the relation between "chronic resource scarcity" and "Moral Character and Behavior" across countries. I am concerned about the sampling strategy used in this study, and how it could affect the author's ability to test the hypotheses across the countries. It sounds like the people in the different countries collecting data could collect data in any way that they wanted, but were requested to try and get a representative sample for age and gender (which many countries were unable to do). I would think it would be important to get representativeness for household income to provide a good test of this hypothesis across societies. I am assuming that many of the people collecting data were other social scientists and that if they were to recruit participants from their personal network (or University), then this would bias the samples to have a higher SES. This would be a serious limitation of the study, especially for being able to test for any differences between the countries. The differences between the countries could be due to the different sampling strategies used in each country. Also, this sample strategy could result in a restriction of range of SES, and not sample may people who have "chronic resource scarcity".*

We thank you for your careful review of our paper and the many constructive suggestions for improvements.

We now provide a clearer description of our sampling strategy in the Methods section of the manuscript (pp. 23-24, line 20-22). Specifically, no samples were recruited through close personal networks of the scholars and no student samples were used. The samples were collected as part of a large-scale international project with more than 200 researchers across the world (see van Bavel et al. 2022 in *Nature Communications*). All samples were recruited by online platforms or panel agencies, with participating researchers asked to fund their own data collection and collect samples which were nationally representative in respect to age, gender, and ethnicity. In total, data from 28 countries were nationally representative on these grounds, while the remaining data from 39 countries were based on convenience samples. In this revision, we have added information regarding which specific samples were representative, and which were not. Consequently, we have added robustness tests focusing on the samples that are nationally representative, both in the main text (Table 1, p. 10) and in the Supplementary Materials (Table S6). These results indicate that our main individual-level and country-level results largely replicate when focusing only on the representative samples. A similar relationship between strength of point estimates and type of sample was found in the publication first using this dataset, albeit with a completely different set of dependent and independent variables (van Bavel, et al. 2022). Importantly, our robustness tests support the notion that the between-country findings are not just a product of differences the sampling strategy used.

Lastly, to answer to your concern about the range of the SES measure, we provide an overview of the range of this measure in two tables with summary statistics in our Supplementary Materials (Table S28 for all 67 countries, Table S29 for nationally representative countries). These distributions clearly show that there is no restriction in our range of the SES measure in general, across the countries.

2. *I also have some concerns about the measures. Specifically, I am concerned that there is not equivalence of the factor structure of the scales across all the societies. This would affect how we can interpret the correlations across societies. I would suggest the authors report a structural equivalence test across the societies for moral identity and morality as cooperation.*

Thank you for this important comment. To test the equivalence of the factor structure across the 67 countries, we used multilevel confirmatory factor alignment, as suggested by Asparouhov and Muthén (2014), for the two multi-item scales in our analyzed data: Moral Identity and Morality-as-Cooperation.

For Moral Identity, the results showed that for factor loadings the scale exhibited 9% of non-invariance for item parameters and 8.2% for intercepts. Hence, following Asparouhov and Muthén's (2014) suggestion of a cut-off value of 25% in order to consider a scale non-invariant, we deem the scale suitable for use in the current investigation.

For Morality-as-Cooperation, we had similar results albeit with slightly higher levels of non-invariance. Here, the results show that for factor loadings the scale exhibited 9.8% of non-invariance for item parameters and 14.3% for intercepts. Again, following

the suggestions by Asparouhov and Muthén (2014), we deem the scale adequate for use.

Of note, our decision to use these scales as reliable measures of Moral Identity and Morality-as-Cooperation aligns with recent investigations using the same scales in similar multi-national collaborations (see Pavlovic et al. 2022, in press).

We report the overall results and the specifics of this multilevel confirmatory factor analysis alignment in the Methods section (pp. 25-27, line 5-2) and the full results of the analysis in the Supplementary Materials (Tables S17-S27).

Naturally, some degree of non-invariance in scales is almost unavoidable when dealing with a dataset of this magnitude, as we incorporate data from many different cultures and societies. However, based on the results from the multilevel confirmatory factor alignment as well as recent literature which has tested these scales across countries (see e.g., Curry et al., 2019 testing the Morality-as-Cooperation across 60 societies), we argue that the interpretation of our results, as currently reported in the manuscript, is not severely affected by the factor structure of the scales of Moral Identity and Morality-as-Cooperation.

3. *Furthermore, the authors only include a single item measuring “moral circle”. They also do not report evidence for the validity of this item. In fact, it sounds like this is the first time this item has been used in prior research. This severely limits the value of this item to be used to test theory. There is some data in the present study that could be used as a validity test: Does the measure of moral circle correlate with a relatively higher amount donated to an international, versus national, charity?*

Thank you for this comment. We have outlined the validity and the prior use of the moral circle framework in the Methods section (see p. 27, lines 3-9). While this measure is a single-item scale, it has been validated, studied and used extensively (see e.g., Waytz et al. 2019 in *Nature Communications* or Graham, et al. 2017 in *Cognition*). Moreover, we have added further argumentation for the importance of this construct in the main body of the manuscript (see p. 6, line 18-20; p. 20, line 1-13). We believe these revisions highlighting the content of the Moral Circle measure and its use in prior research strengthens the justification for including it in the present investigation.

To further situate this measure in relation to the other measures of morality, we report the correlation between Moral Circle and our measure of Prosocial Intentions in the Supplementary Materials along with our other measures of morality (Table S3 and S4). The results show that moral circle has a significant negative correlation with intention to donate to a national charity ($r = -0.12, P < .001$), while it has a significant positive correlation with the intention to donate to an international charity ($r = 0.08, P < .001$). When interpreting these correlations, it is important to note that Moral Circle measures the size of the “circle” of people for which individuals are concerned whether right or wrong is done towards them, which should not necessarily posit a strong correlation with prosocial intentions towards national and international charities (Crimston et al., 2018). For instance, one could imagine that while a participant might report to have a small Moral Circle (i.e., only be concerned about whether right or wrong is done towards close family and friends), such an individual could still be able to recognize the morality of helping others in need. In other words, the low correlations between

Prosocial Intentions and More Circle are to be expected because they capture different dimensions of moral decision-making (p. 3, line 11-17; p. 12, line 12-20).

4. *The authors operationalize “chronic resource scarcity” with a measure of subjective SES. I would have preferred a more objective measure of chronic resource scarcity, such as household income. The paper could benefit from at least reporting the relation between subjective SES and other measures of resource scarcity in prior work. Currently the paper just says this is a common way of measuring SES, while there are also several other methods used in the literature.*

Based on scarcity theory (de Bruijn & Antonides, 2021; Cannon et al. 2019; Shah et al. 2012), we argue that our measure of subjective socioeconomic status is actually the strong point of the current investigation, because we investigate specifically how the psychological experiences of relative deprivation is associated with moral judgment and decision-making. Hence, we identify that it is possible for deprived individuals living in very rich nations (e.g., Denmark) to “feel relatively poorer” compared to deprived individuals in objectively poorer nations (e.g., Nigeria, cautiously using the same example as that outlined by Reviewer 3). Consequently, the use of this measure is in line with recent work which has argued for the importance of investigating how subjective experiences of economic scarcity affects decision-making, while strongly advocating for more research on this subject (see de Bruijn & Antonides, 2021 for a review). Thus, although subjective and objective measures of SES are moderately correlated (Tan et al. 2020), the focus of the current paper is to investigate whether subjective experiences of economic scarcity is associated with moral decision-making.

Yet, to further strengthen our conclusions, we have augmented our data with a more objective measure of resource scarcity in the form of adjusted net-income per capita from the World Bank, to identify how this might be associated with morality as well. Here, we find that our objective measure of scarcity go in the same direction as the subjective one (see Supplementary Materials Table S11). However, because the GINI index is highly correlated with this measure of adjusted net-income per capita ($r = -0.44$, $P < .001$), this measure captures the variance attributed to the GINI index in our main models. Consequently, we included the models using the macro-level objective measure of adjusted net-income in our robustness models.

To enhance readability, we now underline in the introduction why we focus on subjective economic scarcity rather than more objective measures of scarcity in our study. To this end, we have added theoretical arguments based on the extensive literature on this subject (see pp. 7-8, line 3-4). Lastly, regarding the measurement of SES, we completely agree that existing research is scattered with different measures of this construct – a problem which has also received considerable attention (see e.g., Cirino et al., 2002; Diemer et al., 2013). However, these differences in measurement usually concern the measure of *objective* socioeconomic status (e.g., differences in weighting of education, occupation, and household income in an aggregated measure). In contrast, the measurement of subjective SES – as used in the current investigation (the McArthur SES ladder) – is one of the most widely adopted items to capture this construct and has previously been shown to exhibit sufficient reliability across different cultures (see e.g., Giatti, et al. 2012 in *BMC Public Health*).

5. *Another concern I have is that with such a large dataset, everything is statistically significant. For example, the Beta value for SES and moral circle is tiny. Is it even worth discussing this result as supporting a theoretical framework? How many participants would be required to detect the effect size in a subsequent study? I think it would be useful for a paper like this to identify an effect size that they would suggest is reasonable to interpret. I agree with the authors that small effect sizes can still be valuable, but there must be a point when we think its just not important, or it is so trivial that it is not really supporting a prediction from a theoretical framework.*

Thank you for this comment. We agree that because of the size of the dataset (considering the law-of-large-numbers in probability theory) interpreting statistical significance is not as meaningful as in studies with smaller sample sizes. This is also the reason as for why we do not center the manuscript around the significance level of our models, but merely around the size of the standardized beta values, which allow for easy comparison between models and mirror the presentation format in previous investigations using large datasets (see e.g., Götz et al., 2020 in *Nature Human Behaviour*). We argue that utilizing this rich dataset to uncover differences in psychological constructs linked to morality, even if such effects might be small by conventional standards used primarily in experimental research, is a strong point of the current investigation, as “*complex psychological phenomena [...] are likely to be influenced by hundreds, if not thousands, of factors [...], so small effects are to be expected especially when examined in the uncontrolled context of real-world settings [...]*” (Götz et al., 2020 p. 1140). Therefore, we decided to keep the measure of Moral Circle in our analyses, considering that the association for this measure is in the opposite direction of our theoretically informed pre-registered hypothesis, but also because it measures a distinct dimension of moral decision-making. (p. 6, line 18-20). Having noted that, we agree that the size of the “Moral Circle” effect is indeed *very* small (pp. 15-16, line 21-5). Consequently, to address your concerns, we have changed the discussion of this result regarding its implications to articulate those in a more tentative manner (p. 20, line 1-13). We sincerely hope this will suffice.

6. *In figure 2b, it looks like there is no relation between SES and morality as cooperation, but then there is a negative relation between SES and moral circle. However, in Table 1 the beta value is larger for morality as cooperation, compared to a moral circle. Why is there a discrepancy between Table 1 and Figure 2?*

We thank you for raising this point, which reflects a mistake in our initial submission. The table has now been corrected. Furthermore, following the recommendation by Reviewer 1, we have changed the visualization of the mentioned relationship to more clearly show both the within-and between-country variation in our results (see Figure 2, p. 12).

7. *When I was reading the paper, I did not know the definition of many concepts when they were first raised in the paper, such as moral identity, moral circle, and morality-as-cooperation. I also don't understand the definition of morality that is used in this work. Of course, prosocial behavior (i.e., behaviors that benefit others) can also be immoral, such as helping a prisoner escape from prison. And people can cooperate together to do some very horrible, immoral behaviors. In general, I thought the paper could be improved substantially in terms of communicating about the concepts discussed in the paper. For example, a major conclusion of the paper is that: “chronic*

resource scarcity...results in more moral outcomes”, and I was left confused about what “moral outcomes” even mean in this context.

Thank you for this comment, which aligns with one of the comments from Reviewer 1. In the revised manuscript, we have added a clear definition of our constructs in the main body of the manuscript and added supporting information of the origin of these constructs (p. 6, line 11-22) Furthermore, we have tried our best to enhance the communication of the main terms discussed in our paper by clearly conceptualizing moral decision-making and the multidimensional nature of morality (p. 3, line 11-17; p. 5, line 14-20). Additionally, we have streamlined and simplified the terminology to systematically refer to our four dependent variables as measures of moral decision-making rather than using terms such as “moral outcomes”. We have also altered the major conclusion that you outline to describe our results more precisely (instead of referring to “moral outcomes”; p. 17, line 3-12). Finally, we have a clear description of our measures in the method section of the paper (pp. 25-28, lines 5-3).

Reviewer 3:

1. *In this paper the authors report on the relationship between measures inequality (Subjective personal SES, National GINI coefficient) and various measures of “morality” (moral identity, morality-as-cooperation, size of moral circle, and [hypothetical] prosocial behaviour). This paper aims to address a very interesting question that has been hotly debated: Does higher economic status affect one’s moral values and behavior? As the authors explain in a clearly written introduction, the literature on this is mixed (although there are factors that may explain the apparent contradictions in the literature). The present study aims to address this question using a massive data set (over 46,000 participants from 67 nations). They report that economic deprivation is, in various ways, associated with more “moral” outcomes. This study has some notable strengths, such as the aforementioned large international sample as well as some sophisticated statistical features. However, it seems to me that it is flawed in some very basic ways. It’s not clear to me whether these problems are correctable. It seems that they should be, but at the same time, the fact that the authors didn’t address these problems already makes me worry that the fixing the problems leads to a very different set of conclusions.*

Thank you for your careful reading of our paper and for your very constructive comments.

2. *The main problem is that authors use measures of SES/inequality without taking into account absolute wealth—both at the individual and national level. The SES measure asks for a subjective judgment of where the respondent falls on an SES ladder. As one might expect, people’s responses are heavily influenced by the local context. As an illustration, the mean SES rating on this measure is *lower* in wealthy Denmark than in poor Nigeria. It doesn’t follow from this that this measure is meaningless. It tells us that they average Dane sees themselves as lower on the *local* SES ladder than the average Nigerian does. But this is not a meaningful measure of actual SES, deprivation, or anything of that sort. The same holds for The GINI coefficient, the other major predictor variable. Germany and Bangladesh have similar GINI coefficients, but obviously the experience of inequality and deprivation are very different for the average German as compared to the average Bangladeshi. The way to deal with this is obvious: One must (at the very least) account for wealth at the individual and national levels in*

when using measures of perceived relative SES and (objective) national-level inequality. This is so obvious, I was surprised that the authors didn't do this. And I was especially surprised given that the analysis plan was pre-registered. So, I looked at the pre-registration and, sure enough, the authors specify household income as a key predictor variable. But this variable is not used at all in the paper.

Thank you for this comment, which partly aligns with that of Reviewer 2. Based on scarcity theory (de Bruijn & Antonides, 2021; Cannon et al. 2019; Shah et al. 2012), we argue that our measure of subjective socioeconomic status is actually the strong point of the current investigation, because we investigate specifically how subjective, as opposed to objective, experiences of economic scarcity is associated with morality. Hence, we identify that it is possible for deprived individuals living in very rich nations (e.g., Denmark) to “feel relatively poorer” compared to deprived individuals in objectively poorer nations (e.g., Nigeria, cautiously using the same example as yourself). (See Supplementary Table S28 and S29 for an overview of the range the subjective SES measure and country-level GINI coefficients).

To add further robustness to our results, we have augmented our data with a more objective measure of economic scarcity in the form of adjusted disposable net-income per capita from the World Bank, to identify how this might be associated with morality as well. Here we find that our objective measure of scarcity go in the same direction as the subjective one (Supplementary Table S11). However, because the GINI index is highly correlated with this measure of national net income per capita ($r = -0.44$, $P < .001$), this measure captures the variance attributed to the GINI index in our main models. Consequently, we included the models using the macro-level objective measure of adjusted disposable net-income in our robustness models.

We have now elaborated more on the importance of studying subjective economic scarcity in the introduction of the manuscript, while also adding theoretical arguments based on the extensive literature on this subject (p. 3, line 1-10; pp. 7-8, lines 3-4).

Concerning our specific measure of subjective socioeconomic status, the McArthur ladder scale, we argue that the use of this scale is actually a strong point of the current investigation. The scale has been used extensively in previous literature to capture *subjective* socioeconomic status and has been shown to exhibit sufficient reliability across different cultures (see e.g., Giatti, et al. 2012 in *BMC Public Health*). Because this scale is able to capture contextual differences in income, education, and job status regardless of more objective measures of socioeconomic status, we argue that it fulfills the purpose of identifying directly how individuals *perceive* their own socioeconomic status *across* societies.

With respect to the macro-level measure of economic inequality, the GINI index, we recognize that experiences of economic inequality might differ across cultures. However, the importance of including this measure follows previous investigations (e.g., Pickett & Wilkinson, 2015; Wilkinson & Pickett, 2011), which have provided evidence for the importance of considering national level economic inequality instead of focusing on objective measures of household income. That is, prior research has shown that societal problems such as crime rates, mental health problems, etc. have no correlation with objective measures of household income, while being strongly correlated with national measures of economic inequality. Therefore, we argue that our

use of the individual level subjective SES complemented by the national level of income inequality achieves the goal of identifying how individuals who perceive themselves to be living with less resources and in countries characterized by economic inequality act in regards to questions concerning moral thought and intention.

Lastly, regarding the pre-registration on using individual income levels in our analyses, we were initially of the impression that our data would include this measure because the Danish dataset (which we collected for the project) did. However, this was not the case. That is, we only had access to our own dataset when we drafted the pre-registration to use the entire dataset from the 67 countries and thus did not anticipate this lack of data on individual household income. We have clearly noted this deviation from the pre-registration in the method section of the revised manuscript (p. 23, line 7-19).

- 3. I also have concerns about the key outcome measures related to morality. In the introduction it sounds as if the authors are weighing in on the debate about whether higher SES makes people more or less “moral”, meaning more or less inclined to do things that are widely regarded as morally good (e.g. being generous) or morally bad (e.g. cheating). But only the generosity measure falls into this category. And here the authors rely on self-report of generosity, which could vary enormously by culture and/or be generally unreliable. At the very least, one would want to corroborate these subjective self-reports of generosity with country-level data on things like actual charitable giving. And for the other measures, it’s not clear whether they exhibit the same kind of local/cultural relativity that the SES ladder measures do. For example, what counts as having a large vs. small moral circle in Nigeria vs. the US?*

Thank you for this comment, which has made it clear to us to that the central concepts from moral psychology used in this manuscript needs to be more clearly described. In response to your comment, we would like to note that a recent evolutionary conceptualization of morality argues that “*morality consists of a collection of biological and cultural solutions to the problems of cooperation recurrent in human social life*” (Curry et al. 2021, p. 2). Hence, while morality concerns considerations on what is “right” and “wrong,” which could either facilitate or hurt cooperation, we specifically argue that the current investigation is more nuanced than that. In other words, because we assess different underlying constructs of what constitutes morality, we are both weighing in on the discussion on whether lower rather than higher SES makes people exhibit more or less prosocial moral intentions, but we also include a selection of other measures (our three remaining DVs) that aim to capture more general aspects of moral judgment and decision-making. These measures concern what individuals *perceive* as being morally good or bad or how they perceive themselves as exhibiting such moral thoughts and behavior towards others. Indeed, our measure of Moral Identity concerns how individuals perceive themselves as moral individuals and how they exhibit such characteristics in their daily life, whereas our measure of Morality-as-Cooperation examines the weight of importance that individuals attribute to a set of seven generalized moral behaviors (e.g., reciprocation) considered to be universally “moral behaviors” across societies (see Curry et al., 2019 testing this construct across 60 societies). Similarly, our measure of Moral Circle concerns the number of people that individuals are concerned about whether “right” or “wrong” is done toward them. Hence, as now clarified in the revised manuscript (p. 3, line 11-17; pp. 5-7, lines 21-2; pp. 18-20, lines 14-13), our study expands previous research considerably by not only

focusing on *one* form of morality (Prosocial Intentions), but instead including a selection of constructs that have been discussed as central to human morality.

The measure of Prosocial Intentions naturally vary across cultures, but we clearly account for that in our models by the use of random-intercepts in our multilevel modeling. As such, we show how individuals might self-report their prosocial intentions if they were informed that they will receive a median income in their respective economy, but we make this measure more reliable and valid *across* countries considering that we also incorporate national median income in every respective country. We argue that intention to donate to charity (i.e., helping others in need) can be considered a universal moral value across the world (e.g., Aknin et al., 2013). Hence, in defense of this measure, our results show that even after having accounted for cross-cultural differences, we find a significant negative association between SES and prosocial donation intentions (Table 1, p. 10 and Supplementary Tables S7-S10).

When it comes to the comment of corroborating the results of the current study using reports on actual charitable giving, we have added a discussion of this that outline how our findings align with prior literature as well as recent results from the World Giving Index (p. 18, line 14-24). Due to the nature of the data from the World Giving Index, it was not possible for us to augment our data with this additional data source. Yet, the most recent report on this index clearly shows that countries that are traditionally considered as non-WEIRD societies with large populations of people living with chronic scarcity seem to top the charts when it comes to prosocial behaviors such as charitable giving, helping strangers, and volunteering time. While being mindful of not overstating the implications of the current research, such findings seem to be in line with our obtained results. We sincerely hope this answers your concerns.

4. *Finally, it's not clear whether the preregistration is really a preregistration. The authors explain that "some" of the data have already been collected and then refer to this explanation: "The data has already been collected and is a part of an international project on moral psychology during the corona (COVID-19) pandemic. The data will initially be used in the paper for which it was collected." This doesn't tell us which data were collected prior to the pre-registration and which were not. And, most importantly, it doesn't say whether the authors were aware of the outcomes of these analyses prior to the pre-registration. Given that the analyses reported on in the paper differ from the analyses outlined in the pre-registration, it seems that they didn't know the outcomes in advance, which is good—except that we only know this because the authors deviated so significantly from their pre-registered plan. This is a rich data set, and it seems possible to me that the authors could analyze it in a way that makes more sense. However, the fact that the authors pre-registered a more sensible approach and then abandoned it suggests that this would lead to rather different conclusions. Likewise, it's possible that the data could be augmented with nation-level objective measures that would allow them to more directly address the questions they seem to want to answer in the introduction. But, in its present form, this paper has some very significant limitations.*

Thank you for this comment. Following the guidelines from the leading authors of the original project (van Bavel et al., 2022) researchers who wanted to use the dataset for secondary analysis were obliged to pre-register their research outline at <https://icsmp-covid19.netlify.app> using one of the current available options (e.g., AsPredicted or

OSF). This procedure was the same for all projects utilizing this dataset; see e.g., Cutler et al. (2021), published in *Nature Aging* (<https://www.nature.com/articles/s43587-021-00118-3>). In essence, prior to submitting the pre-registration, we did not have access to the cross-national dataset; rather, we only had access to the Danish sample which we collected for the overall study, which included 67 countries. That is, all data were collected before this project was initiated, but we did not have access to the dataset before we made the pre-registration. We have clarified this in the revised version of the manuscript (p. 23, line 7-19).

As outlined in the method section, we only deviated from the pre-registered analyses in terms of implementing more robust and advanced statistical analyses. In particular, the currently employed analytic approach is more sensible and robust than the one we pre-registered, because our used models clearly account for the natural clustering in the data (countries) as well as within and between-country differences (see e.g., Čepulić et al., 2021 or Randall et al., 2021 using similar approaches to analyze large, clustered datasets). That is, while linear mixed effects models rests on many of the same assumptions as fixed effects models (i.e., OLS), the inclusion of random effects allows the researcher to account for possible differences in intercepts and slopes per clustering in the data, hence for instance guarding against concluding on a possible Yule-Simpson effect. Yet, based on your thoughtful comment, we have expanded on this issue in the Methods section to further justify why our selected analytic strategy provides a more robust test of our theorizing (pp. 28-29, line 20-25).

Lastly, as outlined above, we have augmented our dataset with a macro-level measure on adjusted disposable net-income per capita from the World Bank to explore how this might affect our proposed relationship. Here, we find that our objective measure of scarcity goes in the same direction as the subjective one (see Supplementary Table S11). However, because the GINI index is highly correlated with this measure of adjusted disposable net-income per capita ($r = -0.44$, $P < .001$), this measure captures the variance attributed to the GINI index in our main models. Consequently, we included the models using the macro-level objective measure of adjusted disposable net-income in our robustness models.

We sincerely hope that the implemented changes will suffice to address the concerns you outlined in your review.

References:

- Aknin, L. B., Barrington-Leigh, C. P., Dunn, E. W., Helliwell, J. F., Burns, J., Biswas-Diener, R., Kemeza, I., Nyende, P., Ashton-James, C. E., & Norton, M. I. (2013). Prosocial spending and well-being: cross-cultural evidence for a psychological universal. *Journal of Personality and Social Psychology*, 104(4), 635.
- Asparouhov, T., & Muthén, B. (2014). Multiple-Group Factor Analysis Alignment. *Structural Equation Modeling: A Multidisciplinary Journal*, 21(4), 495-508.
- Cannon, C., Goldsmith, K., & Roux, C. (2019). A self-regulatory model of resource scarcity. *Journal of Consumer Psychology*, 29(1), 104-127.
- Čepulić, D. B., Travaglino, G. A., Chrona, S., Uzelac, E., Jeftić, A., Reyna, C., & Kowal, M. (2021). Iron fists and velvet gloves: Investigating the associations between the

stringency of governments' responses to COVID-19, stress, and compliance in the early stages of the pandemic. *British Journal of Social Psychology*.

- Cirino, P. T., Chin, C. E., Sevcik, R. A., Wolf, M., Lovett, M., & Morris, R. D. (2002). Measuring Socioeconomic Status: Reliability and Preliminary Validity for Different Approaches. *Assessment*, 9(2), 145-155. <https://doi.org/10.1177/10791102009002005>
- Crimston, C. R., Hornsey, M. J., Bain, P. G., & Bastian, B. (2018). Toward a Psychology of Moral Expansiveness. *Current Directions in Psychological Science*, 27(1), 14-19. <https://doi.org/10.1177/0963721417730888>
- Curry, O. S., Alfano, M., Brandt, M. J., & Pelican, C. (2021). Moral molecules: Morality as a combinatorial system. *Review of Philosophy and Psychology*, 1-20.
- Curry, O., Whitehouse, H., & Mullins, D. (2019). Is it good to cooperate? Testing the theory of morality-as-cooperation in 60 societies. *Current Anthropology*, 60(1).
- Cutler, J., Nitschke, J. P., Lamm, C., & Lockwood, P. L. (2021). Older adults across the globe exhibit increased prosocial behavior but also greater in-group preferences. *Nature Aging*, 1(10), 880-888.
- de Bruijn, E.-J., & Antonides, G. (2021). Poverty and economic decision making: a review of scarcity theory. *Theory and Decision*, 1-33.
- Diemer, M. A., Mistry, R. S., Wadsworth, M. E., López, I., & Reimers, F. (2013). Best practices in conceptualizing and measuring social class in psychological research. *Analyses of Social Issues and Public Policy*, 13(1), 77-113.
- Giatti, L., Camelo, L. d. V., Rodrigues, J. F. d. C., & Barreto, S. M. (2012). Reliability of the MacArthur scale of subjective social status-Brazilian Longitudinal Study of Adult Health (ELSA-Brasil). *BMC public health*, 12(1), 1-7.
- Götz, F. M., Stieger, S., Gosling, S. D., Potter, J., & Rentfrow, P. J. (2020). Physical topography is associated with human personality. *Nature Human Behaviour*, 1-10.
- Graham, J., Waytz, A., Meindl, P., Iyer, R., & Young, L. (2017). Centripetal and centrifugal forces in the moral circle: Competing constraints on moral learning. *Cognition*, 167, 58-65.
- Mani, A., Mullainathan, S., Shafir, E., & Zhao, J. (2013). Poverty impedes cognitive function. *Science*, 341(6149), 976-980.
- Pavlović, T. A., F.; De, K.; Maglić, M.; Donnelly Kehoe, P. A.; Payán-Gómez, C.; ... van Bavel, J. J. . (2022). Predicting attitudinal and behavioral responses to COVID-19 pandemic using machine learning. *PNAS Nexus* [Accepted].
- Pickett, K. E., & Wilkinson, R. G. (2015). Income inequality and health: a causal review. *Social science & medicine*, 128, 316-326.

- Randall, A. K., Leon, G., Basili, E., Martos, T., Boiger, M., Baldi, M., Hocker, L., Kline, K., Masturzi, A., Aryeetey, R., Bar-Kalifa, E., Boon, S. D., Botella, L., Burke, T., Carnelley, K. B., Carr, A., Dash, A., Fitriana, M., Gaines, S. O., . . . Chiarolanza, C. (2021). Coping with global uncertainty: Perceptions of COVID-19 psychological distress, relationship quality, and dyadic coping for romantic partners across 27 countries. *Journal of Social and Personal Relationships*, 39(1), 3-33. <https://doi.org/10.1177/02654075211034236>
- Shah, A. K., Mullainathan, S., & Shafir, E. (2012). Some consequences of having too little. *Science*, 338(6107), 682-685.
- Van Bavel, J. J., Cichocka, A., Capraro, V., Sjøstad, H., Nezlek, J. B., Pavlović, T., Alfano, M., Gelfand, M. J., Azevedo, F., Birtel, M. D., Cislak, A., Lockwood, P. L., Ross, R. M., Abts, K., Agadullina, E., Aruta, J. J. B., Besharati, S. N., Bor, A., Choma, B. L., . . . Boggio, P. S. (2022). National identity predicts public health support during a global pandemic. *Nature Communications*, 13(1), 517.
- Waytz, A., Iyer, R., Young, L., Haidt, J., & Graham, J. (2019). Ideological differences in the expanse of the moral circle. *Nature Communications*, 10(1), 1-12.
- Wilkinson, R., & Pickett, K. (2011). *The spirit level: Why greater equality makes societies stronger*. Bloomsbury Publishing USA. New York.
-

Reviewers' Comments:

Reviewer #1:

Remarks to the Author:

I am happy with the revisions: the authors have done pretty much all the things I suggested they do, and to my mind they make the picture clearer. My only remaining comments are a few typos (important to fix these):

Line 49: Experiences of chronic relative economic scarcity is a structural characteristic of modern...grammar, either experience is singular or are is plural.

Table 1. There is a -0.8 in the first row that I am sure must be supposed to be -0.08.

Table 2. Aren't these random slopes rather than random intercepts? If I understand correctly, each country should have a random intercept and a random slope term, and it is the random slope term whose sign matters here. I am bit puzzled these numbers are so evenly split between positive and negative. Could you check this, and also check by running the regressions in each country separately and just seeing how many individual slopes are negative and how many positive (should give similar picture).

Reviewer #2:

Remarks to the Author:

This is a revised version of a manuscript that reports the results of a survey conducted across 60+ countries on the association between subjective SES and four different measures of morality. The article has improved since the last version by addressing several of the reviewer comments. However, there remains an outstanding limitation of the paper that the authors are unable to address in the revision, and that is the reliance on the measure of subjective SES as an operationalization of (relative) resource scarcity.

The introduction draws from a literature that has focused on resource scarcity and its implications for cognition and behavior. The authors argue in favor of their measure of subjective SES as an operationalization of resource scarcity by saying "...prior research has found that experiences of relative scarcity can shift cognitive attention and alter decision-making strategies more than extreme and thus absolute scarcity (i.e., extreme poverty). Therefore, we rely on subjective SES to measure individual-level experiences of relative economic scarcity to gain a more nuanced understanding of the link between scarcity experiences and morality." First, in the above sentences "experiences" is used to refer to "subjective evaluations" of resource scarcity, and I find this a bit misleading, because subjective evaluations of SES or resource scarcity, may not always correlate with the actual, objective experience of resource scarcity. Second, the article does not seem to acknowledge possible confounds with individual differences in the measure of subjective SES (e.g., honesty-humility, conscientiousness), which could account for some of the relations observed between subjective SES and measures of morality. Furthermore, in the argument above, the authors draw attention to the subjective measure having a strong association with outcomes, compared to the objective measure. However, I don't find this to be a convincing argument to study this variable. Instead, I think the arguments should be based on the validity of the measure. The authors attempt to resolve the issue of the reliance on a subjective measure of SES by including an objective societal measure of wealth inequality across societies. However, It was not clear to me how the theories about resource scarcity and thinking/behavior in the introduction are also tested by cross-societal differences in the GINI index. The authors state: "By focusing directly on the dispersion of financial resources, this is a more objectively oriented national-level indicator of experiences of relative economic scarcity. That is, this indicator constitutes a measurement of the magnitude of differences in wealth accumulation that individuals in a specific economy are exposed to, which has been discussed as an important factor linked to psychological differences in experiences of relative economic scarcity". The GINI indexes wealth discrepancies in society, and I would think that in societies with high wealth inequality, people don't often interact with others who have all the resources, and instead their comparisons with others would be comparisons with people who have relatively similar resources at their disposal. I would prefer to see (1) the authors reporting data that supports the idea that subjective SES is correlated with GINI, and (2) more

direct discussion about the strengths and limitations in how this cross-societal index is being used to test the theories in the introduction of the paper. Still, I think actual household (or individual) income would be a better objective indicator of resource deprivation, and for testing the ideas outlined in the introduction.

The authors conclude "we find evidence for the notion that experiences of relative economic scarcity as a chronic state is not associated with a "depletion" of moral character or less prosocial intentions. Instead, it seems that such experiences of economic scarcity are associated with a stronger preference for identifying and acting as a moral individual, engaging in cooperative behaviours with a clear moral foundation (e.g., helping kin or reciprocating), and having the intention to engage in prosocial charitable giving." Similarly, in the discussion the authors state: "... individuals who experience relative economic scarcity as a chronic state not only seem more inclined to perceive themselves as moral individuals (i.e., 320 Moral Identity), but also seek to project such morality-related aspects towards their peers and in-group members (i.e., Morality-as-Cooperation, Moral Circle, and Prosocial Intentions)." These conclusions are a bit misleading, because the authors did not measure the actual experience of economic scarcity, but how people subjectively think about their SES, and I think this should be made much more explicit in the discussion of the paper.

Overall, I think the authors do report a study that adds value to this developing literature. I think the paper should be published, and should be given serious consideration by scholars working on this topic. That said, I question whether the methods used to operationalize the variables in this study provide a strong test of the ideas outlined in the introduction. Furthermore, the authors could draw more attention to the limitations of using subjective SES as an index of chronic resource scarcity. For these reasons, the paper may be more suitable for a specialty journal.

Reviewer #3:

Remarks to the Author:

I think the authors have done an excellent job of addressing the concerns raised in my original review and that the manuscript has also benefitted from suggestions from the other reviewers.

I should note, with apologies, that my main concern about the original manuscript was based on a misunderstanding. I thought that the authors were primarily interested in objective economic scarcity. Although they were careful to say "relative" scarcity in many places, much of the literature referenced seems to focus on objective scarcity. This framing led me to believe that objective deprivation was the key variable, despite it's being accessed through self-report. I now understand that subjective scarcity is the primary target, and I think the revised manuscript makes this more clear. But this makes me wonder whether much of the debate that frames this paper can be resolved by distinguishing between objective and subjective scarcity. I'm not sure, and I leave it to the authors to add something about this if they think it would be illuminating.

Revision Notes

We would like to thank the three reviewers for the valuable comments to our manuscript. Below are our point-by-point responses to all comments. The comments from the reviewers are numbered and italicized. Our responses appear in regular type.

Reviewer 1

- 1. I am happy with the revisions: the authors have done pretty much all the things I suggested they do, and to my mind they make the picture clearer. My only remaining comments are a few typos (important to fix these): Line 49: Experiences of chronic relative economic scarcity is a structural characteristic of modern...grammar, either experience is singular or are is plural. Table 1. There is a -0.8 in the first row that I am sure must be supposed to be -0.08. Table 2. Aren't these random slopes rather than random intercepts? If I understand correctly, each country should have a random intercept and a random slope term, and it is the random slope term whose sign matters here. I am bit puzzled these numbers are so evenly split between positive and negative. Could you check this, and also check by running the regressions in each country separately and just seeing how many individual slopes are negative and how many positive (should give similar picture).*

We thank you for your positive evaluation of the revised manuscript and for noting these typos, which we have corrected in the revised version of the manuscript. We have changed the sentence in line 49 (p. 3, line 1) and corrected the number in Table 1. In Table 2, these were random intercepts. As specified in our Methods section, we only specified the models reported in Table 1 with random intercepts. We did so because the models were unable to converge successfully with random slopes, which was also the reason why we decided to supplement our main analysis with the Nested OLS models reported in the Supplementary Results, as well as the models separating the within/between country effects as suggested by yourself (the models reported in Figure 2 are based on these). We agree that the country-level slope terms make more sense to report here. Therefore, we now report the direction of the slopes for the Nested OLS models (equivalent to a random slope setup) in Table 2. We have double-checked the numbers and made sure everything is reported correctly. Again, thank you for your careful consideration.

Reviewer 2

- 1. This is a revised version of a manuscript that reports the results of a survey conducted across 60+ countries on the association between subjective SES and four different measures of morality. The article has improved since the last version by addressing several of the reviewer comments. However, there remains an outstanding limitation of the paper that the authors are unable to address in the revision, and that is the reliance on the measure of subjective SES as an operationalization of (relative) resource scarcity.*

We thank you for your careful review of our revised manuscript. We are pleased that you believe our manuscript has been improved. Below we outline our responses to your remaining comments with direct references to changes in the revised manuscript. We sincerely hope that these revisions satisfactory address your concerns.

2. *The introduction draws from a literature that has focused on resource scarcity and its implications for cognition and behavior. The authors argue in favor of their measure of subjective SES as an operationalization of resource scarcity by saying “...prior research has found that experiences of relative scarcity can shift cognitive attention and alter decision-making strategies more than extreme and thus absolute scarcity (i.e., extreme poverty). Therefore, we rely on subjective SES to measure individual-level experiences of relative economic scarcity to gain a more nuanced understanding of the link between scarcity experiences and morality.” First, in the above sentences “experiences” is used to refer to “subjective evaluations” of resource scarcity, and I find this a bit misleading, because subjective evaluations of SES or resource scarcity, may not always correlate with the actual, objective experience of resource scarcity. Second, the article does not seem to acknowledge possible confounds with individual differences in the measure of subjective SES (e.g., honesty-humility, conscientiousness), which could account for some of the relations observed between subjective SES and measures of morality. Furthermore, in the argument above, the authors draw attention to the subjective measure having a strong association with outcomes, compared to the objective measure. However, I don’t find this to be a convincing argument to study this variable. Instead, I think the arguments should be based on the validity of the measure.*

We thank you for highlighting that the conceptualization of subjective economic scarcity needed to be further clarified and differentiated from objective experiences of scarcity. We acknowledge that “experiences” used to describe “subjective evaluations” of economic scarcity may be confused with objective circumstances of living in resource scarcity. For reasons of conceptual clarity, we now use the concepts “subjective experiences of economic scarcity” and “perceptions of economic scarcity” throughout the entire manuscript (as in, for example, Jachimowicz et al., 2022) to denote subjective evaluations of resource scarcity. This approach follows previous investigations (e.g., Kraus et al., 2009) as well as the current bias-free language guidelines from the American Psychological Association (APA) on socioeconomic status¹, which details that “Socioeconomic status (SES) encompasses not only income but also educational attainment, occupational prestige, and subjective perceptions of social status and social class.” We also clearly outline both in the abstract (lines 2-4) and in the introduction (lines 1-5) how subjective experiences of scarcity refer to a subjective feeling of lack or economic resources compared to others – that is, a result of social comparison – and that it is different from the actual circumstances of living in economic scarcity. With these revisions, the introduction now conveys more clearly that subjective SES is used in the literature an indicator of these subjective experiences of resource scarcity (lines 5-7). We return to this point and elaborate on it at p 7. (lines 17-23), where we argue for the importance of investigating subjective rather than objective perceptions of economic scarcity. We also discuss how subjective experiences of economic scarcity are different from objective circumstances of economic scarcity (see also Tan et al., 2020), while underscoring these points in our final Discussion (p. 24, lines 1-8).

We acknowledge that our measure of subjective SES, like any other individual-level measure, could be influenced by other individual differences, such as the personality dimensions you outline. We have now reflected on this point in our concluding Discussion (p. 25, lines 1-11) to explicitly address your thoughtful remark.

¹ <https://apastyle.apa.org/style-grammar-guidelines/bias-free-language/socioeconomic-status>

Unfortunately, we do not have available individual-level data on, for instance, honesty-humility or conscientiousness to control for these factors in our analyses. Therefore, we now emphasize that future work should examine our studied associations in more realistic environments, preferably using behavioral measures and experimental approaches or a larger range of control variables, including personality dimensions such as conscientiousness to allow for causal inferences (p. 25, lines 8-11).

The use of the MacArthur ladder as a way to measure subjective socioeconomic standing in society is a reliable and well-established measure, which has shown strong construct validity (Cundiff et al., 2013) as well as high predictive validity in previous research regarding outcomes associated with health and well-being (e.g., Boon & Farnsworth, 2011; Kuper & Marmot, 2003; Singh-Manoux, et al., 2005). Further, recent work has argued that a reason for this high predictive validity is that the MacArthur ladder scale is able to measure two distinct constructs that capture inherent characteristics of how people perceive and subjectively experience economic scarcity: economic circumstances and social status (Galvan et al. 2022). Thus, the MacArthur ladder scale should represent a valid way to measure subjective SES, which we have sought to clarify in this revision (pp. 7-8, lines 23-5). While we agree that using an objective measure of economic scarcity could also provide an interesting theoretical contribution to the current literature on scarcity, the current work is focused on how subjective experiences of economic scarcity might be associated with morality. To be responsive, however, we now also discuss the need for further studies on objective measures of economic scarcity as a clear avenue for future research on moral judgment and decision-making (p. 24, lines 1-11; p. 24, lines 12-19).

3. *The authors attempt to resolve the issue of the reliance on a subjective measure of SES by including an objective societal measure of wealth inequality across societies. However, It was not clear to me how the theories about resource scarcity and thinking/behavior in the introduction are also tested by cross-societal differences in the GINI index. The authors state: “By focusing directly on the dispersion of financial resources, this is a more objectively oriented national-level indicator of experiences of relative economic scarcity. That is, this indicator constitutes a measurement of the magnitude of differences in wealth accumulation that individuals in a specific economy are exposed to, which has been discussed as an important factor linked to psychological differences in experiences of relative economic scarcity”. The GINI indexes wealth discrepancies in society, and I would think that in societies with high wealth inequality, people don’t often interact with others who have all the resources, and instead their comparisons with others would be comparisons with people who have relatively similar resources at their disposal. I would prefer to see (1) the authors reporting data that supports the idea that subjective SES is correlated with GINI, and (2) more direct discussion about the strengths and limitations in how this cross-societal index is being using to test the theories in the introduction of the paper. Still, I think actual household (or individual) income would be a better objective indicator of resource deprivation, and for testing the ideas outlined in the introduction.*

Thank you for this comment. We would like to begin our reply by noting that we do not attempt to “resolve” any issues related to using a subjective measure of SES. The contribution of this manuscript is aimed at understanding how perceptions and subjective experiences of economic scarcity measured at the individual-level (by

subjective SES) and at the macro-level (by the GINI coefficient) is associated with the four dependent measures of moral judgment. As now clarified in the revised version of the manuscript, we include GINI as a macro-level indicator of perceptions of economic scarcity, following previous investigations in this domain (e.g., Jachimowicz et al. 2020; Sommet et al., 2018). That is, the GINI index is utilized as an indicator of the exposure to salient differences in financial resources leading to different perceptions of economic scarcity. To make this more explicit, we have expanded on this argument in our revision (pp. 8-9, lines 6-2). This argument is in line with previous work, which has shown that economic inequality impairs the psychological health of people facing scarcity (Sommet et al., 2018), because a central component of perceptions of economic scarcity is rooted in social comparison (Goldsmith et al. 2020) – something that is exacerbated by economic inequality (Cheung & Lucas, 2016). In other words, one of the central components of scarcity theory is that the behavioral manifestations resulting from scarcity are rooted in the subjective component of perceiving to “have less than what is needed” and to perceive to be worse off, in terms of available resources, than others (Ren et al., 2022; Shah et al. 2012).

To address your thoughts about whether individuals living in contexts of high (vs. low) inequality perceive economic scarcity to be more prevalent, we would like to raise the following points. First, we agree that this is an interesting avenue for further investigations, which we now also note in the section on directions for future research (p. 24, lines 12-19). Second, while we cannot directly measure social comparisons in the current data, it is worth mentioning that people living in places of high economic inequality are regularly exposed to notable differences in resource acquisition according to previous research (e.g., Cheung & Lucas, 2016; Sands & de Kadt, 2020). As an example, from the most economically unequal country in the world (South Africa), labor market² and internet usage³ data from the international statistics organization Statista reveal that close to 1.2 million South Africans work in private households (e.g., as servants or housekeepers); more than 800,000 women are domestic workers; and over 80% of South Africans have access to the internet. Hence, it seems plausible that low SES individuals in South Africa are regularly exposed to (and compare themselves with) fellow South Africans with greater economic resources than themselves. Accordingly, the claim that individuals living in highly economically unequal places are not exposed to differences in wealth on a regular basis is a claim that we and other scholars do not necessarily agree with (Buttrick & Oishi, 2017; Cheung & Lucas, 2016). Moreover, individual-level measures of the distribution of subjective socioeconomic status are not necessarily highly correlated with the macro-level measure of GINI (e.g., Di Domenico & Fournier, 2014).

Concerning your point related to more objective indicators of resource deprivation, we have further elaborated on the strengths and limitations associated with addressing our main objective through subjective experiences of economic scarcity (p. 24, lines 12-19). As noted in one of our replies above, we agree that using objective household income could provide an interesting investigation into how objective experiences of resource scarcity might affect morality or judgment and decision-making in general, which we have carefully outlined in the Discussion section when articulating our directions for future research (p. 24, lines 1-11). Having noted that, the current

² <https://www.statista.com/statistics/1129815/number-of-people-employed-in-south-africa-by-industry/>

³ <https://www.statista.com/statistics/972866/south-africa-mobile-internet-penetration/>

manuscript provides a robust investigation of the association between individual and macro-level perceptions of economic scarcity and how these aspects are associated with moral judgment across 67 countries.

4. *The authors conclude “we find evidence for the notion that experiences of relative economic scarcity as a chronic state is not associated with a “depletion” of moral character or less prosocial intentions. Instead, it seems that such experiences of economic scarcity are associated with a stronger preference for identifying and acting as a moral individual, engaging in cooperative behaviours with a clear moral foundation (e.g., helping kin or reciprocating), and having the intention to engage in prosocial charitable giving.” Similarly, in the discussion the authors state: “... individuals who experience relative economic scarcity as a chronic state not only seem more inclined to perceive themselves as moral individuals (i.e., 320 Moral Identity), but also seek to project such morality-related aspects towards their peers and in-group members (i.e., Morality-as-Cooperation, Moral Circle, and Prosocial Intentions).” These conclusions are a bit misleading, because the authors did not measure the actual experience of economic scarcity, but how people subjectively think about their SES, and I think this should be made much more explicit in the discussion of the paper.*

We thank you for highlighting that the conclusions needed further precision. Referring to our reply to your second comment, we have carefully revised the manuscript to clarify that we refer to subjective experiences of economic scarcity as measured by subjective SES at the individual level and income inequality at the macro level. We have also clearly defined in the Introduction what we mean by subjective experiences of economic scarcity and how these experiences differ from objective economic scarcity. Based on your valuable input, we now write the following in the section that you mentioned: “...we find evidence for the notion that subjective experiences of economic scarcity are not associated with a “depletion” of moral character or less prosocial intentions. Instead, it seems that such subjective experiences are associated with a stronger preference for identifying and acting as a moral individual, engaging in cooperative behaviors with a clear moral foundation (e.g., helping kin or reciprocating), and having the intention to engage in prosocial charitable giving.” (p. 19, lines 9-14).

5. *Overall, I think the authors do report a study that adds value to this developing literature. I think the paper should be published, and should be given serious consideration by scholars working on this topic. That said, I question whether the methods used to operationalize the variables in this study provide a strong test of the ideas outlined in the introduction. Furthermore, the authors could draw more attention to the limitations of using subjective SES as an index of chronic resource scarcity. For these reasons, the paper may be more suitable for a specialty journal.*

Again, thank you for your careful re-evaluation of our manuscript. We are happy to learn that you think this study adds value to the rapidly developing literature in this domain. We sincerely hope that our revisions have made it clear how the current operationalizations of our focal variables allow us to test the role that subjective experiences of economic scarcity have in shaping moral judgment and decision-making. As noted above, we have also added limitations associated with the current research, while simultaneously suggesting several fruitful avenues for further studies (pp. 24-25, lines 1-11).

References

- Boon, B., & Farnsworth, J. (2011). Social Exclusion and Poverty: Translating Social Capital into Accessible Resources. *Social Policy & Administration*, 45(5), 507-524.
- Cheung, F., & Lucas, R. E. (2016). Income inequality is associated with stronger social comparison effects: The effect of relative income on life satisfaction. *Journal of Personality and Social Psychology*, 110(2), 332.
- Cundiff, J. M., Smith, T. W., Uchino, B. N., & Berg, C. A. (2013). Subjective Social Status: Construct Validity and Associations with Psychosocial Vulnerability and Self-Rated Health. *International Journal of Behavioral Medicine*, 20(1), 148-158.
- Di Domenico, S. I., & Fournier, M. A. (2014). Socioeconomic Status, Income Inequality, and Health Complaints: A Basic Psychological Needs Perspective. *Social Indicators Research*, 119(3), 1679-1697.
- Galvan, M. J., Payne, K., Hannay, J., Georgeson, A., & Muscatell, K. (2022). What does the MacArthur Scale of Subjective Social Status Measure? Separating Economic Circumstances and Social Status to Predict Health. Preprint at *PsyArxiv*: <https://psyarxiv.com/e9px3/>
- Goldsmith, K., Griskevicius, V., & Hamilton, R. (2020). Scarcity and Consumer Decision Making: Is Scarcity a Mindset, a Threat, a Reference Point, or a Journey? *Journal of the Association for Consumer Research*, 5(4), 358-364.
- Jachimowicz, J. M., Davidai, S., Goya-Tocchetto, D., Szaszi, B., Day, M. V., Tepper, S. J., Phillips, L. T., Mirza, M. U., Ordabayeva, N., & Hauser, O. P. (2022). Inequality in researchers' minds: Four guiding questions for studying subjective perceptions of economic inequality. *Journal of Economic Surveys*.
- Kuper, H., & Marmot, M. (2003). Job Strain, Job Demands, Decision Latitude, and Risk of Coronary Heart Disease Within the Whitehall II Study. *Journal of Epidemiology & Community Health*, 57(2), 147-153.
- Kraus, M. W., Piff, P. K., & Keltner, D. (2009). Social Class, Sense of Control, and Social Explanation. *Journal of Personality and Social Psychology*, 97(6), 992.
- Ren, M., Zou, S., Zhu, S., Shi, M., Li, W., & Ding, D. (2022). Do the poor envy others more? The effects of scarcity mindset on envy. *Current Psychology*, 1-17.
- Sands, M. L., & de Kadt, D. (2020). Local exposure to inequality raises support of people of low wealth for taxing the wealthy. *Nature*. 586(7828), 257-261.
- Singh-Manoux, A., Marmot, M. G., & Adler, N. E. (2005). Does Subjective Social Status Predict Health and Change in Health Status Better than Objective Status? *Psychosomatic Medicine*, 67(6), 855-861.

Sommet, N., Morselli, D., & Spini, D. (2018). Income Inequality Affects the Psychological Health of Only the People Facing Scarcity. *Psychological Science*, 29(12), 1911-1921.

Reviewer 3

1. *I think the authors have done an excellent job of addressing the concerns raised in my original review and that the manuscript has also benefitted from suggestions from the other reviewers.*

I should note, with apologies, that my main concern about the original manuscript was based on a misunderstanding. I thought that the authors were primarily interested in objective economic scarcity. Although they were careful to say “relative” scarcity in many places, much of the literature referenced seems to focus on objective scarcity. This framing led me to believe that objective deprivation was the key variable, despite it’s being accessed through self-report. I now understand that subjective scarcity is the primary target, and I think the revised manuscript makes this more clear. But this makes me wonder whether much of the debate that frames this paper can be resolved by distinguishing between objective and subjective scarcity. I’m not sure, and I leave it to the authors to add something about this if they think it would be illuminating.

Thank you for your careful re-evaluation of our paper. We are very pleased to learn that you think we have done a satisfactory job in addressing the concerns that you and the other two reviewers raised in the first revision. We thank you for your helpful suggestion to carefully distinguish between objective and subjective scarcity. Based on your suggestion, we now use the concepts “subjective experiences of economic scarcity” and “perceptions of economic scarcity” consistently throughout the manuscript, and discuss how such perceptions and experiences differ from objective scarcity circumstances. We have also expanded on this issue in the directions for future research, by urging scholars to disentangle the extent to which subjective and objective measures of economic scarcity are interrelated and predictive of psychological mechanisms across different domains (p. 24, lines 1-11). Again, thank you for your constructive comments.

Reviewers' Comments:

Reviewer #2:

Remarks to the Author:

I appreciate that the authors addressed my previous set of comments. The paper is now more explicit about focusing on subjective perceptions of SES and draws attention to some limitations of this approach.

That said, I was puzzled that the authors still frame GINI (wealth inequality) as an operationalization of country level perceptions of SES. The authors state the "manuscript provides a robust investigation of the association between individual and macro-level perceptions of economic scarcity and how these aspects are associated with moral judgment across 67 countries." This would imply that in societies with higher wealth inequality, people would be making more social comparisons with others who have greater resources and therefore indicate lower subjective SES. I previously asked the authors to report the association between their measure of subjective SES and GINI. I didn't see this statistic reported in the current manuscript or in the response to reviewers. Instead, in the response to reviewers the authors claim, "individual-level measures of the distribution of subjective socioeconomic status are not necessarily highly correlated with the macrolevel measure of GINI (e.g., Di Domenico & Fournier, 2014)." I went to the paper the authors cite in this claim, and this research reports a non-significant positive relation between GINI and subjective SES, and so higher feelings of own SES in areas with higher wealth inequality. This finding is the opposite of what the authors argue here, which is that higher wealth inequality should be associated with lower subjective perceptions of SES.

If the authors are going to consider GINI as an operationalization of subjective perceptions of SES at the country level, I would think it would be beneficial to offer existing evidence to support that claim within their own dataset. Specifically, I continue to encourage the authors to report the association between GINI and subjective perceptions of SES in their own study, and report and discuss the outcome of that analysis in the main text of the paper. At this point, I am still not convinced that wealth inequality causes societal differences in macro-level perceptions of relative economic scarcity, and therefore skeptical that the outcomes of the associations between GINI and the measures of morality can be interpreted as the result of subjective perceptions of relative scarcity.

Reviewer #3:

Remarks to the Author:

As indicated by my second round review, I think this manuscript is excellent and that it has been much improved by the review process.

Revision Notes

We thank the reviewers for the valuable comments to our manuscript. Below are our point-by-point responses to all comments. The comments from the reviewers are numbered and italicized. Our responses appear in regular type.

Reviewer 2

- 1. I appreciate that the authors addressed my previous set of comments. The paper is now more explicit about focusing on subjective perceptions of SES and draws attention to some limitations of this approach.*

Thank you for your careful evaluation of our work. We are happy to learn that you think the manuscript has improved after the last round of revisions. Below, we outline our responses to your additional comments, with direct references to changes in the manuscript.

- 2. That said, I was puzzled that the authors still frame GINI (wealth inequality) as an operationalization of country level perceptions of SES. The authors state the “manuscript provides a robust investigation of the association between individual and macro-level perceptions of economic scarcity and how these aspects are associated with moral judgment across 67 countries.” This would imply that in societies with higher wealth inequality, people would be making more social comparisons with others who have greater resources and therefor indicate lower subjective SES. I previously asked the authors to report the association between their measure of subjective SES and GINI. I didn’t see this statistic reported in the current manuscript or in the response to reviewers. Instead, in the response to reviewers the authors claim, “individual-level measures of the distribution of subjective socioeconomic status are not necessarily highly correlated with the macrolevel measure of GINI (e.g., Di Domenico & Fournier, 2014).” I went to the paper the authors cite in this claim, and this research reports a non-significant positive relation between GINI and subjective SES, and so higher feelings of own SES in areas with higher wealth inequality. This finding is the opposite of what the authors argue here, which is that higher wealth inequality should be associated with lower subjective perceptions of SES. If the authors are going to consider GINI as an operationalization of subjective perceptions of SES at the country level, I would think it would be beneficial to offer existing evidence to support that claim within their own dataset. Specifically, I continue to encourage the authors to report the association between GINI and subjective perceptions of SES in their own study, and report and discuss the outcome of that analysis in the main text of the paper. At this point, I am still not convinced that wealth inequality causes societal differences in macro-level perceptions of relative economic scarcity, and therefore skeptical that the outcomes of the associations between GINI and the measures of morality can be interpreted as the result of subjective perceptions of relative scarcity.*

Thank you for this comment. First, we would like to note that we frame GINI as an operationalization of country-level perceptions of economic scarcity and not country-level perceptions of SES, as you also point out in the referenced quote from our manuscript. We have clarified this on page 7, lines 12-13, and page 8, lines 14-15, in the revised manuscript. Our approach builds on prior work arguing that economic inequality can elicit perceptions of economic scarcity (Buttrick & Oishi, 2017;

Jachimowicz et al., 2020; Kondo et al., 2008; Sommet et al., 2019; Wilkinson & Pickett, 2006). We have also emphasized that further in the revised manuscript (p. 8, lines 13-14).

We have also provided further justifications for using GINI as an indirect measure of macro-level differences in perceptions of relative economic scarcity (p. 8, lines 17-22, and pp. 8-9, lines 22-2), while also acknowledging some limitations of doing so (p. 25, lines 12-21).

Moreover, as you suggested, we have reported the overall correlation between GINI and SES in our data ($r = -0.07$, $P < 0.001$; p. 8, lines 24-25) when discussing the use of GINI as an indirect macro-level indicator of perceptions of economic scarcity. The result affirms what you outlined in your comment, namely that "... in societies with higher wealth inequality, people would be making more social comparisons with others who have greater resources and therefore indicate lower subjective SES". To further corroborate this finding, we added an additional correlational analysis on the relationship between GINI and SES, where we split our measure of self-reported SES based on quantiles (25% = *Lower SES*, Median = *Medium SES*, 75% *Higher SES*), again finding a statistically significant correlation. This analysis affirms that GINI is correlated with differences in self-reported SES (p. 9, lines 1-2).

In summary, we sincerely hope that our revisions have clarified our grounds for using economic inequality as an indirect indicator of macro-level perceptions of relative economic scarcity, despite the discussed limitations of doing so.

References:

- Buttrick, N. R., & Oishi, S. (2017). The psychological consequences of income inequality. *Social and Personality Psychology Compass*, *11*(3), e12304.
- Jachimowicz, J. M., Szaszi, B., Lukas, M., Smerdon, D., Prabhu, J., & Weber, E. U. (2020). Higher economic inequality intensifies the financial hardship of people living in poverty by fraying the community buffer. *Nature Human Behaviour*, *4*(7), 702-712.
- Kondo, N., Kawachi, I., Subramanian, S., Takeda, Y., & Yamagata, Z. (2008). Do social comparisons explain the association between income inequality and health?: Relative deprivation and perceived health among male and female Japanese individuals. *Social Science & Medicine*, *67*(6), 982-987.
- Sommet, N., Elliot, A. J., Jamieson, J. P., & Butera, F. (2019). Income inequality, perceived competitiveness, and approach-avoidance motivation. *Journal of Personality*, *87*(4), 767-784.
- Wilkinson, R. G., & Pickett, K. E. (2006). Income inequality and population health: a review and explanation of the evidence. *Social Science & Medicine*, *62*(7), 1768-1784.

Reviewer 3

- 1. As indicated by my second round review, I think this manuscript is excellent and that it has been much improved by the review process.*

We would like to thank you again for your constructive and careful review of our manuscript. We believe that all your suggestions have strongly improved the current work.

Reviewers' Comments:

Reviewer #2:

Remarks to the Author:

I appreciate how the authors have addressed my previous comments, and I have no further comments on the manuscript.

---

## [Peer Review File · Nature Communications]

Reviewers' Comments:

Reviewer #1:

Remarks to the Author:

The paper reports a really interesting cross-national dataset on the relationships between SES and morality/cooperation. They find mainly negative relationships (higher SES, lower morality), addressing a literature that has previously been very mixed in terms of direction of associations. The dataset is well worth publishing and the findings are useful, but some more work is needed before the paper is publishable or interpretable. Below I list major then minor issues.

1. Analysis. For SES, the authors need to decompose the associations into their within-country and between-country components., using the method of Van der Pol and Wright (*Animal Behaviour* 77(3):753-758). Given that countries differ in mean SES, the overall findings could be explained by (a) within countries, individuals with higher SES have lower morality; or (b) countries with higher mean SES contain individuals with lower morality; or some combination of the two. The within- and between-country associations can even be in opposite directions (looking at figure 2c and table 1 that looks like it could be true here for moral circle). Anyway, neither the variance partition analysis reported at 144 nor including a random effect of country achieves this, but it is essential to interpret what is going on.

2. Visualization. Figure 2 is not the right visualization for the SES analyses, exactly because it plots a between-country association for results that appear to be driven by the within-country associations. Figure 2c even shows a trend in the opposite direction to the main finding. Instead, I suggest plots with 68 regression lines: 67 faint ones, one per country, and then the overall model to show the central tendency. There would be too many lines to colour code by country, but the reader would at least be able to see whether most of them had similar slopes or not. In fact, it would be great because the reader would be able to see BOTH the main pattern (most lines slope down) and the heterogeneity (a few go the opposite way). It is really easy to make these plots in R (`ggplot2::geom_smooth(method="lm", aes(group=factor(country)))`), or whatever. BTW, another useful thing I thought of was a table saying, for each DV, how many countries showed a significant positive, significant negative, or non-significant association for SES. This information is in the supplement but could be summarised in the main paper (around line 160 I thought of this).

3. Framing. The authors have an unfortunate tendency to claim that poverty 'distorts' or 'impedes' cognition (e.g. 44, 188). This is contentious and potentially stigmatizing. It is more balanced to say that poverty 'shifts' cognition. People in poverty may be better at some kinds of cognition. Moreover, discounting the future and other 'impairments' could be perfectly rational in an uncertain environment. So I would urge a more neutral framing here (see <https://doi.org/10.1177/0963721419881154> or <https://doi.org/10.1017/S0140525X1600234X>). It would actually fit better with the authors' argument, which is that people in poverty shift to greater moral interdependence as a strategy to cope with their environments.

4. Interpretation. I have never been a big fan of the 'are poorer people more or less prosocial' debate. This is because the answer seems to me obvious: both are somewhat prosocial, but in different ways. Therefore, the direction of result you get is going to be totally dependent on the measure of prosociality (or morality) that you choose (who is the prosocial interaction with, what is the resource, etc.). I think the authors should bear this in mind more in their discussion. Both sides of the debate are probably right, given that poor people do more of some kinds of morality and less of others. This study, big as it is, does not settle the matter because it has only limited measures of morality/prosociality. Other measures might have produced opposite associations.

5. External validity. At a couple of places, the authors confuse external validity with generalizability (e.g. 88, 257). These are not the same thing at all. I would say the present study has high generalizability, because of the large sample across many countries, but very low external validity. External validity is to do with the mapping of your measures onto real-world behaviour. These rather abstract short self-report measures probably have very low external validity in this sense (if you interpret them as proxies for moral or prosocial behaviour). A behavioural economist would dismiss these findings as interesting because all the measures are hypothetical 'cheap talk'. They probably say nothing about how those people would allocate actual time or resources. This

needs to be stated as a limitation. Relatedly, the authors should not claim to have measured 'prosocial behaviour' (e.g 32, 196). No behaviour was measured in this study. 'Hypothetical prosocial intentions' is quite another thing.

Minor points

Line 67. 'their immediate environment' is unclear; presumably this means 'their social relationships'

Sentences beginning at line 72 are poorly written and ungrammatical

109. What the four DVs are needs to be explained, either in the Introduction or when they are introduced in the results. At present, results come without any prior explanation of what the measures are. This can often be a problem in formats where the methods section comes at the end. A brief sentence or two introducing the measures (both the DVs and the predictors) is therefore needed in the introduction or in the results text.

116. If the alternative analyses use imputation, how do the main models treat missing data?

Table 1: For Gender, give the ref category to make the direction of effect interpretable

Reviewer #2:

Remarks to the Author:

This paper reports the results of a survey conducted across many countries that measures subjective SES and then correlates this with measures of moral identity, morality as cooperation, moral circle, and prosocial behavior. The authors use these data to test pre-registered predictions that higher subjective SES would be associated with higher moral character and behavior. However, they found the opposite to those predictions. They found people reporting lower subjective SES reported higher moral character and behaviors. I have some concerns about the sampling strategy and measures used in the study that may limit the study's ability to test hypotheses about the relation between "chronic resource scarcity" and "Moral Character and Behavior" across countries.

I am concerned about the sampling strategy used in this study, and how it could affect the author's ability to test the hypotheses across the countries. It sounds like the people in the different countries collecting data could collect data in any way that they wanted, but were requested to try and get a representative sample for age and gender (which many countries were unable to do). I would think it would be important to get representativeness for household income to provide a good test of this hypothesis across societies. I am assuming that many of the people collecting data were other social scientists and that if they were to recruit participants from their personal network (or University), then this would bias the samples to have a higher SES. This would be a serious limitation of the study, especially for being able to test for any differences between the countries. The differences between the countries could be due to the different sampling strategies used in each country. Also, this sample strategy could result in a restriction of range of SES, and not sample many people who have "chronic resource scarcity".

I also have some concerns about the measures. Specifically, I am concerned that there is not equivalence of the factor structure of the scales across all the societies. This would affect how we can interpret the correlations across societies. I would suggest the authors report a structural equivalence test across the societies for moral identity and morality as cooperation. Furthermore, the authors only include a single item measuring "moral circle". They also do not report evidence for the validity of this item. In fact, it sounds like this is the first time this item has been used in prior research. This severely limits the value of this item to be used to test theory. There is some data in the present study that could be used as a validity test: Does the measure of moral circle correlate with a relatively higher amount donated to an international, versus national, charity? The authors operationalize "chronic resource scarcity" with a measure of subjective SES. I would have preferred a more objective measure of chronic resource scarcity, such as household income. The paper could benefit from at least reporting the relation between subjective SES and other

measures of resource scarcity in prior work. Currently the paper just says this is a common way of measuring SES, while there are also several other methods used in the literature.

Another concern I have is that with such a large dataset, everything is statistically significant. For example, the Beta value for SES and moral circle is tiny. Is it even worth discussing this result as supporting a theoretical framework? How many participants would be required to detect the effect size in a subsequent study? I think it would be useful for a paper like this to identify an effect size that they would suggest is reasonable to interpret. I agree with the authors that small effect sizes can still be valuable, but there must be a point when we think its just not important, or it is so trivial that it is not really supporting a prediction from a theoretical framework.

In figure 2b, it looks like there is no relation between SES and morality as cooperation, but then there is a negative relation between SES and moral circle. However, in Table 1 the beta value is larger for morality as cooperation, compared to a moral circle. Why is there a discrepancy between Table 1 and Figure 2?

When I was reading the paper, I did not know the definition of many concepts when they were first raised in the paper, such as moral identity, moral circle, and morality-as-cooperation. I also don't understand the definition of morality that is used in this work. Of course, prosocial behavior (i.e., behaviors that benefit others) can also be immoral, such as helping a prisoner escape from prison. And people can cooperate together to do some very horrible, immoral behaviors. In general, I thought the paper could be improved substantially in terms of communicating about the concepts discussed in the paper. For example, a major conclusion of the paper is that: "chronic resource scarcity...results in more moral outcomes", and I was left confused about what "moral outcomes" even mean in this context.

Reviewer #3:

Remarks to the Author:

In this paper the authors report on the relationship between measures inequality (Subjective personal SES, National GINI coefficient) and various measures of "morality" (moral identity, morality-as-cooperation, size of moral circle, and [hypothetical] prosocial behaviour).

This paper aims to address a very interesting question that has been hotly debated: Does higher economic status affect one's moral values and behavior? As the authors explain in a clearly written introduction, the literature on this is mixed (although there are factors that may explain the apparent contradictions in the literature). The present study aims to address this question using a massive data set (over 46,000 participants from 67 nations). They report that economic deprivation is, in various ways, associated with more "moral" outcomes.

This study has some notable strengths, such as the aforementioned large international sample as well as some sophisticated statistical features. However, it seems to me that it is flawed in some very basic ways. It's not clear to me whether these problems are correctable. It seems that they should be, but at the same time, the fact that the authors didn't address these problems already makes me worry that the fixing the problems leads to a very different set of conclusions.

The main problem is that authors use measures of SES/inequality without taking into account absolute wealth—both at the individual and national level. The SES measure asks for a subjective judgment of where the respondent falls on an SES ladder. As one might expect, people's responses are heavily influenced by the local context. As an illustration, the mean SES rating on this measure is *lower* in wealthy Denmark than in poor Nigeria. It doesn't follow from this that this measure is meaningless. It tells us that they average Dane sees themselves as lower on the *local* SES ladder than the average Nigerian does. But this is not a meaningful measure of actual SES, deprivation, or anything of that sort.

The same holds for The GINI coefficient, the other major predictor variable. Germany and Bangladesh have similar GINI coefficients, but obviously the experience of inequality and deprivation are very different for the average German as compared to the average Bangladeshi.

The way to deal with this is obvious: One must (at the very least) account for wealth at the individual and national levels in when using measures of perceived relative SES and (objective)

national-level inequality. This is so obvious, I was surprised that the authors didn't do this. And I was especially surprised given that the analysis plan was pre-registered. So, I looked at the pre-registration and, sure enough, the authors specify household income as a key predictor variable. But this variable is not used at all in the paper.

I also have concerns about the key outcome measures related to morality. In the introduction it sounds as if the authors are weighing in on the debate about whether higher SES makes people more or less "moral", meaning more or less inclined to do things that are widely regarded as morally good (e.g. being generous) or morally bad (e.g. cheating). But only the generosity measure falls into this category. And here the authors rely on self-report of generosity, which could vary enormously by culture and/or be generally unreliable. At the very least, one would want to corroborate these subjective self-reports of generosity with country-level data on things like actual charitable giving. And for the other measures, it's not clear whether they exhibit the same kind of local/cultural relativity that the SES ladder measures do. For example, what counts as having a large vs. small moral circle in Nigeria vs. the US?

Finally, it's not clear whether the preregistration is really a preregistration. The authors explain that "some" of the data have already been collected and then refer to this explanation: "The data has already been collected and is a part of an international project on moral psychology during the corona (COVID-19) pandemic. The data will initially be used in the paper for which it was collected."

This doesn't tell us which data were collected prior to the pre-registration and which were not. And, most importantly, it doesn't say whether the authors were aware of the outcomes of these analyses prior to the pre-registration. Given that the analyses reported on in the paper differ from the analyses outlined in the pre-registration, it seems that they didn't know the outcomes in advance, which is good—except that we only know this because the authors deviated so significantly from their pre-registered plan.

This is a rich data set, and it seems possible to me that the authors could analyze it in a way that makes more sense. However, the fact that the authors pre-registered a more sensible approach and then abandoned it suggests that this would lead to rather different conclusions. Likewise, it's possible that the data could be augmented with nation-level objective measures that would allow them to more directly address the questions they seem to want to answer in the introduction. But, in its present form, this paper has some very significant limitations.

Revision Notes

Reviewer 1:

1. *The paper reports a really interesting cross-national dataset on the relationships between SES and morality/cooperation. They find mainly negative relationships (higher SES, lower morality), addressing a literature that has previously been very mixed in terms of direction of associations. The dataset is well worth publishing and the findings are useful, but some more work is needed before the paper is publishable or interpretable. Below I list major then minor issues.*

Thank you very much for your constructive review of our paper. We are pleased to hear that you find our research and the data to be interesting, useful, and worth publishing. Below you will find our answers to your comments, with direct references to the revised manuscript.

2. *1. Analysis. For SES, the authors need to decompose the associations into their within-country and between-country components., using the method of Van der Pol and Wright (Animal Behaviour 77(3):753-758). Given that countries differ in mean SES, the overall findings could be explained by (a) within countries, individuals with higher SES have lower morality; or (b) countries with higher mean SES contain individuals with lower morality; or some combination of the two. The within- and between-country associations can even be in opposite directions (looking at figure 2c and table 1 that looks like it could be true here for moral circle). Anyway, nether the variance partition analysis reported at 144 nor including a random effect of country achieves this, but it is essential to interpret what is going on.*

Thank you for this detailed and valuable suggestion. We have now implemented the suggested analyses to complement our original analytic approach. We find that our effects are primarily explained by within-country differences, such that within countries, individuals with higher SES seem to have lower Moral Identity, lower Morality-as-Cooperation, and less Prosocial Intentions. Yet, for the measure of Moral Circle, we find that the within-country component of subjective SES was very weakly associated with a larger Moral Circle and that the relationship between subjective SES and the size of one's Moral Circle is primarily explained by between-country differences (pp. 15-16, line 21-5; and Supplementary Tables S7, S8, S9 and S10). We have revised the analyses and the interpretation of them to reflect these new findings.

3. *2. Visualization. Figure 2 is not the right visualization for the SES analyses, exactly because it plots a between-country association for results that appear to be driven by the within-country associations. Figure 2c even shows a trend in the opposite direction to the main finding. Instead, I suggest plots with 68 regression lines: 67 faint ones, one per country, and then the overall model to show the central tendency. There would be too many lines to colour code by country, but the reader would at least be able to see whether most of them had similar slopes or not. In fact, it would be great because the reader would be able to see BOTH the main pattern (most lines slope down) and the heterogeneity (a few go the opposite way). It is really easy to make these plots in R (ggplot2::geom_smooth(method="lm", aes(group=factor(country))), or whatever). BTW, another useful thing I thought of was a table saying, for each DV, how many countries showed a significant positive, significant negative, or non-significant*

association for SES. This information is in the supplement but could be summarised in the main paper (around line 160 I thought of this).

We thank you for these helpful suggestions. As requested, we have implemented these plots, which give a much clearer illustration of the central tendency in the relationship between subjective SES and each of our measures of morality as well as the variation around the central tendencies (see Figure 2, p. 12). Furthermore, we have also included a table that summarizes the number of countries showing a significant positive, significant negative, or non-significant association for SES (Table 2, p. 13). An even more fine-grained overview is available in the Supplementary Materials.

4. *3. Framing. The authors have an unfortunate tendency to claim that poverty 'distorts' or 'impedes' cognition (e.g. 44, 188). This is contentious and potentially stigmatizing. It is more balanced to say that poverty 'shifts' cognition. People in poverty may be better at some kinds of cognition. Moreover, discounting the future and other 'impairments' could be perfectly rational in an uncertain environment. So I would urge a more neutral framing here (see <https://doi.org/10.1177/0963721419881154> or <https://doi.org/10.1017/S0140525X1600234X>). It would actually fit better with the authors' argument, which is that people in poverty shift to greater moral interdependence as a strategy to cope with their environments.*

Thank you for this important comment. The use of the more negatively-laden framing was adopted from seminal research in the field (see e.g., Mani et. al, 2013 in *Science: Poverty impedes cognitive functioning*). Yet, based on your comment, we fully agree that this framing can be potentially stigmatizing to individuals with lower levels of resources, which goes directly against one of the main points of our research. Consequently, we have changed the framing throughout the manuscript to be more neutral in language. For example, we now describe that chronic resource scarcity can shift cognitive focus and attention (e.g., p. 2, line 5, p 3., lines 3-6). We believe that this new framing is more balanced and we hope that you agree.

5. *4. Interpretation. I have never been a big fan of the 'are poorer people more or less prosocial' debate. This is because the answer seems to me obvious: both are somewhat prosocial, but in different ways. Therefore, the direction of result you get is going to be totally dependent on the measure of prosociality (or morality) that you choose (who is the prosocial interaction with, what is the resource, etc.). I think the authors should bear this in mind more in their discussion. Both sides of the debate are probably right, given that poor people do more of some kinds of morality and less of others. This study, big as it is, does not settle the matter because it has only limited measures of morality/prosociality. Other measures might have produced opposite associations.*

Thank you for raising this point. We agree that our study only measures one specific form of prosociality in the form of donation intentions towards charity organizations. Yet, we argue that by including our additional measures of moral character and cooperation, our study offers a nuanced view on whether subjective economic scarcity is related to morality. In the manuscript, we have now added a clear conceptualization of moral decision-making as a multidimensional term (p. 3, line 11-17) as well as we have strengthened our argument for the value of including four different indicators of morality (p. 5, line 12-20, pp. 6-7, line 11-2). Moreover, we have expanded our discussion by emphasizing that although the current study expands the current state-of-

the-art on how experiences of relative economic scarcity influence moral decision-making through the four well-validated morality measures that we selected, other measures might have been relevant to include as well (p. 21, line 12-22). We acknowledge that our investigation does not aim to conclude whether low (vs. high) SES makes people more or less moral *in general*. Thus, other types of morality measures might well be more (or less) pronounced in individuals with more abundant resources, depending on the precise context and the specific type of morality measures used. Finally, we also discuss what other types of prosocial behavior would be interesting to explore and how such behavior might differ depending on the demographic profile of participants.

6. *5. External validity. At a couple of places, the authors confuse external validity with generalizability (e.g. 88, 257). These are not the same thing at all. I would say the present study has high generalizability, because of the large sample across many countries, but very low external validity. External validity is to do with the mapping of your measures onto real-world behaviour. These rather abstract short self-report measures probably have very low external validity in this sense (if you interpret them as proxies for moral or prosocial behaviour). A behavioural economist would dismiss these findings as interesting because all the measures are hypothetical 'cheap talk'. They probably say nothing about how those people would allocate actual time or resources. This needs to be stated as a limitation. Relatedly, the authors should not claim to have measured 'prosocial behaviour' (e.g. 32, 196). No behaviour was measured in this study. 'Hypothetical prosocial intentions' is quite another thing.*

Thank you for this valuable comment. We have rewritten the parts about external validity and added supporting references to clarify that we are referring to cross-cultural generalizability (pp. 5-6, line 21-10). Additionally, we have also clarified that we are studying prosocial intentions and self-reported moral constructs (e.g., pp. 6, line 11-2). Lastly, we also discuss our use of self-report measures as a limitation in the discussion and provide recommendations for future research (pp. 21-22, line 23-9). We believe these revisions mitigated the confusion between external validity and generalizability and we hope that our discussed limitations related to relying on self-reported dependent measures of morality are now more saliently stated.

7. *Minor points.
Line 67. 'their immediate environment' is unclear; presumably this means 'their social relationships'*

We have changed this sentence in accordance with your suggestion.

8. *Sentences beginning at line 72 are poorly written and ungrammatical*

Thank you for pointing this out. We have rewritten these sentences to improve language.

9. *109. What the four DVs are needs to be explained, either in the Introduction or when they are introduced in the results. At present, results come without any prior explanation of what the measures are. This can often be a problem in formats where the methods section comes at the end. A brief sentence or two introducing the measures*

(both the DVs and the predictors) is therefore needed in the introduction or in the results text.

We thank you for highlighting that the DVs needed further explanation. We have added a clear description of our four DVs and our primary predictors in the main text (pp. 6-8, line 11-4). Additionally, we have expanded the description of them in the Methods section (pp. 25-28, line 5-3).

10. 116. *If the alternative analyses use imputation, how do the main models treat missing data?*

The linear mixed effects models reported in manuscript use the Restricted Maximum Likelihood algorithm, which relies on pair-wise deletion before the maximum likelihood algorithm is applied. As we have a very large dataset and a relatively small amount of missing data, we opted for this algorithm based on the notion that we would not lose a lot of statistical power in these calculations. However, to demonstrate robustness of our findings, we also imputed missing data and we now clearly report the results of these models in the Supplementary Materials. To further enhance transparency, we have added additional information to the Methods section that clarifies these analytic decisions (pp. 28-29, lines 25-4).

11. *Table 1: For Gender, give the ref category to make the direction of effect interpretable*

Thank you noting this. We have now added the reference category to Table 1.

Reviewer 2:

1. *This paper reports the results of a survey conducted across many countries that measures subjective SES and then correlates this with measures of moral identity, morality as cooperation, moral circle, and prosocial behavior. The authors use these data to test pre-registered predictions that higher subjective SES would be associated with higher moral character and behavior. However, they found the opposite to those predictions. They found people reporting lower subjective SES reported higher moral character and behaviors. I have some concerns about the sampling strategy and measures used in the study that may limit the study's ability to test hypotheses about the relation between "chronic resource scarcity" and "Moral Character and Behavior" across countries. I am concerned about the sampling strategy used in this study, and how it could affect the author's ability to test the hypotheses across the countries. It sounds like the people in the different countries collecting data could collect data in any way that they wanted, but were requested to try and get a representative sample for age and gender (which many countries were unable to do). I would think it would be important to get representativeness for household income to provide a good test of this hypothesis across societies. I am assuming that many of the people collecting data were other social scientists and that if they were to recruit participants from their personal network (or University), then this would bias the samples to have a higher SES. This would be a serious limitation of the study, especially for being able to test for any differences between the countries. The differences between the countries could be due to the different sampling strategies used in each country. Also, this sample strategy could result in a restriction of range of SES, and not sample may people who have "chronic resource scarcity".*

We thank you for your careful review of our paper and the many constructive suggestions for improvements.

We now provide a clearer description of our sampling strategy in the Methods section of the manuscript (pp. 23-24, line 20-22). Specifically, no samples were recruited through close personal networks of the scholars and no student samples were used. The samples were collected as part of a large-scale international project with more than 200 researchers across the world (see van Bavel et al. 2022 in *Nature Communications*). All samples were recruited by online platforms or panel agencies, with participating researchers asked to fund their own data collection and collect samples which were nationally representative in respect to age, gender, and ethnicity. In total, data from 28 countries were nationally representative on these grounds, while the remaining data from 39 countries were based on convenience samples. In this revision, we have added information regarding which specific samples were representative, and which were not. Consequently, we have added robustness tests focusing on the samples that are nationally representative, both in the main text (Table 1, p. 10) and in the Supplementary Materials (Table S6). These results indicate that our main individual-level and country-level results largely replicate when focusing only on the representative samples. A similar relationship between strength of point estimates and type of sample was found in the publication first using this dataset, albeit with a completely different set of dependent and independent variables (van Bavel, et al. 2022). Importantly, our robustness tests support the notion that the between-country findings are not just a product of differences the sampling strategy used.

Lastly, to answer to your concern about the range of the SES measure, we provide an overview of the range of this measure in two tables with summary statistics in our Supplementary Materials (Table S28 for all 67 countries, Table S29 for nationally representative countries). These distributions clearly show that there is no restriction in our range of the SES measure in general, across the countries.

2. *I also have some concerns about the measures. Specifically, I am concerned that there is not equivalence of the factor structure of the scales across all the societies. This would affect how we can interpret the correlations across societies. I would suggest the authors report a structural equivalence test across the societies for moral identity and morality as cooperation.*

Thank you for this important comment. To test the equivalence of the factor structure across the 67 countries, we used multilevel confirmatory factor alignment, as suggested by Asparouhov and Muthén (2014), for the two multi-item scales in our analyzed data: Moral Identity and Morality-as-Cooperation.

For Moral Identity, the results showed that for factor loadings the scale exhibited 9% of non-invariance for item parameters and 8.2% for intercepts. Hence, following Asparouhov and Muthén's (2014) suggestion of a cut-off value of 25% in order to consider a scale non-invariant, we deem the scale suitable for use in the current investigation.

For Morality-as-Cooperation, we had similar results albeit with slightly higher levels of non-invariance. Here, the results show that for factor loadings the scale exhibited 9.8% of non-invariance for item parameters and 14.3% for intercepts. Again, following

the suggestions by Asparouhov and Muthén (2014), we deem the scale adequate for use.

Of note, our decision to use these scales as reliable measures of Moral Identity and Morality-as-Cooperation aligns with recent investigations using the same scales in similar multi-national collaborations (see Pavlovic et al. 2022, in press).

We report the overall results and the specifics of this multilevel confirmatory factor analysis alignment in the Methods section (pp. 25-27, line 5-2) and the full results of the analysis in the Supplementary Materials (Tables S17-S27).

Naturally, some degree of non-invariance in scales is almost unavoidable when dealing with a dataset of this magnitude, as we incorporate data from many different cultures and societies. However, based on the results from the multilevel confirmatory factor alignment as well as recent literature which has tested these scales across countries (see e.g., Curry et al., 2019 testing the Morality-as-Cooperation across 60 societies), we argue that the interpretation of our results, as currently reported in the manuscript, is not severely affected by the factor structure of the scales of Moral Identity and Morality-as-Cooperation.

3. *Furthermore, the authors only include a single item measuring “moral circle”. They also do not report evidence for the validity of this item. In fact, it sounds like this is the first time this item has been used in prior research. This severely limits the value of this item to be used to test theory. There is some data in the present study that could be used as a validity test: Does the measure of moral circle correlate with a relatively higher amount donated to an international, versus national, charity?*

Thank you for this comment. We have outlined the validity and the prior use of the moral circle framework in the Methods section (see p. 27, lines 3-9). While this measure is a single-item scale, it has been validated, studied and used extensively (see e.g., Waytz et al. 2019 in *Nature Communications* or Graham, et al. 2017 in *Cognition*). Moreover, we have added further argumentation for the importance of this construct in the main body of the manuscript (see p. 6, line 18-20; p. 20, line 1-13). We believe these revisions highlighting the content of the Moral Circle measure and its use in prior research strengthens the justification for including it in the present investigation.

To further situate this measure in relation to the other measures of morality, we report the correlation between Moral Circle and our measure of Prosocial Intentions in the Supplementary Materials along with our other measures of morality (Table S3 and S4). The results show that moral circle has a significant negative correlation with intention to donate to a national charity ($r = -0.12, P < .001$), while it has a significant positive correlation with the intention to donate to an international charity ($r = 0.08, P < .001$). When interpreting these correlations, it is important to note that Moral Circle measures the size of the “circle” of people for which individuals are concerned whether right or wrong is done towards them, which should not necessarily posit a strong correlation with prosocial intentions towards national and international charities (Crimston et al., 2018). For instance, one could imagine that while a participant might report to have a small Moral Circle (i.e., only be concerned about whether right or wrong is done towards close family and friends), such an individual could still be able to recognize the morality of helping others in need. In other words, the low correlations between

Prosocial Intentions and More Circle are to be expected because they capture different dimensions of moral decision-making (p. 3, line 11-17; p. 12, line 12-20).

4. *The authors operationalize “chronic resource scarcity” with a measure of subjective SES. I would have preferred a more objective measure of chronic resource scarcity, such as household income. The paper could benefit from at least reporting the relation between subjective SES and other measures of resource scarcity in prior work. Currently the paper just says this is a common way of measuring SES, while there are also several other methods used in the literature.*

Based on scarcity theory (de Bruijn & Antonides, 2021; Cannon et al. 2019; Shah et al. 2012), we argue that our measure of subjective socioeconomic status is actually the strong point of the current investigation, because we investigate specifically how the psychological experiences of relative deprivation is associated with moral judgment and decision-making. Hence, we identify that it is possible for deprived individuals living in very rich nations (e.g., Denmark) to “feel relatively poorer” compared to deprived individuals in objectively poorer nations (e.g., Nigeria, cautiously using the same example as that outlined by Reviewer 3). Consequently, the use of this measure is in line with recent work which has argued for the importance of investigating how subjective experiences of economic scarcity affects decision-making, while strongly advocating for more research on this subject (see de Bruijn & Antonides, 2021 for a review). Thus, although subjective and objective measures of SES are moderately correlated (Tan et al. 2020), the focus of the current paper is to investigate whether subjective experiences of economic scarcity is associated with moral decision-making.

Yet, to further strengthen our conclusions, we have augmented our data with a more objective measure of resource scarcity in the form of adjusted net-income per capita from the World Bank, to identify how this might be associated with morality as well. Here, we find that our objective measure of scarcity go in the same direction as the subjective one (see Supplementary Materials Table S11). However, because the GINI index is highly correlated with this measure of adjusted net-income per capita ($r = -0.44$, $P < .001$), this measure captures the variance attributed to the GINI index in our main models. Consequently, we included the models using the macro-level objective measure of adjusted net-income in our robustness models.

To enhance readability, we now underline in the introduction why we focus on subjective economic scarcity rather than more objective measures of scarcity in our study. To this end, we have added theoretical arguments based on the extensive literature on this subject (see pp. 7-8, line 3-4). Lastly, regarding the measurement of SES, we completely agree that existing research is scattered with different measures of this construct – a problem which has also received considerable attention (see e.g., Cirino et al., 2002; Diemer et al., 2013). However, these differences in measurement usually concern the measure of *objective* socioeconomic status (e.g., differences in weighting of education, occupation, and household income in an aggregated measure). In contrast, the measurement of subjective SES – as used in the current investigation (the McArthur SES ladder) – is one of the most widely adopted items to capture this construct and has previously been shown to exhibit sufficient reliability across different cultures (see e.g., Giatti, et al. 2012 in *BMC Public Health*).

5. *Another concern I have is that with such a large dataset, everything is statistically significant. For example, the Beta value for SES and moral circle is tiny. Is it even worth discussing this result as supporting a theoretical framework? How many participants would be required to detect the effect size in a subsequent study? I think it would be useful for a paper like this to identify an effect size that they would suggest is reasonable to interpret. I agree with the authors that small effect sizes can still be valuable, but there must be a point when we think its just not important, or it is so trivial that it is not really supporting a prediction from a theoretical framework.*

Thank you for this comment. We agree that because of the size of the dataset (considering the law-of-large-numbers in probability theory) interpreting statistical significance is not as meaningful as in studies with smaller sample sizes. This is also the reason as for why we do not center the manuscript around the significance level of our models, but merely around the size of the standardized beta values, which allow for easy comparison between models and mirror the presentation format in previous investigations using large datasets (see e.g., Götz et al., 2020 in *Nature Human Behaviour*). We argue that utilizing this rich dataset to uncover differences in psychological constructs linked to morality, even if such effects might be small by conventional standards used primarily in experimental research, is a strong point of the current investigation, as “*complex psychological phenomena [...] are likely to be influenced by hundreds, if not thousands, of factors [...], so small effects are to be expected especially when examined in the uncontrolled context of real-world settings [...]*” (Götz et al., 2020 p. 1140). Therefore, we decided to keep the measure of Moral Circle in our analyses, considering that the association for this measure is in the opposite direction of our theoretically informed pre-registered hypothesis, but also because it measures a distinct dimension of moral decision-making. (p. 6, line 18-20). Having noted that, we agree that the size of the “Moral Circle” effect is indeed *very* small (pp. 15-16, line 21-5). Consequently, to address your concerns, we have changed the discussion of this result regarding its implications to articulate those in a more tentative manner (p. 20, line 1-13). We sincerely hope this will suffice.

6. *In figure 2b, it looks like there is no relation between SES and morality as cooperation, but then there is a negative relation between SES and moral circle. However, in Table 1 the beta value is larger for morality as cooperation, compared to a moral circle. Why is there a discrepancy between Table 1 and Figure 2?*

We thank you for raising this point, which reflects a mistake in our initial submission. The table has now been corrected. Furthermore, following the recommendation by Reviewer 1, we have changed the visualization of the mentioned relationship to more clearly show both the within-and between-country variation in our results (see Figure 2, p. 12).

7. *When I was reading the paper, I did not know the definition of many concepts when they were first raised in the paper, such as moral identity, moral circle, and morality-as-cooperation. I also don't understand the definition of morality that is used in this work. Of course, prosocial behavior (i.e., behaviors that benefit others) can also be immoral, such as helping a prisoner escape from prison. And people can cooperate together to do some very horrible, immoral behaviors. In general, I thought the paper could be improved substantially in terms of communicating about the concepts discussed in the paper. For example, a major conclusion of the paper is that: “chronic*

resource scarcity...results in more moral outcomes”, and I was left confused about what “moral outcomes” even mean in this context.

Thank you for this comment, which aligns with one of the comments from Reviewer 1. In the revised manuscript, we have added a clear definition of our constructs in the main body of the manuscript and added supporting information of the origin of these constructs (p. 6, line 11-22) Furthermore, we have tried our best to enhance the communication of the main terms discussed in our paper by clearly conceptualizing moral decision-making and the multidimensional nature of morality (p. 3, line 11-17; p. 5, line 14-20). Additionally, we have streamlined and simplified the terminology to systematically refer to our four dependent variables as measures of moral decision-making rather than using terms such as “moral outcomes”. We have also altered the major conclusion that you outline to describe our results more precisely (instead of referring to “moral outcomes”; p. 17, line 3-12). Finally, we have a clear description of our measures in the method section of the paper (pp. 25-28, lines 5-3).

Reviewer 3:

- 1. In this paper the authors report on the relationship between measures inequality (Subjective personal SES, National GINI coefficient) and various measures of “morality” (moral identity, morality-as-cooperation, size of moral circle, and [hypothetical] prosocial behaviour). This paper aims to address a very interesting question that has been hotly debated: Does higher economic status affect one’s moral values and behavior? As the authors explain in a clearly written introduction, the literature on this is mixed (although there are factors that may explain the apparent contradictions in the literature). The present study aims to address this question using a massive data set (over 46,000 participants from 67 nations). They report that economic deprivation is, in various ways, associated with more “moral” outcomes. This study has some notable strengths, such as the aforementioned large international sample as well as some sophisticated statistical features. However, it seems to me that it is flawed in some very basic ways. It’s not clear to me whether these problems are correctable. It seems that they should be, but at the same time, the fact that the authors didn’t address these problems already makes me worry that the fixing the problems leads to a very different set of conclusions.*

Thank you for your careful reading of our paper and for your very constructive comments.

- 2. The main problem is that authors use measures of SES/inequality without taking into account absolute wealth—both at the individual and national level. The SES measure asks for a subjective judgment of where the respondent falls on an SES ladder. As one might expect, people’s responses are heavily influenced by the local context. As an illustration, the mean SES rating on this measure is *lower* in wealthy Denmark than in poor Nigeria. It doesn’t follow from this that this measure is meaningless. It tells us that they average Dane sees themselves as lower on the *local* SES ladder than the average Nigerian does. But this is not a meaningful measure of actual SES, deprivation, or anything of that sort. The same holds for The GINI coefficient, the other major predictor variable. Germany and Bangladesh have similar GINI coefficients, but obviously the experience of inequality and deprivation are very different for the average German as compared to the average Bangladeshi. The way to deal with this is obvious: One must (at the very least) account for wealth at the individual and national levels in*

when using measures of perceived relative SES and (objective) national-level inequality. This is so obvious, I was surprised that the authors didn't do this. And I was especially surprised given that the analysis plan was pre-registered. So, I looked at the pre-registration and, sure enough, the authors specify household income as a key predictor variable. But this variable is not used at all in the paper.

Thank you for this comment, which partly aligns with that of Reviewer 2. Based on scarcity theory (de Bruijn & Antonides, 2021; Cannon et al. 2019; Shah et al. 2012), we argue that our measure of subjective socioeconomic status is actually the strong point of the current investigation, because we investigate specifically how subjective, as opposed to objective, experiences of economic scarcity is associated with morality. Hence, we identify that it is possible for deprived individuals living in very rich nations (e.g., Denmark) to “feel relatively poorer” compared to deprived individuals in objectively poorer nations (e.g., Nigeria, cautiously using the same example as yourself). (See Supplementary Table S28 and S29 for an overview of the range the subjective SES measure and country-level GINI coefficients).

To add further robustness to our results, we have augmented our data with a more objective measure of economic scarcity in the form of adjusted disposable net-income per capita from the World Bank, to identify how this might be associated with morality as well. Here we find that our objective measure of scarcity go in the same direction as the subjective one (Supplementary Table S11). However, because the GINI index is highly correlated with this measure of national net income per capita ($r = -0.44$, $P < .001$), this measure captures the variance attributed to the GINI index in our main models. Consequently, we included the models using the macro-level objective measure of adjusted disposable net-income in our robustness models.

We have now elaborated more on the importance of studying subjective economic scarcity in the introduction of the manuscript, while also adding theoretical arguments based on the extensive literature on this subject (p. 3, line 1-10; pp. 7-8, lines 3-4).

Concerning our specific measure of subjective socioeconomic status, the McArthur ladder scale, we argue that the use of this scale is actually a strong point of the current investigation. The scale has been used extensively in previous literature to capture *subjective* socioeconomic status and has been shown to exhibit sufficient reliability across different cultures (see e.g., Giatti, et al. 2012 in *BMC Public Health*). Because this scale is able to capture contextual differences in income, education, and job status regardless of more objective measures of socioeconomic status, we argue that it fulfills the purpose of identifying directly how individuals *perceive* their own socioeconomic status *across* societies.

With respect to the macro-level measure of economic inequality, the GINI index, we recognize that experiences of economic inequality might differ across cultures. However, the importance of including this measure follows previous investigations (e.g., Pickett & Wilkinson, 2015; Wilkinson & Pickett, 2011), which have provided evidence for the importance of considering national level economic inequality instead of focusing on objective measures of household income. That is, prior research has shown that societal problems such as crime rates, mental health problems, etc. have no correlation with objective measures of household income, while being strongly correlated with national measures of economic inequality. Therefore, we argue that our

use of the individual level subjective SES complemented by the national level of income inequality achieves the goal of identifying how individuals who perceive themselves to be living with less resources and in countries characterized by economic inequality act in regards to questions concerning moral thought and intention.

Lastly, regarding the pre-registration on using individual income levels in our analyses, we were initially of the impression that our data would include this measure because the Danish dataset (which we collected for the project) did. However, this was not the case. That is, we only had access to our own dataset when we drafted the pre-registration to use the entire dataset from the 67 countries and thus did not anticipate this lack of data on individual household income. We have clearly noted this deviation from the pre-registration in the method section of the revised manuscript (p. 23, line 7-19).

3. *I also have concerns about the key outcome measures related to morality. In the introduction it sounds as if the authors are weighing in on the debate about whether higher SES makes people more or less “moral”, meaning more or less inclined to do things that are widely regarded as morally good (e.g. being generous) or morally bad (e.g. cheating). But only the generosity measure falls into this category. And here the authors rely on self-report of generosity, which could vary enormously by culture and/or be generally unreliable. At the very least, one would want to corroborate these subjective self-reports of generosity with country-level data on things like actual charitable giving. And for the other measures, it’s not clear whether they exhibit the same kind of local/cultural relativity that the SES ladder measures do. For example, what counts as having a large vs. small moral circle in Nigeria vs. the US?*

Thank you for this comment, which has made it clear to us to that the central concepts from moral psychology used in this manuscript needs to be more clearly described. In response to your comment, we would like to note that a recent evolutionary conceptualization of morality argues that “*morality consists of a collection of biological and cultural solutions to the problems of cooperation recurrent in human social life*” (Curry et al. 2021, p. 2). Hence, while morality concerns considerations on what is “right” and “wrong,” which could either facilitate or hurt cooperation, we specifically argue that the current investigation is more nuanced than that. In other words, because we assess different underlying constructs of what constitutes morality, we are both weighing in on the discussion on whether lower rather than higher SES makes people exhibit more or less prosocial moral intentions, but we also include a selection of other measures (our three remaining DVs) that aim to capture more general aspects of moral judgment and decision-making. These measures concern what individuals *perceive* as being morally good or bad or how they perceive themselves as exhibiting such moral thoughts and behavior towards others. Indeed, our measure of Moral Identity concerns how individuals perceive themselves as moral individuals and how they exhibit such characteristics in their daily life, whereas our measure of Morality-as-Cooperation examines the weight of importance that individuals attribute to a set of seven generalized moral behaviors (e.g., reciprocation) considered to be universally “moral behaviors” across societies (see Curry et al., 2019 testing this construct across 60 societies). Similarly, our measure of Moral Circle concerns the number of people that individuals are concerned about whether “right” or “wrong” is done toward them. Hence, as now clarified in the revised manuscript (p. 3, line 11-17; pp. 5-7, lines 21-2; pp. 18-20, lines 14-13), our study expands previous research considerably by not only

focusing on *one* form of morality (Prosocial Intentions), but instead including a selection of constructs that have been discussed as central to human morality.

The measure of Prosocial Intentions naturally vary across cultures, but we clearly account for that in our models by the use of random-intercepts in our multilevel modeling. As such, we show how individuals might self-report their prosocial intentions if they were informed that they will receive a median income in their respective economy, but we make this measure more reliable and valid *across* countries considering that we also incorporate national median income in every respective country. We argue that intention to donate to charity (i.e., helping others in need) can be considered a universal moral value across the world (e.g., Aknin et al., 2013). Hence, in defense of this measure, our results show that even after having accounted for cross-cultural differences, we find a significant negative association between SES and prosocial donation intentions (Table 1, p. 10 and Supplementary Tables S7-S10).

When it comes to the comment of corroborating the results of the current study using reports on actual charitable giving, we have added a discussion of this that outline how our findings align with prior literature as well as recent results from the World Giving Index (p. 18, line 14-24). Due to the nature of the data from the World Giving Index, it was not possible for us to augment our data with this additional data source. Yet, the most recent report on this index clearly shows that countries that are traditionally considered as non-WEIRD societies with large populations of people living with chronic scarcity seem to top the charts when it comes to prosocial behaviors such as charitable giving, helping strangers, and volunteering time. While being mindful of not overstating the implications of the current research, such findings seem to be in line with our obtained results. We sincerely hope this answers your concerns.

4. *Finally, it's not clear whether the preregistration is really a preregistration. The authors explain that "some" of the data have already been collected and then refer to this explanation: "The data has already been collected and is a part of an international project on moral psychology during the corona (COVID-19) pandemic. The data will initially be used in the paper for which it was collected." This doesn't tell us which data were collected prior to the pre-registration and which were not. And, most importantly, it doesn't say whether the authors were aware of the outcomes of these analyses prior to the pre-registration. Given that the analyses reported on in the paper differ from the analyses outlined in the pre-registration, it seems that they didn't know the outcomes in advance, which is good—except that we only know this because the authors deviated so significantly from their pre-registered plan. This is a rich data set, and it seems possible to me that the authors could analyze it in a way that makes more sense. However, the fact that the authors pre-registered a more sensible approach and then abandoned it suggests that this would lead to rather different conclusions. Likewise, it's possible that the data could be augmented with nation-level objective measures that would allow them to more directly address the questions they seem to want to answer in the introduction. But, in its present form, this paper has some very significant limitations.*

Thank you for this comment. Following the guidelines from the leading authors of the original project (van Bavel et al., 2022) researchers who wanted to use the dataset for secondary analysis were obliged to pre-register their research outline at <https://icsmp-covid19.netlify.app> using one of the current available options (e.g., AsPredicted or

OSF). This procedure was the same for all projects utilizing this dataset; see e.g., Cutler et al. (2021), published in *Nature Aging* (<https://www.nature.com/articles/s43587-021-00118-3>). In essence, prior to submitting the pre-registration, we did not have access to the cross-national dataset; rather, we only had access to the Danish sample which we collected for the overall study, which included 67 countries. That is, all data were collected before this project was initiated, but we did not have access to the dataset before we made the pre-registration. We have clarified this in the revised version of the manuscript (p. 23, line 7-19).

As outlined in the method section, we only deviated from the pre-registered analyses in terms of implementing more robust and advanced statistical analyses. In particular, the currently employed analytic approach is more sensible and robust than the one we pre-registered, because our used models clearly account for the natural clustering in the data (countries) as well as within and between-country differences (see e.g., Čepulić et al., 2021 or Randall et al., 2021 using similar approaches to analyze large, clustered datasets). That is, while linear mixed effects models rests on many of the same assumptions as fixed effects models (i.e., OLS), the inclusion of random effects allows the researcher to account for possible differences in intercepts and slopes per clustering in the data, hence for instance guarding against concluding on a possible Yule-Simpson effect. Yet, based on your thoughtful comment, we have expanded on this issue in the Methods section to further justify why our selected analytic strategy provides a more robust test of our theorizing (pp. 28-29, line 20-25).

Lastly, as outlined above, we have augmented our dataset with a macro-level measure on adjusted disposable net-income per capita from the World Bank to explore how this might affect our proposed relationship. Here, we find that our objective measure of scarcity goes in the same direction as the subjective one (see Supplementary Table S11). However, because the GINI index is highly correlated with this measure of adjusted disposable net-income per capita ($r = -0.44$, $P < .001$), this measure captures the variance attributed to the GINI index in our main models. Consequently, we included the models using the macro-level objective measure of adjusted disposable net-income in our robustness models.

We sincerely hope that the implemented changes will suffice to address the concerns you outlined in your review.

References:

- Aknin, L. B., Barrington-Leigh, C. P., Dunn, E. W., Helliwell, J. F., Burns, J., Biswas-Diener, R., Kemeza, I., Nyende, P., Ashton-James, C. E., & Norton, M. I. (2013). Prosocial spending and well-being: cross-cultural evidence for a psychological universal. *Journal of Personality and Social Psychology*, 104(4), 635.
- Asparouhov, T., & Muthén, B. (2014). Multiple-Group Factor Analysis Alignment. *Structural Equation Modeling: A Multidisciplinary Journal*, 21(4), 495-508.
- Cannon, C., Goldsmith, K., & Roux, C. (2019). A self-regulatory model of resource scarcity. *Journal of Consumer Psychology*, 29(1), 104-127.
- Čepulić, D. B., Travaglino, G. A., Chrona, S., Uzelac, E., Jeftić, A., Reyna, C., & Kowal, M. (2021). Iron fists and velvet gloves: Investigating the associations between the

stringency of governments' responses to COVID-19, stress, and compliance in the early stages of the pandemic. *British Journal of Social Psychology*.

- Cirino, P. T., Chin, C. E., Sevcik, R. A., Wolf, M., Lovett, M., & Morris, R. D. (2002). Measuring Socioeconomic Status: Reliability and Preliminary Validity for Different Approaches. *Assessment*, 9(2), 145-155. <https://doi.org/10.1177/10791102009002005>
- Crimston, C. R., Hornsey, M. J., Bain, P. G., & Bastian, B. (2018). Toward a Psychology of Moral Expansiveness. *Current Directions in Psychological Science*, 27(1), 14-19. <https://doi.org/10.1177/0963721417730888>
- Curry, O. S., Alfano, M., Brandt, M. J., & Pelican, C. (2021). Moral molecules: Morality as a combinatorial system. *Review of Philosophy and Psychology*, 1-20.
- Curry, O., Whitehouse, H., & Mullins, D. (2019). Is it good to cooperate? Testing the theory of morality-as-cooperation in 60 societies. *Current Anthropology*, 60(1).
- Cutler, J., Nitschke, J. P., Lamm, C., & Lockwood, P. L. (2021). Older adults across the globe exhibit increased prosocial behavior but also greater in-group preferences. *Nature Aging*, 1(10), 880-888.
- de Bruijn, E.-J., & Antonides, G. (2021). Poverty and economic decision making: a review of scarcity theory. *Theory and Decision*, 1-33.
- Diemer, M. A., Mistry, R. S., Wadsworth, M. E., López, I., & Reimers, F. (2013). Best practices in conceptualizing and measuring social class in psychological research. *Analyses of Social Issues and Public Policy*, 13(1), 77-113.
- Giatti, L., Camelo, L. d. V., Rodrigues, J. F. d. C., & Barreto, S. M. (2012). Reliability of the MacArthur scale of subjective social status-Brazilian Longitudinal Study of Adult Health (ELSA-Brasil). *BMC public health*, 12(1), 1-7.
- Götz, F. M., Stieger, S., Gosling, S. D., Potter, J., & Rentfrow, P. J. (2020). Physical topography is associated with human personality. *Nature Human Behaviour*, 1-10.
- Graham, J., Waytz, A., Meindl, P., Iyer, R., & Young, L. (2017). Centripetal and centrifugal forces in the moral circle: Competing constraints on moral learning. *Cognition*, 167, 58-65.
- Mani, A., Mullainathan, S., Shafir, E., & Zhao, J. (2013). Poverty impedes cognitive function. *Science*, 341(6149), 976-980.
- Pavlović, T. A., F.; De, K.; Maglić, M.; Donnelly Kehoe, P. A.; Payán-Gómez, C.; ... van Bavel, J. J. . (2022). Predicting attitudinal and behavioral responses to COVID-19 pandemic using machine learning. *PNAS Nexus* [Accepted].
- Pickett, K. E., & Wilkinson, R. G. (2015). Income inequality and health: a causal review. *Social science & medicine*, 128, 316-326.

- Randall, A. K., Leon, G., Basili, E., Martos, T., Boiger, M., Baldi, M., Hocker, L., Kline, K., Masturzi, A., Aryeetey, R., Bar-Kalifa, E., Boon, S. D., Botella, L., Burke, T., Carnelley, K. B., Carr, A., Dash, A., Fitriana, M., Gaines, S. O., . . . Chiarolanza, C. (2021). Coping with global uncertainty: Perceptions of COVID-19 psychological distress, relationship quality, and dyadic coping for romantic partners across 27 countries. *Journal of Social and Personal Relationships*, 39(1), 3-33. <https://doi.org/10.1177/02654075211034236>
- Shah, A. K., Mullainathan, S., & Shafir, E. (2012). Some consequences of having too little. *Science*, 338(6107), 682-685.
- Van Bavel, J. J., Cichocka, A., Capraro, V., Sjøstad, H., Nezlek, J. B., Pavlović, T., Alfano, M., Gelfand, M. J., Azevedo, F., Birtel, M. D., Cislak, A., Lockwood, P. L., Ross, R. M., Abts, K., Agadullina, E., Aruta, J. J. B., Besharati, S. N., Bor, A., Choma, B. L., . . . Boggio, P. S. (2022). National identity predicts public health support during a global pandemic. *Nature Communications*, 13(1), 517.
- Waytz, A., Iyer, R., Young, L., Haidt, J., & Graham, J. (2019). Ideological differences in the expanse of the moral circle. *Nature Communications*, 10(1), 1-12.
- Wilkinson, R., & Pickett, K. (2011). *The spirit level: Why greater equality makes societies stronger*. Bloomsbury Publishing USA. New York.
-

Reviewers' Comments:

Reviewer #1:

Remarks to the Author:

I am happy with the revisions: the authors have done pretty much all the things I suggested they do, and to my mind they make the picture clearer. My only remaining comments are a few typos (important to fix these):

Line 49: Experiences of chronic relative economic scarcity is a structural characteristic of modern...grammar, either experience is singular or are is plural.

Table 1. There is a -0.8 in the first row that I am sure must be supposed to be -0.08.

Table 2. Aren't these random slopes rather than random intercepts? If I understand correctly, each country should have a random intercept and a random slope term, and it is the random slope term whose sign matters here. I am bit puzzled these numbers are so evenly split between positive and negative. Could you check this, and also check by running the regressions in each country separately and just seeing how many individual slopes are negative and how many positive (should give similar picture).

Reviewer #2:

Remarks to the Author:

This is a revised version of a manuscript that reports the results of a survey conducted across 60+ countries on the association between subjective SES and four different measures of morality. The article has improved since the last version by addressing several of the reviewer comments. However, there remains an outstanding limitation of the paper that the authors are unable to address in the revision, and that is the reliance on the measure of subjective SES as an operationalization of (relative) resource scarcity.

The introduction draws from a literature that has focused on resource scarcity and its implications for cognition and behavior. The authors argue in favor of their measure of subjective SES as an operationalization of resource scarcity by saying "...prior research has found that experiences of relative scarcity can shift cognitive attention and alter decision-making strategies more than extreme and thus absolute scarcity (i.e., extreme poverty). Therefore, we rely on subjective SES to measure individual-level experiences of relative economic scarcity to gain a more nuanced understanding of the link between scarcity experiences and morality." First, in the above sentences "experiences" is used to refer to "subjective evaluations" of resource scarcity, and I find this a bit misleading, because subjective evaluations of SES or resource scarcity, may not always correlate with the actual, objective experience of resource scarcity. Second, the article does not seem to acknowledge possible confounds with individual differences in the measure of subjective SES (e.g., honesty-humility, conscientiousness), which could account for some of the relations observed between subjective SES and measures of morality. Furthermore, in the argument above, the authors draw attention to the subjective measure having a strong association with outcomes, compared to the objective measure. However, I don't find this to be a convincing argument to study this variable. Instead, I think the arguments should be based on the validity of the measure. The authors attempt to resolve the issue of the reliance on a subjective measure of SES by including an objective societal measure of wealth inequality across societies. However, It was not clear to me how the theories about resource scarcity and thinking/behavior in the introduction are also tested by cross-societal differences in the GINI index. The authors state: "By focusing directly on the dispersion of financial resources, this is a more objectively oriented national-level indicator of experiences of relative economic scarcity. That is, this indicator constitutes a measurement of the magnitude of differences in wealth accumulation that individuals in a specific economy are exposed to, which has been discussed as an important factor linked to psychological differences in experiences of relative economic scarcity". The GINI indexes wealth discrepancies in society, and I would think that in societies with high wealth inequality, people don't often interact with others who have all the resources, and instead their comparisons with others would be comparisons with people who have relatively similar resources at their disposal. I would prefer to see (1) the authors reporting data that supports the idea that subjective SES is correlated with GINI, and (2) more

direct discussion about the strengths and limitations in how this cross-societal index is being used to test the theories in the introduction of the paper. Still, I think actual household (or individual) income would be a better objective indicator of resource deprivation, and for testing the ideas outlined in the introduction.

The authors conclude "we find evidence for the notion that experiences of relative economic scarcity as a chronic state is not associated with a "depletion" of moral character or less prosocial intentions. Instead, it seems that such experiences of economic scarcity are associated with a stronger preference for identifying and acting as a moral individual, engaging in cooperative behaviours with a clear moral foundation (e.g., helping kin or reciprocating), and having the intention to engage in prosocial charitable giving." Similarly, in the discussion the authors state: "... individuals who experience relative economic scarcity as a chronic state not only seem more inclined to perceive themselves as moral individuals (i.e., Moral Identity), but also seek to project such morality-related aspects towards their peers and in-group members (i.e., Morality-as-Cooperation, Moral Circle, and Prosocial Intentions)." These conclusions are a bit misleading, because the authors did not measure the actual experience of economic scarcity, but how people subjectively think about their SES, and I think this should be made much more explicit in the discussion of the paper.

Overall, I think the authors do report a study that adds value to this developing literature. I think the paper should be published, and should be given serious consideration by scholars working on this topic. That said, I question whether the methods used to operationalize the variables in this study provide a strong test of the ideas outlined in the introduction. Furthermore, the authors could draw more attention to the limitations of using subjective SES as an index of chronic resource scarcity. For these reasons, the paper may be more suitable for a specialty journal.

Reviewer #3:

Remarks to the Author:

I think the authors have done an excellent job of addressing the concerns raised in my original review and that the manuscript has also benefitted from suggestions from the other reviewers.

I should note, with apologies, that my main concern about the original manuscript was based on a misunderstanding. I thought that the authors were primarily interested in objective economic scarcity. Although they were careful to say "relative" scarcity in many places, much of the literature referenced seems to focus on objective scarcity. This framing led me to believe that objective deprivation was the key variable, despite it's being accessed through self-report. I now understand that subjective scarcity is the primary target, and I think the revised manuscript makes this more clear. But this makes me wonder whether much of the debate that frames this paper can be resolved by distinguishing between objective and subjective scarcity. I'm not sure, and I leave it to the authors to add something about this if they think it would be illuminating.

Revision Notes

We would like to thank the three reviewers for the valuable comments to our manuscript. Below are our point-by-point responses to all comments. The comments from the reviewers are numbered and italicized. Our responses appear in regular type.

Reviewer 1

- 1. I am happy with the revisions: the authors have done pretty much all the things I suggested they do, and to my mind they make the picture clearer. My only remaining comments are a few typos (important to fix these): Line 49: Experiences of chronic relative economic scarcity is a structural characteristic of modern...grammar, either experience is singular or are is plural. Table 1. There is a -0.8 in the first row that I am sure must be supposed to be -0.08. Table 2. Aren't these random slopes rather than random intercepts? If I understand correctly, each country should have a random intercept and a random slope term, and it is the random slope term whose sign matters here. I am bit puzzled these numbers are so evenly split between positive and negative. Could you check this, and also check by running the regressions in each country separately and just seeing how many individual slopes are negative and how many positive (should give similar picture).*

We thank you for your positive evaluation of the revised manuscript and for noting these typos, which we have corrected in the revised version of the manuscript. We have changed the sentence in line 49 (p. 3, line 1) and corrected the number in Table 1. In Table 2, these were random intercepts. As specified in our Methods section, we only specified the models reported in Table 1 with random intercepts. We did so because the models were unable to converge successfully with random slopes, which was also the reason why we decided to supplement our main analysis with the Nested OLS models reported in the Supplementary Results, as well as the models separating the within/between country effects as suggested by yourself (the models reported in Figure 2 are based on these). We agree that the country-level slope terms make more sense to report here. Therefore, we now report the direction of the slopes for the Nested OLS models (equivalent to a random slope setup) in Table 2. We have double-checked the numbers and made sure everything is reported correctly. Again, thank you for your careful consideration.

Reviewer 2

- 1. This is a revised version of a manuscript that reports the results of a survey conducted across 60+ countries on the association between subjective SES and four different measures of morality. The article has improved since the last version by addressing several of the reviewer comments. However, there remains an outstanding limitation of the paper that the authors are unable to address in the revision, and that is the reliance on the measure of subjective SES as an operationalization of (relative) resource scarcity.*

We thank you for your careful review of our revised manuscript. We are pleased that you believe our manuscript has been improved. Below we outline our responses to your remaining comments with direct references to changes in the revised manuscript. We sincerely hope that these revisions satisfactory address your concerns.

2. *The introduction draws from a literature that has focused on resource scarcity and its implications for cognition and behavior. The authors argue in favor of their measure of subjective SES as an operationalization of resource scarcity by saying “...prior research has found that experiences of relative scarcity can shift cognitive attention and alter decision-making strategies more than extreme and thus absolute scarcity (i.e., extreme poverty). Therefore, we rely on subjective SES to measure individual-level experiences of relative economic scarcity to gain a more nuanced understanding of the link between scarcity experiences and morality.” First, in the above sentences “experiences” is used to refer to “subjective evaluations” of resource scarcity, and I find this a bit misleading, because subjective evaluations of SES or resource scarcity, may not always correlate with the actual, objective experience of resource scarcity. Second, the article does not seem to acknowledge possible confounds with individual differences in the measure of subjective SES (e.g., honesty-humility, conscientiousness), which could account for some of the relations observed between subjective SES and measures of morality. Furthermore, in the argument above, the authors draw attention to the subjective measure having a strong association with outcomes, compared to the objective measure. However, I don’t find this to be a convincing argument to study this variable. Instead, I think the arguments should be based on the validity of the measure.*

We thank you for highlighting that the conceptualization of subjective economic scarcity needed to be further clarified and differentiated from objective experiences of scarcity. We acknowledge that “experiences” used to describe “subjective evaluations” of economic scarcity may be confused with objective circumstances of living in resource scarcity. For reasons of conceptual clarity, we now use the concepts “subjective experiences of economic scarcity” and “perceptions of economic scarcity” throughout the entire manuscript (as in, for example, Jachimowicz et al., 2022) to denote subjective evaluations of resource scarcity. This approach follows previous investigations (e.g., Kraus et al., 2009) as well as the current bias-free language guidelines from the American Psychological Association (APA) on socioeconomic status¹, which details that “Socioeconomic status (SES) encompasses not only income but also educational attainment, occupational prestige, and subjective perceptions of social status and social class.” We also clearly outline both in the abstract (lines 2-4) and in the introduction (lines 1-5) how subjective experiences of scarcity refer to a subjective feeling of lack or economic resources compared to others – that is, a result of social comparison – and that it is different from the actual circumstances of living in economic scarcity. With these revisions, the introduction now conveys more clearly that subjective SES is used in the literature an indicator of these subjective experiences of resource scarcity (lines 5-7). We return to this point and elaborate on it at p 7. (lines 17-23), where we argue for the importance of investigating subjective rather than objective perceptions of economic scarcity. We also discuss how subjective experiences of economic scarcity are different from objective circumstances of economic scarcity (see also Tan et al., 2020), while underscoring these points in our final Discussion (p. 24, lines 1-8).

We acknowledge that our measure of subjective SES, like any other individual-level measure, could be influenced by other individual differences, such as the personality dimensions you outline. We have now reflected on this point in our concluding Discussion (p. 25, lines 1-11) to explicitly address your thoughtful remark.

¹ <https://apastyle.apa.org/style-grammar-guidelines/bias-free-language/socioeconomic-status>

Unfortunately, we do not have available individual-level data on, for instance, honesty-humility or conscientiousness to control for these factors in our analyses. Therefore, we now emphasize that future work should examine our studied associations in more realistic environments, preferably using behavioral measures and experimental approaches or a larger range of control variables, including personality dimensions such as conscientiousness to allow for causal inferences (p. 25, lines 8-11).

The use of the MacArthur ladder as a way to measure subjective socioeconomic standing in society is a reliable and well-established measure, which has shown strong construct validity (Cundiff et al., 2013) as well as high predictive validity in previous research regarding outcomes associated with health and well-being (e.g., Boon & Farnsworth, 2011; Kuper & Marmot, 2003; Singh-Manoux, et al., 2005). Further, recent work has argued that a reason for this high predictive validity is that the MacArthur ladder scale is able to measure two distinct constructs that capture inherent characteristics of how people perceive and subjectively experience economic scarcity: economic circumstances and social status (Galvan et al. 2022). Thus, the MacArthur ladder scale should represent a valid way to measure subjective SES, which we have sought to clarify in this revision (pp. 7-8, lines 23-5). While we agree that using an objective measure of economic scarcity could also provide an interesting theoretical contribution to the current literature on scarcity, the current work is focused on how subjective experiences of economic scarcity might be associated with morality. To be responsive, however, we now also discuss the need for further studies on objective measures of economic scarcity as a clear avenue for future research on moral judgment and decision-making (p. 24, lines 1-11; p. 24, lines 12-19).

- 3. The authors attempt to resolve the issue of the reliance on a subjective measure of SES by including an objective societal measure of wealth inequality across societies. However, It was not clear to me how the theories about resource scarcity and thinking/behavior in the introduction are also tested by cross-societal differences in the GINI index. The authors state: “By focusing directly on the dispersion of financial resources, this is a more objectively oriented national-level indicator of experiences of relative economic scarcity. That is, this indicator constitutes a measurement of the magnitude of differences in wealth accumulation that individuals in a specific economy are exposed to, which has been discussed as an important factor linked to psychological differences in experiences of relative economic scarcity”. The GINI indexes wealth discrepancies in society, and I would think that in societies with high wealth inequality, people don’t often interact with others who have all the resources, and instead their comparisons with others would be comparisons with people who have relatively similar resources at their disposal. I would prefer to see (1) the authors reporting data that supports the idea that subjective SES is correlated with GINI, and (2) more direct discussion about the strengths and limitations in how this cross-societal index is being using to test the theories in the introduction of the paper. Still, I think actual household (or individual) income would be a better objective indicator of resource deprivation, and for testing the ideas outlined in the introduction.*

Thank you for this comment. We would like to begin our reply by noting that we do not attempt to “resolve” any issues related to using a subjective measure of SES. The contribution of this manuscript is aimed at understanding how perceptions and subjective experiences of economic scarcity measured at the individual-level (by

subjective SES) and at the macro-level (by the GINI coefficient) is associated with the four dependent measures of moral judgment. As now clarified in the revised version of the manuscript, we include GINI as a macro-level indicator of perceptions of economic scarcity, following previous investigations in this domain (e.g., Jachimowicz et al. 2020; Sommet et al., 2018). That is, the GINI index is utilized as an indicator of the exposure to salient differences in financial resources leading to different perceptions of economic scarcity. To make this more explicit, we have expanded on this argument in our revision (pp. 8-9, lines 6-2). This argument is in line with previous work, which has shown that economic inequality impairs the psychological health of people facing scarcity (Sommet et al., 2018), because a central component of perceptions of economic scarcity is rooted in social comparison (Goldsmith et al. 2020) – something that is exacerbated by economic inequality (Cheung & Lucas, 2016). In other words, one of the central components of scarcity theory is that the behavioral manifestations resulting from scarcity are rooted in the subjective component of perceiving to “have less than what is needed” and to perceive to be worse off, in terms of available resources, than others (Ren et al., 2022; Shah et al. 2012).

To address your thoughts about whether individuals living in contexts of high (vs. low) inequality perceive economic scarcity to be more prevalent, we would like to raise the following points. First, we agree that this is an interesting avenue for further investigations, which we now also note in the section on directions for future research (p. 24, lines 12-19). Second, while we cannot directly measure social comparisons in the current data, it is worth mentioning that people living in places of high economic inequality are regularly exposed to notable differences in resource acquisition according to previous research (e.g., Cheung & Lucas, 2016; Sands & de Kadt, 2020). As an example, from the most economically unequal country in the world (South Africa), labor market² and internet usage³ data from the international statistics organization Statista reveal that close to 1.2 million South Africans work in private households (e.g., as servants or housekeepers); more than 800,000 women are domestic workers; and over 80% of South Africans have access to the internet. Hence, it seems plausible that low SES individuals in South Africa are regularly exposed to (and compare themselves with) fellow South Africans with greater economic resources than themselves. Accordingly, the claim that individuals living in highly economically unequal places are not exposed to differences in wealth on a regular basis is a claim that we and other scholars do not necessarily agree with (Buttrick & Oishi, 2017; Cheung & Lucas, 2016). Moreover, individual-level measures of the distribution of subjective socioeconomic status are not necessarily highly correlated with the macro-level measure of GINI (e.g., Di Domenico & Fournier, 2014).

Concerning your point related to more objective indicators of resource deprivation, we have further elaborated on the strengths and limitations associated with addressing our main objective through subjective experiences of economic scarcity (p. 24, lines 12-19). As noted in one of our replies above, we agree that using objective household income could provide an interesting investigation into how objective experiences of resource scarcity might affect morality or judgment and decision-making in general, which we have carefully outlined in the Discussion section when articulating our directions for future research (p. 24, lines 1-11). Having noted that, the current

² <https://www.statista.com/statistics/1129815/number-of-people-employed-in-south-africa-by-industry/>

³ <https://www.statista.com/statistics/972866/south-africa-mobile-internet-penetration/>

manuscript provides a robust investigation of the association between individual and macro-level perceptions of economic scarcity and how these aspects are associated with moral judgment across 67 countries.

4. *The authors conclude “we find evidence for the notion that experiences of relative economic scarcity as a chronic state is not associated with a “depletion” of moral character or less prosocial intentions. Instead, it seems that such experiences of economic scarcity are associated with a stronger preference for identifying and acting as a moral individual, engaging in cooperative behaviours with a clear moral foundation (e.g., helping kin or reciprocating), and having the intention to engage in prosocial charitable giving.” Similarly, in the discussion the authors state: “... individuals who experience relative economic scarcity as a chronic state not only seem more inclined to perceive themselves as moral individuals (i.e., 320 Moral Identity), but also seek to project such morality-related aspects towards their peers and in-group members (i.e., Morality-as-Cooperation, Moral Circle, and Prosocial Intentions).” These conclusions are a bit misleading, because the authors did not measure the actual experience of economic scarcity, but how people subjectively think about their SES, and I think this should be made much more explicit in the discussion of the paper.*

We thank you for highlighting that the conclusions needed further precision. Referring to our reply to your second comment, we have carefully revised the manuscript to clarify that we refer to subjective experiences of economic scarcity as measured by subjective SES at the individual level and income inequality at the macro level. We have also clearly defined in the Introduction what we mean by subjective experiences of economic scarcity and how these experiences differ from objective economic scarcity. Based on your valuable input, we now write the following in the section that you mentioned: “...we find evidence for the notion that subjective experiences of economic scarcity are not associated with a “depletion” of moral character or less prosocial intentions. Instead, it seems that such subjective experiences are associated with a stronger preference for identifying and acting as a moral individual, engaging in cooperative behaviors with a clear moral foundation (e.g., helping kin or reciprocating), and having the intention to engage in prosocial charitable giving.” (p. 19, lines 9-14).

5. *Overall, I think the authors do report a study that adds value to this developing literature. I think the paper should be published, and should be given serious consideration by scholars working on this topic. That said, I question whether the methods used to operationalize the variables in this study provide a strong test of the ideas outlined in the introduction. Furthermore, the authors could draw more attention to the limitations of using subjective SES as an index of chronic resource scarcity. For these reasons, the paper may be more suitable for a specialty journal.*

Again, thank you for your careful re-evaluation of our manuscript. We are happy to learn that you think this study adds value to the rapidly developing literature in this domain. We sincerely hope that our revisions have made it clear how the current operationalizations of our focal variables allow us to test the role that subjective experiences of economic scarcity have in shaping moral judgment and decision-making. As noted above, we have also added limitations associated with the current research, while simultaneously suggesting several fruitful avenues for further studies (pp. 24-25, lines 1-11).

References

- Boon, B., & Farnsworth, J. (2011). Social Exclusion and Poverty: Translating Social Capital into Accessible Resources. *Social Policy & Administration*, 45(5), 507-524.
- Cheung, F., & Lucas, R. E. (2016). Income inequality is associated with stronger social comparison effects: The effect of relative income on life satisfaction. *Journal of Personality and Social Psychology*, 110(2), 332.
- Cundiff, J. M., Smith, T. W., Uchino, B. N., & Berg, C. A. (2013). Subjective Social Status: Construct Validity and Associations with Psychosocial Vulnerability and Self-Rated Health. *International Journal of Behavioral Medicine*, 20(1), 148-158.
- Di Domenico, S. I., & Fournier, M. A. (2014). Socioeconomic Status, Income Inequality, and Health Complaints: A Basic Psychological Needs Perspective. *Social Indicators Research*, 119(3), 1679-1697.
- Galvan, M. J., Payne, K., Hannay, J., Georgeson, A., & Muscatell, K. (2022). What does the MacArthur Scale of Subjective Social Status Measure? Separating Economic Circumstances and Social Status to Predict Health. Preprint at *PsyArxiv*: <https://psyarxiv.com/e9px3/>
- Goldsmith, K., Griskevicius, V., & Hamilton, R. (2020). Scarcity and Consumer Decision Making: Is Scarcity a Mindset, a Threat, a Reference Point, or a Journey? *Journal of the Association for Consumer Research*, 5(4), 358-364.
- Jachimowicz, J. M., Davidai, S., Goya-Tocchetto, D., Szaszi, B., Day, M. V., Tepper, S. J., Phillips, L. T., Mirza, M. U., Ordabayeva, N., & Hauser, O. P. (2022). Inequality in researchers' minds: Four guiding questions for studying subjective perceptions of economic inequality. *Journal of Economic Surveys*.
- Kuper, H., & Marmot, M. (2003). Job Strain, Job Demands, Decision Latitude, and Risk of Coronary Heart Disease Within the Whitehall II Study. *Journal of Epidemiology & Community Health*, 57(2), 147-153.
- Kraus, M. W., Piff, P. K., & Keltner, D. (2009). Social Class, Sense of Control, and Social Explanation. *Journal of Personality and Social Psychology*, 97(6), 992.
- Ren, M., Zou, S., Zhu, S., Shi, M., Li, W., & Ding, D. (2022). Do the poor envy others more? The effects of scarcity mindset on envy. *Current Psychology*, 1-17.
- Sands, M. L., & de Kadt, D. (2020). Local exposure to inequality raises support of people of low wealth for taxing the wealthy. *Nature*. 586(7828), 257-261.
- Singh-Manoux, A., Marmot, M. G., & Adler, N. E. (2005). Does Subjective Social Status Predict Health and Change in Health Status Better than Objective Status? *Psychosomatic Medicine*, 67(6), 855-861.

Sommet, N., Morselli, D., & Spini, D. (2018). Income Inequality Affects the Psychological Health of Only the People Facing Scarcity. *Psychological Science*, 29(12), 1911-1921.

Reviewer 3

1. *I think the authors have done an excellent job of addressing the concerns raised in my original review and that the manuscript has also benefitted from suggestions from the other reviewers.*

I should note, with apologies, that my main concern about the original manuscript was based on a misunderstanding. I thought that the authors were primarily interested in objective economic scarcity. Although they were careful to say “relative” scarcity in many places, much of the literature referenced seems to focus on objective scarcity. This framing led me to believe that objective deprivation was the key variable, despite it’s being accessed through self-report. I now understand that subjective scarcity is the primary target, and I think the revised manuscript makes this more clear. But this makes me wonder whether much of the debate that frames this paper can be resolved by distinguishing between objective and subjective scarcity. I’m not sure, and I leave it to the authors to add something about this if they think it would be illuminating.

Thank you for your careful re-evaluation of our paper. We are very pleased to learn that you think we have done a satisfactory job in addressing the concerns that you and the other two reviewers raised in the first revision. We thank you for your helpful suggestion to carefully distinguish between objective and subjective scarcity. Based on your suggestion, we now use the concepts “subjective experiences of economic scarcity” and “perceptions of economic scarcity” consistently throughout the manuscript, and discuss how such perceptions and experiences differ from objective scarcity circumstances. We have also expanded on this issue in the directions for future research, by urging scholars to disentangle the extent to which subjective and objective measures of economic scarcity are interrelated and predictive of psychological mechanisms across different domains (p. 24, lines 1-11). Again, thank you for your constructive comments.

Reviewers' Comments:

Reviewer #2:

Remarks to the Author:

I appreciate that the authors addressed my previous set of comments. The paper is now more explicit about focusing on subjective perceptions of SES and draws attention to some limitations of this approach.

That said, I was puzzled that the authors still frame GINI (wealth inequality) as an operationalization of country level perceptions of SES. The authors state the "manuscript provides a robust investigation of the association between individual and macro-level perceptions of economic scarcity and how these aspects are associated with moral judgment across 67 countries." This would imply that in societies with higher wealth inequality, people would be making more social comparisons with others who have greater resources and therefore indicate lower subjective SES. I previously asked the authors to report the association between their measure of subjective SES and GINI. I didn't see this statistic reported in the current manuscript or in the response to reviewers. Instead, in the response to reviewers the authors claim, "individual-level measures of the distribution of subjective socioeconomic status are not necessarily highly correlated with the macrolevel measure of GINI (e.g., Di Domenico & Fournier, 2014)." I went to the paper the authors cite in this claim, and this research reports a non-significant positive relation between GINI and subjective SES, and so higher feelings of own SES in areas with higher wealth inequality. This finding is the opposite of what the authors argue here, which is that higher wealth inequality should be associated with lower subjective perceptions of SES.

If the authors are going to consider GINI as an operationalization of subjective perceptions of SES at the country level, I would think it would be beneficial to offer existing evidence to support that claim within their own dataset. Specifically, I continue to encourage the authors to report the association between GINI and subjective perceptions of SES in their own study, and report and discuss the outcome of that analysis in the main text of the paper. At this point, I am still not convinced that wealth inequality causes societal differences in macro-level perceptions of relative economic scarcity, and therefore skeptical that the outcomes of the associations between GINI and the measures of morality can be interpreted as the result of subjective perceptions of relative scarcity.

Reviewer #3:

Remarks to the Author:

As indicated by my second round review, I think this manuscript is excellent and that it has been much improved by the review process.

Revision Notes

We thank the reviewers for the valuable comments to our manuscript. Below are our point-by-point responses to all comments. The comments from the reviewers are numbered and italicized. Our responses appear in regular type.

Reviewer 2

- 1. I appreciate that the authors addressed my previous set of comments. The paper is now more explicit about focusing on subjective perceptions of SES and draws attention to some limitations of this approach.*

Thank you for your careful evaluation of our work. We are happy to learn that you think the manuscript has improved after the last round of revisions. Below, we outline our responses to your additional comments, with direct references to changes in the manuscript.

- 2. That said, I was puzzled that the authors still frame GINI (wealth inequality) as an operationalization of country level perceptions of SES. The authors state the “manuscript provides a robust investigation of the association between individual and macro-level perceptions of economic scarcity and how these aspects are associated with moral judgment across 67 countries.” This would imply that in societies with higher wealth inequality, people would be making more social comparisons with others who have greater resources and therefore indicate lower subjective SES. I previously asked the authors to report the association between their measure of subjective SES and GINI. I didn’t see this statistic reported in the current manuscript or in the response to reviewers. Instead, in the response to reviewers the authors claim, “individual-level measures of the distribution of subjective socioeconomic status are not necessarily highly correlated with the macrolevel measure of GINI (e.g., Di Domenico & Fournier, 2014).” I went to the paper the authors cite in this claim, and this research reports a non-significant positive relation between GINI and subjective SES, and so higher feelings of own SES in areas with higher wealth inequality. This finding is the opposite of what the authors argue here, which is that higher wealth inequality should be associated with lower subjective perceptions of SES. If the authors are going to consider GINI as an operationalization of subjective perceptions of SES at the country level, I would think it would be beneficial to offer existing evidence to support that claim within their own dataset. Specifically, I continue to encourage the authors to report the association between GINI and subjective perceptions of SES in their own study, and report and discuss the outcome of that analysis in the main text of the paper. At this point, I am still not convinced that wealth inequality causes societal differences in macro-level perceptions of relative economic scarcity, and therefore skeptical that the outcomes of the associations between GINI and the measures of morality can be interpreted as the result of subjective perceptions of relative scarcity.*

Thank you for this comment. First, we would like to note that we frame GINI as an operationalization of country-level perceptions of economic scarcity and not country-level perceptions of SES, as you also point out in the referenced quote from our manuscript. We have clarified this on page 7, lines 12-13, and page 8, lines 14-15, in the revised manuscript. Our approach builds on prior work arguing that economic inequality can elicit perceptions of economic scarcity (Buttrick & Oishi, 2017;

Jachimowicz et al., 2020; Kondo et al., 2008; Sommet et al., 2019; Wilkinson & Pickett, 2006). We have also emphasized that further in the revised manuscript (p. 8, lines 13-14).

We have also provided further justifications for using GINI as an indirect measure of macro-level differences in perceptions of relative economic scarcity (p. 8, lines 17-22, and pp. 8-9, lines 22-2), while also acknowledging some limitations of doing so (p. 25, lines 12-21).

Moreover, as you suggested, we have reported the overall correlation between GINI and SES in our data ($r = -0.07$, $P < 0.001$; p. 8, lines 24-25) when discussing the use of GINI as an indirect macro-level indicator of perceptions of economic scarcity. The result affirms what you outlined in your comment, namely that "... in societies with higher wealth inequality, people would be making more social comparisons with others who have greater resources and therefore indicate lower subjective SES". To further corroborate this finding, we added an additional correlational analysis on the relationship between GINI and SES, where we split our measure of self-reported SES based on quantiles (25% = *Lower SES*, Median = *Medium SES*, 75% *Higher SES*), again finding a statistically significant correlation. This analysis affirms that GINI is correlated with differences in self-reported SES (p. 9, lines 1-2).

In summary, we sincerely hope that our revisions have clarified our grounds for using economic inequality as an indirect indicator of macro-level perceptions of relative economic scarcity, despite the discussed limitations of doing so.

References:

- Buttrick, N. R., & Oishi, S. (2017). The psychological consequences of income inequality. *Social and Personality Psychology Compass*, *11*(3), e12304.
- Jachimowicz, J. M., Szaszi, B., Lukas, M., Smerdon, D., Prabhu, J., & Weber, E. U. (2020). Higher economic inequality intensifies the financial hardship of people living in poverty by fraying the community buffer. *Nature Human Behaviour*, *4*(7), 702-712.
- Kondo, N., Kawachi, I., Subramanian, S., Takeda, Y., & Yamagata, Z. (2008). Do social comparisons explain the association between income inequality and health?: Relative deprivation and perceived health among male and female Japanese individuals. *Social Science & Medicine*, *67*(6), 982-987.
- Sommet, N., Elliot, A. J., Jamieson, J. P., & Butera, F. (2019). Income inequality, perceived competitiveness, and approach-avoidance motivation. *Journal of Personality*, *87*(4), 767-784.
- Wilkinson, R. G., & Pickett, K. E. (2006). Income inequality and population health: a review and explanation of the evidence. *Social Science & Medicine*, *62*(7), 1768-1784.

Reviewer 3

- 1. As indicated by my second round review, I think this manuscript is excellent and that it has been much improved by the review process.*

We would like to thank you again for your constructive and careful review of our manuscript. We believe that all your suggestions have strongly improved the current work.

Reviewers' Comments:

Reviewer #2:

Remarks to the Author:

I appreciate how the authors have addressed my previous comments, and I have no further comments on the manuscript.